# RETHINKING STRUCTURE LEARNING FOR GRAPH NEURAL NETWORKS

## ABSTRACT

To improve the performance of Graph Neural Networks (GNNs), Graph Structure Learning (GSL) has been extensively applied to reconstruct or refine original graph structures. While GSL is generally thought to improve GNN performance, it often leads to longer training times and more hyperparameter tuning. Besides, the distinctions among current GSL methods remain ambiguous from the perspective of GNN training, and there is a lack of theoretical analysis to quantify their effectiveness. Recent studies further suggest that GSL does not consistently outperform baseline GNNs under the same hyperparameter tuning. This motivates us to ask a critical question: *Is GSL really useful for improving GNN performance?* To address this question, we first propose a new GSL framework, which includes three steps: GSL bases (*i.e.,* node representations used to construct new graphs) construction, new structure construction, and view fusion, to better understand GSL. Then, our empirical studies and theoretical analysis show that the mutual information (MI) between node representations and labels does not increase after applying graph convolution on GSL graphs that are constructed by similarity, indicating GSL could be unnecessary in most cases. Our experiments fairly re-assess the performance of GSL and reveal that adding GSL to GNN baselines or removing GSL in state-of-the-art models has negligible impact on node classification accuracy. We also report that pretrained GSL bases, parameter separation, and early fusion are effective designs within GSL. Our findings challenge the necessity of complex GSL methods and underscore the value of simplicity in GNN design.

## 1 INTRODUCTION

Graph Neural Networks (GNNs) (Kipf & Welling, 2016) are effective in capturing structural information from non-Euclidean data, which can be used in many applications such as recommendation (Wu et al., 2022; 2019b), telecommunication (Lu et al., 2024a), bio-informatics (Zhang et al., 2021; Hua et al., 2024), and social networks (Li et al., 2023b; Luan et al., 2019). However, conventional GNNs suffer from issues including heterophily (Lu et al., 2024b; Luan et al., 2024a), over-squashing (Brody et al., 2021), adversarial attacks (Jin et al., 2020; Li et al., 2022a), and missing or noisy structures (Lao et al., 2022; Liu et al., 2022b). To address these issues, Graph Structure Learning (GSL) has been widely used (Zhu et al., 2021a), which reconstructs or refines the original graph structures to enhance the performance of GNNs. However, GSL brings more hyperparameters and adds plenty of computational cost in both the construction process and the learning process. In addition, recent studies (Luo et al., 2024; Platonov et al., 2023) have shown that GSL methods cannot consistently outperform traditional GNNs with the same hyperparameter tuning strategy. Therefore, an in-depth analysis of the effectiveness and necessity of GSL is highly needed.

First, to better understand GSL, we propose a comprehensive framework to carefully break down GSL into 3 steps: **(1) GSL Bases Generation.** GSL bases are the processed node embeddings used before constructing new structures, which are constructed by either graph-aware or graph-agnostic models with fixed or learnable parameters. **(2) New Structure Construction.** Based on the GSL bases, new structures are constructed with similarity-based (Pei et al., 2020; Jiang et al., 2019), structural-based (Zhao et al., 2020; Liu et al., 2022a), or optimization-based approaches (Jin et al., 2020). Then, graph refinements are followed. **(3) View Fusion.** To incorporate the original graph or combine multiple GSL-generated graphs, various view fusion strategies are applied, *e.g.,* late fusion

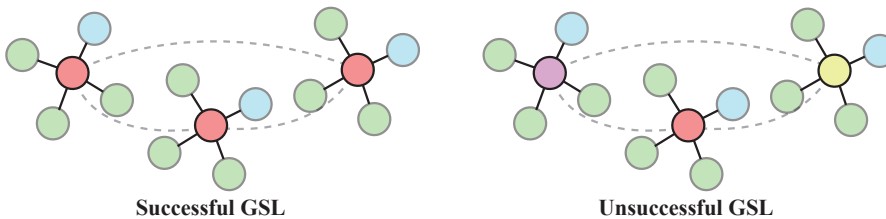

Figure 1: Examples of GSL that use neighbor distribution as GSL bases. The left shows a case of successful GSL, where new connections between red nodes are constructed using their GSL bases 3 green nodes and 1 blue node. The right shows a case of unsuccessful GSL, because the GSL bases of intra-class nodes are not consistent, and nodes with different classes are connected.

(Wang et al., 2021), early fusion (Li et al., 2022a), or separation (Liu et al., 2022b). Compared with the existing categorization of GSL (Zhu et al., 2021a;b; Kolbeck et al., 2022; Qiao et al., 2018) that mainly focuses on step (2), our proposed framework carefully disentangles each component in GSL, which enhances our understanding of GSL in GNNs.

With the above framework, we rethink when GSL is helpful in GNNs. As examples shown in Figure 1, a GSL method creates new connections between nodes with similar GSL bases, which is denoted as the contextual information of the ego node and its neighbors in this case. When the GSL bases show high consistency with intra-class nodes, nodes within the same class are connected, which is beneficial for GNNs (Luan et al., 2024b) and we denote it as **successful GSL**. Conversely, when the GSL bases show high consistency between inter-class nodes, nodes in different classes are likely to be connected, which is harmful to GNNs and we denote it as **unsuccessful GSL**. These examples highlight that the effectiveness of GSL is highly contingent on the quality of GSL bases. However, even if most GSL methods are supposed to be successful GSL, do we really need GSL in these cases? In this paper, our answer is **"No"**. The prerequisite of successful GSL is that the GSL bases are highly consistent within intra-class nodes, which inherently ensures a high quality of node representations (Kothapalli et al., 2024). Therefore, even a successful GSL is unnecessary because the GSL bases are already informative enough to provide distinguishable node embeddings.

Based on the above example, we empirically and theoretically analyze the effectiveness of similarity-based GSL, one of the most representative approaches in GSL. Our findings reveal that the mutual information (MI) between node representations and labels does not increase after applying graph convolution on similarity-based GSL graphs. Our results indicate that even though GSL sometimes outperforms GNNs in certain scenarios of heterophily (Pei et al., 2020; Luan et al., 2024c) or inconsistent neighbor distributions (Zheng et al., 2024a; Ma et al., 2021), its performance is still upper bound by Multilayer Perceptrons (MLP) on the same GSL bases in most cases. These results also explain why ego node separation (Zhu et al., 2020a) is an important part of model design. Our extensive experiments show that, under the same hyperparameter tuning and GSL bases, no matter adding GSL to 4 baseline GNNs or deleting GSL in 8 state-of-the-art (SOTA) GSL-based methods, there are no significant changes in model performance on node classification. Furthermore, we also show that while GSL fails to improve model performance, it does offer marginal improvements in model robustness. In conclusion, our main contributions are as follows:

- We propose a new framework to decompose the process of GSL into 3 steps, which is more comprehensive than the previous literature and helps better understand each component in GSL.

- Both of our empirical experiments and theoretical analysis show that the mutual information (MI) between node representations and labels does not increase after applying graph convolution on similarity-based GSL graphs, indicating that most GSL methods are unnecessary.

- We fairly re-evaluate the effectiveness of GSL by adding GSL to GNN baselines and removing GSL in SOTA GSL-based models. The results indicate that GSL does not consistently improve GNN performance in most cases.

- Under our proposed framework, we report effective GSL designs include pretrained GSL bases, parameter separation, and early fusion.

## 2 PRELIMINARY

**Graphs.** Suppose we have an undirected graph $\mathcal{G} = \{\mathcal{V}, \mathcal{E}\}$ with node set $\mathcal{V}$ and edge set $\mathcal{E}$. Let $\boldsymbol{Y} \in \mathbb{R}^{N \times C}$ denote the node labels and $\boldsymbol{X} \in \mathbb{R}^{N \times M}$ represent the node features, where $N$ is the number of nodes, $C$ is the number of classes, and $M$ is the number of features. The graph structure is represented by an adjacency matrix $\boldsymbol{A}$, where $\boldsymbol{A}_{u,v} = \boldsymbol{A}_{v,u} = 1$ indicates the existence of an edge $e_{uv}, e_{vu} \in \mathcal{E}$ between nodes $u$ and $v$. The normalized adjacency matrix is given by $\hat{\boldsymbol{A}} = \tilde{\boldsymbol{D}}^{-\frac{1}{2}} \tilde{\boldsymbol{A}} \tilde{\boldsymbol{D}}^{-\frac{1}{2}}$, where $\tilde{\boldsymbol{D}} = \boldsymbol{D} + \boldsymbol{I_n}$ and $\tilde{\boldsymbol{A}} = \boldsymbol{A} + \boldsymbol{I_n}$ represent the degree matrix and adjacency matrix with added self-loops. The neighbors of node $u$ is denoted as $\mathcal{N}_u = \{v | e_{uv} \in \mathcal{E}\}$. Graph Structure Learning (GSL) generates a new graph topology $\boldsymbol{A}'$, where the new neighbors of node $u$ are denoted as $\mathcal{N}'_u$. **Graph-aware models** $\mathcal{M}^{\mathcal{G}}$, such as Graph Convolutional Networks (GCN) (Kipf & Welling, 2016), are powerful in extracting structural information in graphs by message aggregation or graph filters (Luan et al., 2022b). In contrast, **graph-agnostic models** $\mathcal{M}^{\neg \mathcal{G}}$, such as Multilayer Perceptrons (MLP), only use $\boldsymbol{X}$ without considering $\mathcal{G}$. For example, the updating process of node embeddings in GCN and MLP can be represented as:

$$\text{GCN}: \ \boldsymbol{H}^l = \sigma(\hat{\boldsymbol{A}} \boldsymbol{H}^{l-1} \boldsymbol{W}^{l-1}), \ \text{MLP}: \ \boldsymbol{H}^l = \sigma(\boldsymbol{H}^{l-1} \boldsymbol{W}^{l-1}) \tag{1}$$

where $\boldsymbol{H}^l$ and $\boldsymbol{W}^l$ are the node embeddings and weight matrix at the $l$-th layer, respectively, and $\sigma(\cdot)$ is an activation function.

**Graph Homophily.** The concept of homophily originates from social network analysis and is defined as the tendency of individuals to connect with others who have similar characteristics (Khanam et al., 2023). A higher level of graph homophily makes the topological information of each node more informative, thereby improving the performance of graph-aware models $\mathcal{M}^{\mathcal{G}}$ (Luan et al., 2024b; 2022a; Zheng et al., 2024a). Commonly used homophily metrics include edge homophily (Zhu et al., 2020a; Abu-El-Haija et al., 2019) and node homophily (Pei et al., 2020):

$$h_{\text{edge}}(\mathcal{G}, \boldsymbol{Y}) = \frac{\left| \{e_{uv} | e_{uv} \in \mathcal{E}, Y_u = Y_v\} \right|}{|\mathcal{E}|}, \ h_{\text{node}}(\mathcal{G}, \boldsymbol{Y}) = \frac{1}{|\mathcal{V}|} \sum_{v \in \mathcal{V}} \frac{\left| \{u | u \in \mathcal{N}_v, Y_u = Y_v\} \right|}{\left| \mathcal{N}_v \right|} \tag{2}$$

**Contextual Stochastic Block Models with Homophily (CSBM-H).** To study the behavior of GNNs, CSBM-H (Luan et al., 2024b; Ma et al., 2021) have been proposed to create synthetic graphs with a controlled homophily degree. Specifically, in CSBM-H, for a node $u$ with label $y$, its features $\boldsymbol{X_u} \in \mathbb{R}^M$ are sampled from a class-wised Gaussian distribution $\boldsymbol{X}_u \sim \boldsymbol{N}_{Y_u}(\boldsymbol{\mu}_{Y_u}, \boldsymbol{\Sigma}_{Y_u})$ with $\boldsymbol{\mu}_{Y_u} \in \mathbb{R}^F$ and $\boldsymbol{\Sigma}_{Y_u} \in \mathbb{R}^{F \times F}$, where each dimension of $\boldsymbol{X}_u$ is independent from each other, i.e., $\boldsymbol{\Sigma}_{Y_u} = \text{diag}(\mathbb{R}^n_{\geq 0})$. Then, to generate graph structure $\mathcal{G}$ with given homophily degree $h$ with the range of $[0, 1]$, the node $u$ has the probability $h$ to connect intra-class nodes and the probability $\frac{1-h}{C-1}$ to connect inter-class nodes. After applying neighbor sampling, both of the node homophily $h_{node}$ and edge homophily $h_{edge}$ in $\mathcal{G}$ are approximately equal to $h$.

**Mutual Information.** Mutual Information quantifies the amount of information obtained about one random variable given another variable (mut, 2024). The mutual information between variable $X$ and $Y$ can be expressed as:

$$I(\boldsymbol{X}; \boldsymbol{Y}) = \sum_{y \in \mathcal{Y}} \sum_{x \in \mathcal{X}} p(x, y) \ \log \frac{p(x, y)}{p(x)p(y)} \tag{3}$$

where $p(x, y)$ is joint probability, and $p(x)$ and $p(y)$ are marginal probability.

Mutual information could be used to analyze the quality of input features by measuring how much information the inputs $\boldsymbol{X}$ retain about the outputs $\boldsymbol{Y}$. However, in graphs under the task of node classification, the mutual information between a discrete variable $\boldsymbol{Y}$ and a continuous variable $\boldsymbol{X}$ cannot be directly measured by Eq. (3). Therefore, in this paper, we measure the mutual information $I(\boldsymbol{X}; \boldsymbol{Y})$ based on entropy estimation from k-nearest neighbors distances following (Kraskov et al., 2004; Ross, 2014; Kozachenko & Leonenko, 1987).

## 3 GRAPH STRUCTURE LEARNING

This section introduces our proposed Graph Structure Learning (GSL) framework. Previous surveys (Zhu et al., 2021a;b; Qiao et al., 2018) only focus on new structure construction, constituting one

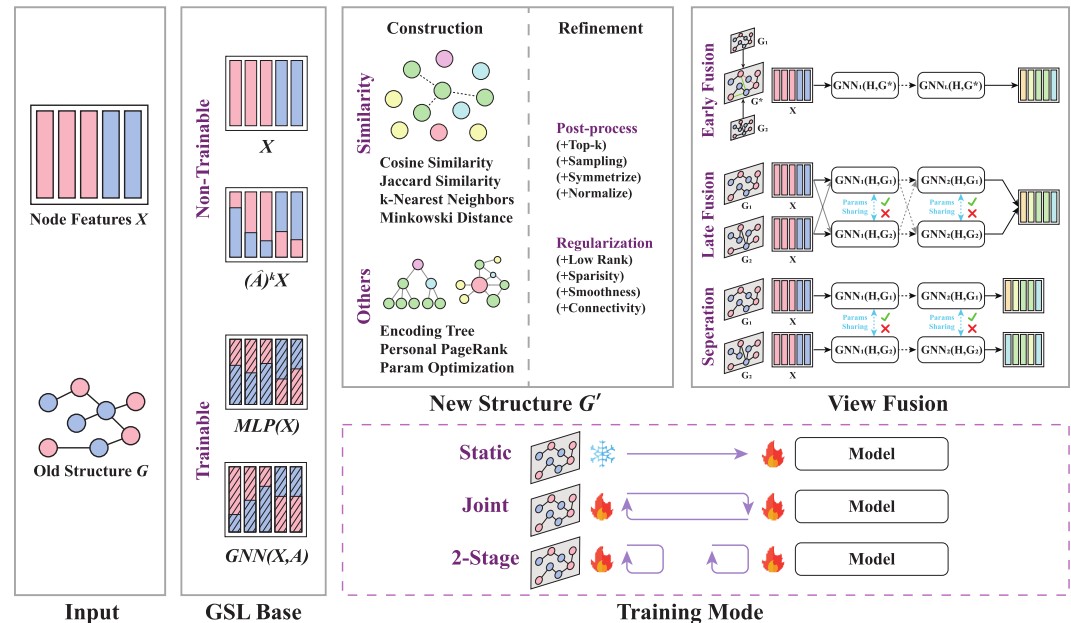

Figure 2: A framework of GSL.

step in GNNs learning. To provide a comprehensive understanding of GSL with GNNs, as shown in Figure 2, our framework includes three steps: GSL bases generation, new structure construction, and view fusion. Then we describe the pipeline of the GSL framework: First, GSL bases $B$ is constructed based on node features $X$ (and input graphs $\mathcal{G}$). Then, new graph structures $\mathcal{G}'$ are constructed with the GSL bases. Last, the information from new graph $\mathcal{G}'$ (multiple views if possible) and original graph $\mathcal{G}$ are combined with different view fusion strategies for the training of GNNs. Please refer to Appendix A for a more detailed explanation of the representative GSL methods within our proposed GSL framework.

### 3.1 GSL BASES

The GSL bases $B$ is defined as the processed node embeddings used before constructing new structures. The quality of the GSL bases plays a crucial role in determining the performance of GNNs with GSL. For node classification tasks, an effective GSL bases $B$ should exhibit consistency among intra-class nodes, as shown in the left part of Figure 1, is expected to be consistent among intra-class nodes. From the embedding training perspective, the construction of $B$ can be categorized as either non-parametric approaches (Franceschi et al., 2020; Pei et al., 2020; Zou et al., 2023), which generate static $B$, or parametric approaches (Jin et al., 2020; Chen et al., 2020; Yu et al., 2020), which updates $B$ dynamically during training. From the perspective of information usage, the construction of $B$ can be categorized into graph-agnostic approaches (Franceschi et al., 2020; Jin et al., 2020; Zou et al., 2023) or graph-aware approaches (Pei et al., 2020; Yu et al., 2020; Wang et al., 2021). Combining these two perspectives, in Figure 2, we show the diagrams of four types of of $B$ construction: $B = X$, $B = (A)^k X$, $B = \text{MLP}(X)$, and $B = \text{GNN}(X, A)$.

### 3.2 NEW STRUCTURE CONSTRUCTION

The construction of the new structure $\mathcal{G}'$, based on $B$, is a key element of GSL. From the perspective of relation extraction, methods for constructing $\mathcal{G}'$ can be categorized into similarity-based (Pei et al., 2020; Jiang et al., 2019; Li et al., 2023a), structure-based (Zhao et al., 2020; Liu et al., 2022a; Zou et al., 2023), and parametric optimization-based (Jin et al., 2020; Liu et al., 2022b; Li et al., 2022b) approaches. Similarity-based methods are the most prevalent, and the choice of similarity measurement, such as k-Nearest Neighbors (Franceschi et al., 2020), cosine similarity (Chen et al., 2020), or Minkowski distance (Liu et al., 2022b), plays a critical role in the quality of the reconstructed graphs. However, the initial $\mathcal{G}'$ produced by these methods often results in a coarse graph structure, which may not be optimal for GNN training. Thus, further refinements are often

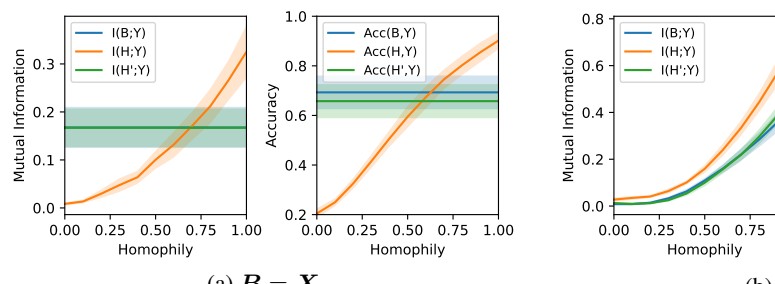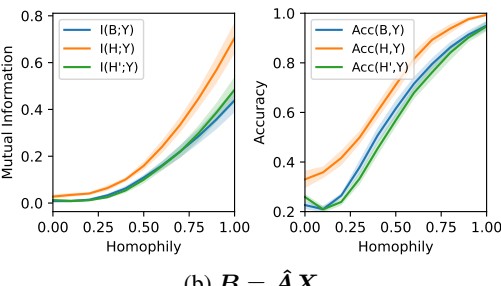

(a) $\boldsymbol{B} = \boldsymbol{X}$  (b) $\boldsymbol{B} = \hat{\boldsymbol{A}}\boldsymbol{X}$

Figure 3: Mutual information and accuracy of node classification on GSL bases $\boldsymbol{B}$, convoluted bases of old graphs $\boldsymbol{H} = \hat{\boldsymbol{A}}\boldsymbol{B}$, convoluted bases of new graphs $\boldsymbol{H}' = \hat{\boldsymbol{A}}'\boldsymbol{B}$, across varying homophily degrees. The rewriting bases $\boldsymbol{B}$ is set to node features $\boldsymbol{B} = \boldsymbol{X}$ (left) or aggregated features $\boldsymbol{B} = \hat{\boldsymbol{A}}\boldsymbol{X}$ (right).

necessary, such as sampling (Zhao et al., 2020; Li et al., 2022a; Liu et al., 2022b), symmetrization (Yu et al., 2020; Fatemi et al., 2021; Liu et al., 2022b), normalization (Zhao et al., 2020; Liu et al., 2022b; Jiang et al., 2019), or applying graph regularization (Jin et al., 2020; Jiang et al., 2019; Li et al., 2022b).

### 3.3 View Fusion

In cases where the methods (Fatemi et al., 2021; Zou et al., 2023; Jiang et al., 2019) already implicitly fuse the information from $\mathcal{G}$ into $\mathcal{G}'$, further view fusion is unnecessary. However, for other approaches, the fusion of information from the original graph structure $\mathcal{G}$ and the reconstructed structure $\mathcal{G}'$ is crucial. Based on the fusion stage, methods can be classified as early fusion (Li et al., 2022a; Lao et al., 2022; Liu et al., 2022a), late fusion (Wang et al., 2021; Liu et al., 2022b; Zheng et al., 2024b), or separation (Liu et al., 2022b). Early fusion, often seen as "graph editing", modifies $\mathcal{G}$ by adding or removing edges with $\mathcal{G}'$ before training. Late fusion keeps both views as input, fusing node embeddings either at each layer or in the final layer. Separation methods, typically paired with contrastive learning, maintain multiple views without embedding fusion during GNN training. Additionally, view fusion methods can be further distinguished by whether they involve parameter sharing across layers during training.

### 3.4 Training Mode

In addition to the previous three steps, the training mode of $\mathcal{G}'$ plays a crucial role in GSL and can be categorized into static, joint, and 2-stage approaches. Most methods (Jin et al., 2020; Li et al., 2022b; Yan et al., 2022) use joint training where $\mathcal{G}'$ and model parameters are optimized simultaneously. In contrast, some methods (Wang et al., 2021; Liu et al., 2022a; Franceschi et al., 2020) follow a 2-stage mode, iteratively updating $\mathcal{G}'$ and model parameters. While dynamic updates offer greater flexibility for learning complex structures through parameter optimization, they also significantly increase computational complexity, especially during the bases and graph construction steps. To address this, other methods (Zheng et al., 2024b; Suresh et al., 2021; Li et al., 2023a) opt for a static $\mathcal{G}'$ during training. Although this fixed structure may limit performance, it avoids the time-consuming process of frequent graph updates.

## 4 Effectiveness of Graph Structure Learning

This section analyzes the impact of GSL on GNN performance with empirical observations in Section 4.1 and theoretical analysis in Section 4.2. Then, the time complexity of GSL is analyzed in Section 4.3.

### 4.1 Empirical Observations

**Setting** Based on CSBM-H, we generate synthetic graphs with 10 random seeds for each homophily degree $h \in \{0, 0.1, \dots, 1.0\}$ to mitigate randomness effects. Each graph $\mathcal{G}$ contains 1000

nodes, with each node characterized by 10 features, 5 balanced classes, and a degree sampled from the range $[2, 10]$. Then, for the GSL, we apply k-Nearest-Neighbors algorithm on GSL bases $\boldsymbol{B}$ with $k = 5$ to generate new graphs, *i.e.,* $\mathcal{G}' = \text{kNN}(\boldsymbol{B})$. To inspect the effectiveness of GSL, based on $\boldsymbol{B}$, $\mathcal{G}$, and $\mathcal{G}'$, we can get several forms of node representation: $\boldsymbol{B}$, $\boldsymbol{H} = \text{GCN}(\boldsymbol{B}, \mathcal{G})$, and $\boldsymbol{H}' = \text{GCN}(\boldsymbol{B}, \mathcal{G}')$, corresponding to the node representation of MLP, GCN, GCN+GSL respectively. We assess the performance of these methods on node classification tasks using mutual information, $\text{MI}(\cdot)$, as a non-parametric measure, and accuracy, $\text{Acc}(\cdot)$, as a parametric measure.

To consider both the graph-agnostic bases and graph-aware bases as discussed in Section 3.1, the GSL bases is selected as $\boldsymbol{B} = \boldsymbol{X}$ and $\boldsymbol{B} = \hat{\boldsymbol{A}}\boldsymbol{X}$ as shown in Figure 3a and Figure 3b respectively. For example, on the left part of Figure 3a, the performance of MLP, GCN, GCN+GSL is shown as mutual information $I(\boldsymbol{B}; \boldsymbol{Y})$, $I(\boldsymbol{H}; \boldsymbol{Y})$, and $I(\boldsymbol{H}'; \boldsymbol{Y})$ respectively. Based on these results, we make several observations as follows:

**Observation 1. Mutual information is an effective non-parametric measure of model performance.** As shown in Figure 3a and Figure 3b, the trend of mutual information $I(\cdot)$ (left) closely mirrors the model accuracy $\text{ACC}(\cdot)$ (right). Additionally, mutual information effectively distinguishes performance differences between methods. Since mutual information is non-parametric, it offers a flexible and reliable measure, making it suitable for theoretical analysis in the next section.

**Observation 2. MLP performs comparably to GCN+GSL under the same GSL bases.** In Figure 3a, the mutual information $I(\boldsymbol{B}; \boldsymbol{Y})$ and classification accuracy $\text{ACC}(\boldsymbol{B}, \boldsymbol{Y})$ are close to $I(\boldsymbol{H}'; \boldsymbol{Y})$ and $\text{ACC}(\boldsymbol{H}', \boldsymbol{Y})$, respectively, across both graph-agnostic and graph-aware GSL bases. This suggests that, contrary to the expectation that GSL might enhance performance, the results indicate that the model performance does not improve significantly after applying graph convolution on $\mathcal{G}'$, reinforcing the GSL controversy discussed in Figure 1.

**Observation 3. GCN+GSL sometimes outperforms GCN in heterophilous graphs.** As shown in Figure 3a, the $I(\boldsymbol{B}; \boldsymbol{Y})$ or $\text{ACC}(\boldsymbol{B}, \boldsymbol{Y})$ increases with a higher homophily degree, while $I(\boldsymbol{H}'; \boldsymbol{Y})$ or $\text{ACC}(\boldsymbol{H}', \boldsymbol{Y})$ remain stable across homophily degrees. In graphs with low homophily, the neighbors identified by GSL are more likely to share the same labels as the target node compared to the original graph neighbors, which causes GCN to underperform relative to GCN+GSL. However, this effect is observed only when $\boldsymbol{B} = \boldsymbol{X}$ (3a). When $\boldsymbol{B} = \hat{\boldsymbol{A}}\boldsymbol{X}$ (3b), the GSL bases does not exhibit consistency among intra-class nodes in low homophily settings, leading GCN+GSL to perform worse than GCN.

These observations highlight that even when GCN+GSL outperforms GCN, its performance remains close to MLP under the same GSL bases. Recent studies (Luo et al., 2024; Platonov et al., 2023) also indicate that under consistent hyperparameter tuning, GSL does not always consistently outperform classic GNN baselines. This leads us to reconsider the necessity of GSL. Thus, in addition to the empirical observations above, we proceed with a theoretical analysis of GSL's effectiveness in the following section.

## 4.2 Theoretical Analysis

To explain the above empirical observations, in this section, we first prove that the mutual information $I(Y; H)$ of label $Y$ and aggregated features $H$ serve as a non-parametric measurement of the performance of graph convolution. Following this, we compare the mutual information between the node labels $\boldsymbol{Y}$ and either the original GSL bases $\boldsymbol{B}$ or the aggregated GSL bases $\boldsymbol{H}'$ (on GSL graph $\mathcal{G}'$), to highlight the impact of GSL on model performance.

**Theorem 1.** Given a graph $\mathcal{G} = \{\mathcal{V}, \mathcal{E}\}$ with node labels $\boldsymbol{Y}$ and node features $\boldsymbol{X}$, the accuracy of graph convolution on node classification is upper bounded by the mutual information of node label $Y$ and aggregated node features $H$:

$$P_A \leq \frac{I(Y; H) + \log 2}{\log(C)} \tag{4}$$

**Proposition 1.** Consider a graph $\mathcal{G} = \{\mathcal{V}, \mathcal{E}\}$ characterized by node labels $Y$ and $n$-dimensional node bases $\mathbf{B} = \{B_1, B_2, \ldots, B_n\}$ with $C$ classes. Each base $B_i$ is independent and follows a class-dependent Gaussian distribution, *i.e.,* $B_i \sim \mathcal{N}(\mu_Y, \sigma_Y)$. A new graph $\mathcal{G}' = \{\mathcal{V}, \mathcal{E}'\}$ is generated using a non-parametric method based on the bases $\mathbf{B}$. For the aggregated bases $\boldsymbol{B}'$ on $\mathcal{G}'$, we have $\inf I(Y; \boldsymbol{B}') \leq \inf I(Y; \boldsymbol{B})$.

where the proofs are shown in Appendix B.

Theorem 1 shows that the mutual information $I(Y;H)$ provides an upper bound on the accuracy of graph convolution for node classification, which justifies why mutual information serves as an effective measure of model performance, as demonstrated in Observation 1.

Based on the conclusion of mutual information in Theorem 1, we analyze the effectiveness of GSL. Proposition 1 shows that the graph convolution on new graphs generated by GSL does not increase the lower bound of mutual information. This explains why MLP performs similarly to, or slightly better than, GCN+GSL in Observation 2 and the GSL controversy in Figure 1[1]

To further explain Observation 3 in Section 4.1, we refer again to Proposition 1. In conjunction with previous studies on graph homophily (Pei et al., 2020; Luan et al., 2022a; Zheng et al., 2024a), we know that the performance of GCN could be inferior to MLP on heterophilous graphs. Since GCN+GSL is upper bounded by the MLP on the same GSL bases, when MLP outperforms GCN, GCN+GSL may also outperform GCN, as seen in Figure 3a. However, even when GCN+GSL surpasses GCN in some cases, it still lags behind MLP, a much simpler model, on the same GSL bases. Therefore, we hypothesize that previous GSL improvements stem from the construction of the GSL bases or the introduction of additional model parameters. A fair comparison of GSL with other GNNs or MLP baselines should be conducted using the same GSL bases, as demonstrated in our experiments.

### 4.3 COMPLEXITY ANALYSIS

After investigating the difference in the performance of GCN+GSL and GCN, we then analyze the time complexity of some representative methods of GSL, such as IDGL (Chen et al., 2020), GRCN (Yu et al., 2020), GAug (Zhao et al., 2020), and HOG-GCN (Wang et al., 2022), as shown in Table 3. Assume the dimension of node representation is $F$ for all the layers, the additional time complexity introduced by GSL generally includes: 1. Construction of GSL bases: $O(|\mathcal{E}|F + |\mathcal{V}|F^2)$ for graph-aware bases or $O(|\mathcal{V}|F^2)$ for graph-agnostic bases, 2. Graph construction: $O(|\mathcal{V}|^2 F)$, 3. Graph refinement: $O(|\mathcal{V}|^2)$, and 4: View Fusion $O(|\mathcal{V}|^2)$. Apart from the complexity of the new graph construction in GSL, during the graph convolution, compared with GNNs without using GSL, the additional complexity is further introduced by single view GSL $O(|\mathcal{E}'|F)$ or multiple view GSL $O((N_{\mathcal{G}}-1)(|\mathcal{E}|F + |\mathcal{V}|F^2))$, where $|\mathcal{E}'|$ is the additional edges introduced in GSL and $N_{\mathcal{G}}$ is the number of views in GSL. Consider the fact that $|\mathcal{V}|^2 \gg |\mathcal{E}|$, we have the total additional complexity of GSL by summing up all these terms: $O(|\mathcal{V}|^2 F + |\mathcal{V}|F^2)$. Compared with the complexity in normal GCN $O(|\mathcal{E}|F + |\mathcal{V}|F^2)$ (Blakely et al., 2021), this additional complexity $O((|\mathcal{V}|^2 - |\mathcal{E}|)F)$ adds tremendous training time and grows exponentially with the number of nodes in graphs, which is shown in our experiments.

## 5 EXPERIMENTS

In this section, we evaluate the effectiveness of GSL by comparing the performance of baseline GNNs and GNNs augmented with GSL (GNN+GSL), as well as the performance of GSL-based state-of-the-art (SOTA) methods against their non-GSL counterparts (SOTA vs. SOTA-GSL) on node classification tasks. The results of these comparisons are presented in Section 5.1. Additionally, we analyze the quality of the newly constructed graphs generated by GSL in Section 5.2 and investigate how different components of GSL impact model performance in Section 5.3.

**Settings.** Our experiments include several baseline GNNs: GCN (Kipf & Welling, 2016), SGC (Wu et al., 2019a), GraphSage (Hamilton et al., 2017), and GAT (Velickovic et al., 2017), and GSL-based SOTA models: GAug (Zhao et al., 2020), GEN (Wang et al., 2021), GRCN (Yu et al., 2020), IDGL (Chen et al., 2020), NodeFormer (Wu et al., 2023), GloGNN (Li et al., 2022b), WRGAT (Suresh et al., 2021), and WRGCN (Suresh et al., 2021). The datasets used in our experiments include heterophilous graphs: Squirrel, Chameleon, Actor, Texas, Cornell, and Wisconsin (Pei et al., 2020; Rozemberczki et al., 2021), homophilous graphs: Cora, PubMed, and Citeseer (Yang et al., 2016),

---

[1]Admittedly, this theoretical analysis cannot be extended to optimization-based GSL due to the complexity of non-linear optimization problems. As such, the unnecessity of GSL in these methods is confirmed through our experiments.

and Minesweeper, Roman-empire, Amazon-ratings, Tolokers, and Questions (Platonov et al., 2023). We show more dataset details in Appendix C. The model performance is measured by accuracy for multi-class datasets or AUC-ROC for binary-class datasets on node classification tasks. For the data splits, we use $50\%/25\%/25\%$ in train/validation/test sets for GNN+GSL and follow the default splits in OpenGSL (Zhiyao et al., 2024) for each SOTA or SOTA-GSL method. Please refer to Appendix D for more implementation details.

## 5.1 PERFORMANCE COMPARISON

**GNN+GSL.** We investigate the impact of GSL by comparing the performance of GNN and GNN+GSL. As GSL introduces significant variations in three key aspects, outlined in Table 3, we aim to comprehensively evaluate all possible GSL configurations through a combination of various GSL components, which include 1) GSL bases: original features $X$, aggregated features $\hat{A}X$, MLP-pretrained features MLP($X$), GCN-pretrained features GCN($X$, $A$), GCL(Graph Contrastive Learning)-pretrained features GCL($X$, $A$); 2) GSL Graph Construction: Graphs constructed via cosine similarity at the graph level (cos-graph), node level (cos-node), and k-nearest neighbors (kNN); and 3) View Fusion: early fusion $\{\mathcal{G}'\}$, late fusion $\{\mathcal{G}, \mathcal{G}'\}$ with parameter sharing $\theta_1 = \theta_2$ or not $\theta_1 \neq \theta_2$. To ensure a fair comparison of the performance between GNN+GSL, GNN, and MLP, we consider 5 GSL bases as input choices and train all models on each GSL bases. Detailed explanations of these GSL modules can be found in Appendix D.1.

Table 1: Performance of GNNs with GNN+GSL.

| Model | Construct | Fusion | Param Sharing | Mines. | Roman. | Amazon. | Tolokers | Questions | Squirrel | Chameleon | Actor | Texas | Cornell | Wisconsin | Cora | CiteSeer | PubMed | Rank |
|---|---|---|---|---|---|---|---|---|---|---|---|---|---|---|---|---|---|---|
| MLP | None | - | - | 79.55±1.23 | 65.45±0.99 | 46.65±0.83 | 75.94±1.38 | 74.92±1.39 | 39.29±2.22 | 43.57±4.18 | 35.40±1.38 | 80.46±6.44 | 73.78±7.34 | 85.88±7.78 | 87.97±1.80 | 76.68±2.10 | 87.39±2.18 | 3.93 |
| GCN | None | - | - | 90.07±5.79 | 81.46±1.25 | 50.89±1.16 | 84.61±0.99 | 77.68±1.10 | 41.26±2.47 | 43.24±3.86 | 34.34±1.17 | 73.08±8.68 | 67.03±10.54 | 78.24±8.32 | 87.97±1.51 | 76.75±2.30 | 89.47±0.64 | 1.36 |
| GCN | cos-graph | $\{\mathcal{G}'\}$ | - | 77.91±5.25 | 67.40±1.02 | 46.72±1.51 | 76.11±1.52 | 72.56±1.14 | 38.15±2.45 | 39.87±4.87 | 33.47±1.61 | 63.06±9.85 | 65.68±7.76 | 72.75±5.70 | 85.21±1.39 | 75.52±1.14 | 89.03±0.42 | 6.71 |
| GCN | cos-graph | $\{\mathcal{G},\mathcal{G}'\}$ | $\theta_1=\theta_2$ | 52.53±6.45 | 62.57±0.81 | 41.29±1.61 | 74.22±1.79 | 69.63±1.52 | 37.62±1.74 | 39.78±4.00 | 32.74±0.92 | 57.88±8.75 | 66.49±9.12 | 73.14±5.92 | 64.68±1.61 | 67.32±1.89 | 86.43±0.76 | 9.32 |
| GCN | cos-graph | $\{\mathcal{G},\mathcal{G}'\}$ | $\theta_1\neq\theta_2$ | 88.70±0.86 | 69.90±2.38 | 47.35±0.83 | 82.85±0.95 | 75.29±1.38 | 38.84±2.87 | 40.30±4.31 | 33.73±1.49 | 65.47±8.48 | 62.97±10.89 | 75.29±6.54 | 85.51±1.87 | 75.23±1.14 | 88.74±0.59 | 4.79 |
| GCN | cos-node | $\{\mathcal{G}'\}$ | - | 85.57±6.63 | 68.24±2.49 | 47.56±1.32 | 77.26±1.44 | 74.16±1.80 | 38.14±2.40 | 40.16±3.13 | 34.04±1.66 | 61.13±8.19 | 61.08±8.16 | 71.18±6.98 | 86.06±1.95 | 75.76±1.39 | 88.92±0.50 | 5.93 |
| GCN | cos-node | $\{\mathcal{G},\mathcal{G}'\}$ | $\theta_1=\theta_2$ | 52.53±6.45 | 62.57±0.81 | 41.29±1.61 | 74.22±1.79 | 69.63±1.52 | 37.62±1.74 | 39.78±4.00 | 32.74±0.92 | 57.88±8.75 | 66.49±9.12 | 73.14±5.92 | 64.68±1.61 | 67.32±1.89 | 86.43±0.76 | 9.36 |
| GCN | cos-node | $\{\mathcal{G},\mathcal{G}'\}$ | $\theta_1\neq\theta_2$ | 89.17±0.68 | 72.63±1.45 | 48.31±0.96 | 82.91±0.97 | 75.56±1.05 | 38.41±2.32 | 39.94±4.49 | 34.10±1.53 | 64.68±8.85 | 63.24±9.47 | 73.92±7.51 | 85.69±1.73 | 75.49±1.42 | 88.72±0.71 | 4.29 |
| GCN | kNN | $\{\mathcal{G}'\}$ | - | 82.89±6.66 | 68.44±0.83 | 47.13±1.00 | 78.92±1.79 | 73.90±1.73 | 38.15±2.02 | 40.22±3.82 | 33.94±1.24 | 63.03±8.53 | 61.35±9.28 | 72.16±7.41 | 86.08±1.62 | 75.56±1.42 | 88.59±0.58 | 5.93 |
| GCN | kNN | $\{\mathcal{G},\mathcal{G}'\}$ | $\theta_1=\theta_2$ | 52.53±6.45 | 62.57±0.81 | 41.29±1.61 | 74.22±1.79 | 69.63±1.52 | 37.62±1.74 | 39.78±4.00 | 32.74±0.92 | 57.88±8.75 | 66.49±9.12 | 73.14±5.92 | 64.68±1.61 | 67.32±1.89 | 86.43±0.76 | 9.39 |
| GCN | kNN | $\{\mathcal{G},\mathcal{G}'\}$ | $\theta_1\neq\theta_2$ | 88.96±0.73 | 72.44±1.61 | 47.06±0.83 | 83.10±0.80 | 75.61±1.19 | 37.63±1.93 | 40.18±4.76 | 33.84±1.94 | 63.87±9.68 | 62.16±9.77 | 75.49±7.29 | 85.82±1.55 | 75.50±1.30 | 88.54±0.55 | 5.00 |
| MLP | None | - | - | 79.55±1.23 | 65.45±0.99 | 46.65±0.83 | 75.94±1.38 | 74.92±1.39 | 39.29±2.22 | 43.57±4.18 | 35.40±1.38 | 80.46±6.44 | 73.78±7.34 | 85.88±7.78 | 87.97±1.80 | 76.68±2.10 | 87.39±2.18 | 3.71 |
| SGC | None | - | - | 83.45±4.47 | 78.04±0.69 | 51.38±0.68 | 84.88±1.13 | 77.39±1.23 | 41.18±2.73 | 42.35±4.10 | 34.05±1.41 | 73.63±6.94 | 70.27±9.91 | 80.59±5.13 | 88.10±1.89 | 77.52±2.20 | 89.39±0.62 | 1.57 |
| SGC | cos-graph | $\{\mathcal{G}'\}$ | - | 73.76±4.46 | 67.17±0.81 | 47.15±0.88 | 76.28±1.63 | 73.93±2.66 | 38.66±2.53 | 40.07±4.39 | 33.87±1.45 | 71.19±7.38 | 67.57±9.19 | 77.65±6.08 | 86.95±2.01 | 76.12±1.29 | 89.10±0.43 | 5.79 |
| SGC | cos-graph | $\{\mathcal{G},\mathcal{G}'\}$ | $\theta_1=\theta_2$ | 52.53±4.89 | 62.97±0.78 | 42.42±1.57 | 74.29±1.79 | 70.56±1.27 | 37.56±2.25 | 39.33±3.60 | 32.85±0.90 | 57.60±7.53 | 66.49±10.37 | 71.57±4.46 | 64.82±2.11 | 67.55±1.80 | 86.56±0.72 | 9.64 |
| SGC | cos-graph | $\{\mathcal{G},\mathcal{G}'\}$ | $\theta_1\neq\theta_2$ | 79.70±1.21 | 62.02±2.06 | 47.24±0.93 | 83.22±1.52 | 77.19±0.99 | 38.32±1.80 | 40.85±4.61 | 33.51±1.50 | 70.34±7.31 | 64.86±9.01 | 75.29±6.82 | 87.47±1.70 | 75.70±1.28 | 88.65±0.49 | 6.14 |
| SGC | cos-node | $\{\mathcal{G}'\}$ | - | 79.03±3.76 | 67.84±1.87 | 47.93±0.94 | 78.09±1.84 | 75.46±1.43 | 38.61±2.20 | 40.50±4.10 | 34.03±1.27 | 70.08±6.84 | 68.11±9.23 | 77.45±4.63 | 87.47±1.86 | 76.36±1.27 | 89.37±0.41 | 4.54 |
| SGC | cos-node | $\{\mathcal{G},\mathcal{G}'\}$ | $\theta_1=\theta_2$ | 52.53±4.89 | 62.97±0.78 | 42.42±1.57 | 74.29±1.79 | 70.56±1.27 | 37.56±2.25 | 39.33±3.60 | 32.85±0.90 | 57.60±7.53 | 66.49±10.37 | 71.57±4.46 | 64.82±2.11 | 67.55±1.80 | 86.58±0.72 | 9.57 |
| SGC | cos-node | $\{\mathcal{G},\mathcal{G}'\}$ | $\theta_1\neq\theta_2$ | 80.12±1.36 | 66.90±1.66 | 48.04±0.97 | 83.53±1.43 | 77.11±1.09 | 38.52±2.29 | 40.20±4.66 | 34.20±1.79 | 68.47±8.11 | 64.59±9.74 | 75.29±6.05 | 87.54±1.63 | 75.88±1.26 | 88.68±0.43 | 5.11 |
| SGC | kNN | $\{\mathcal{G}'\}$ | - | 75.53±4.98 | 67.94±0.70 | 47.68±0.84 | 79.45±2.06 | 74.22±2.47 | 37.32±2.10 | 39.92±3.91 | 34.05±1.55 | 72.81±6.15 | 70.00±7.98 | 77.84±6.02 | 87.82±1.77 | 76.54±1.44 | 89.19±0.42 | 4.64 |
| SGC | kNN | $\{\mathcal{G},\mathcal{G}'\}$ | $\theta_1=\theta_2$ | 52.53±4.89 | 62.97±0.78 | 42.42±1.57 | 74.29±1.79 | 70.56±1.27 | 37.56±2.25 | 39.33±3.60 | 32.85±0.90 | 57.60±7.53 | 66.49±10.37 | 71.57±4.46 | 64.82±2.11 | 67.55±1.80 | 86.58±0.72 | 9.50 |
| SGC | kNN | $\{\mathcal{G},\mathcal{G}'\}$ | $\theta_1\neq\theta_2$ | 80.78±1.08 | 64.59±1.93 | 47.48±0.99 | 83.17±1.43 | 76.80±1.09 | 36.53±2.06 | 40.17±4.24 | 34.23±1.72 | 69.26±6.77 | 65.95±8.87 | 76.08±5.92 | 87.38±1.49 | 76.02±1.22 | 88.77±0.45 | 5.79 |
| MLP | None | - | - | 79.55±1.23 | 65.45±0.99 | 46.65±0.83 | 75.94±1.38 | 74.92±1.39 | 39.29±2.22 | 43.57±4.18 | 35.40±1.38 | 80.46±6.44 | 73.78±7.34 | 85.88±7.78 | 87.97±1.80 | 76.68±2.10 | 87.39±2.18 | 4.14 |
| SAGE | None | - | - | 90.66±0.88 | 85.02±0.97 | 52.93±0.83 | 83.31±1.12 | 75.95±1.41 | 40.43±2.64 | 42.95±5.37 | 34.83±1.20 | 80.17±6.90 | 75.68±7.52 | 86.27±6.67 | 88.13±1.77 | 76.65±2.00 | 89.18±0.65 | 1.71 |
| SAGE | cos-graph | $\{\mathcal{G}'\}$ | - | 80.39±4.66 | 70.13±1.05 | 47.55±1.17 | 76.77±1.28 | 72.86±1.18 | 39.03±2.69 | 40.84±5.42 | 34.75±1.39 | 70.91±8.58 | 70.00±7.56 | 78.24±6.87 | 83.64±2.03 | 75.53±1.36 | 89.18±0.55 | 6.07 |
| SAGE | cos-graph | $\{\mathcal{G},\mathcal{G}'\}$ | $\theta_1=\theta_2$ | 53.02±6.49 | 59.98±1.73 | 39.99±2.29 | 71.57±2.28 | 66.01±3.58 | 35.05±2.41 | 38.49±3.68 | 31.32±1.04 | 60.30±7.05 | 67.57±4.59 | 76.47±5.92 | 64.58±1.74 | 67.77±1.31 | 85.53±0.51 | 9.93 |
| SAGE | cos-graph | $\{\mathcal{G},\mathcal{G}'\}$ | $\theta_1\neq\theta_2$ | 90.67±0.66 | 79.02±1.21 | 52.10±0.84 | 82.17±0.89 | 75.38±0.96 | 39.36±2.14 | 40.64±6.06 | 35.14±1.08 | 76.08±6.30 | 70.27±6.62 | 79.41±5.71 | 83.60±1.78 | 74.39±1.35 | 88.88±0.50 | 3.86 |
| SAGE | cos-node | $\{\mathcal{G}'\}$ | - | 85.26±4.64 | 71.25±1.76 | 48.96±0.87 | 78.39±1.75 | 73.01±1.11 | 38.68±2.75 | 40.81±4.51 | 35.10±1.26 | 71.47±9.47 | 68.11±7.87 | 75.49±6.32 | 84.88±1.90 | 75.58±1.04 | 89.17±0.35 | 5.64 |
| SAGE | cos-node | $\{\mathcal{G},\mathcal{G}'\}$ | $\theta_1=\theta_2$ | 53.02±6.49 | 59.98±1.73 | 39.99±2.29 | 71.59±2.28 | 66.01±3.58 | 35.05±2.41 | 38.49±3.68 | 31.32±1.04 | 60.30±7.05 | 67.57±4.59 | 76.47±5.92 | 64.58±1.74 | 67.77±1.31 | 85.53±0.51 | 9.79 |
| SAGE | cos-node | $\{\mathcal{G},\mathcal{G}'\}$ | $\theta_1\neq\theta_2$ | 90.64±0.65 | 78.60±0.98 | 52.08±0.90 | 82.02±0.88 | 75.31±1.12 | 39.18±2.54 | 40.86±6.17 | 35.18±1.24 | 74.71±5.65 | 69.73±7.43 | 80.00±5.68 | 83.96±1.65 | 74.63±1.26 | 88.93±0.64 | 3.93 |
| SAGE | kNN | $\{\mathcal{G}'\}$ | - | 82.86±3.14 | 70.74±0.80 | 48.40±1.01 | 78.12±2.17 | 72.70±1.15 | 38.93±2.84 | 39.68±5.40 | 35.09±1.14 | 70.91±9.05 | 68.92±6.88 | 75.69±6.73 | 84.40±1.75 | 75.68±1.43 | 88.86±0.44 | 6.50 |
| SAGE | kNN | $\{\mathcal{G},\mathcal{G}'\}$ | $\theta_1=\theta_2$ | 53.02±6.49 | 59.98±1.73 | 39.99±2.29 | 71.59±2.28 | 66.01±3.58 | 35.05±2.41 | 38.49±3.68 | 31.32±1.04 | 60.30±7.05 | 67.57±4.59 | 76.47±5.92 | 64.58±1.74 | 67.77±1.31 | 85.53±0.51 | 9.86 |
| SAGE | kNN | $\{\mathcal{G},\mathcal{G}'\}$ | $\theta_1\neq\theta_2$ | 90.61±0.63 | 79.16±1.15 | 51.56±1.07 | 81.66±0.87 | 75.22±0.97 | 39.20±2.39 | 40.44±5.82 | 35.13±1.38 | 74.17±6.31 | 70.54±7.32 | 79.61±6.61 | 84.05±1.63 | 74.59±1.25 | 88.67±0.55 | 4.57 |
| MLP | None | - | - | 79.55±1.23 | 65.45±0.99 | 46.65±0.83 | 75.94±1.38 | 74.92±1.39 | 39.29±2.22 | 43.57±4.18 | 35.40±1.38 | 80.46±6.44 | 73.78±7.34 | 85.88±7.78 | 87.97±1.80 | 76.68±2.10 | 87.39±2.18 | 3.86 |
| GAT | None | - | - | 90.41±1.34 | 84.51±0.84 | 52.00±2.84 | 84.37±0.96 | 77.78±1.27 | 41.67±2.51 | 43.83±3.66 | 33.73±1.77 | 75.28±8.12 | 65.41±12.14 | 77.84±7.41 | 88.02±1.92 | 76.77±2.02 | 89.21±0.67 | 2.04 |
| GAT | cos-graph | $\{\mathcal{G}'\}$ | - | 80.78±8.24 | 67.68±1.25 | 45.79±1.10 | 74.84±1.84 | 72.34±1.49 | 38.74±2.54 | 40.21±3.53 | 33.37±1.10 | 62.73±9.06 | 67.57±7.03 | 77.06±7.29 | 86.03±1.85 | 75.46±1.49 | 89.63±0.59 | 6.29 |
| GAT | cos-graph | $\{\mathcal{G},\mathcal{G}'\}$ | $\theta_1=\theta_2$ | 53.16±7.93 | 63.67±1.08 | 44.83±2.04 | 73.46±1.07 | 68.92±1.53 | 37.14±2.13 | 39.85±2.87 | 32.06±1.12 | 57.03±8.70 | 67.30±4.67 | 75.10±5.85 | 64.84±1.45 | 67.82±0.62 | 86.47±0.66 | 9.46 |
| GAT | cos-graph | $\{\mathcal{G},\mathcal{G}'\}$ | $\theta_1\neq\theta_2$ | 89.97±0.80 | 76.08±1.70 | 49.61±0.73 | 82.75±0.90 | 77.13±1.20 | 39.21±2.81 | 40.40±3.30 | 33.05±1.20 | 70.66±7.77 | 66.76±7.23 | 78.82±6.76 | 86.60±1.75 | 75.05±1.36 | 87.85±0.72 | 4.71 |
| GAT | cos-node | $\{\mathcal{G}'\}$ | - | 87.64±8.40 | 68.80±2.39 | 46.37±1.06 | 77.77±1.86 | 73.65±1.47 | 38.65±2.46 | 40.33±3.25 | 33.43±0.94 | 64.64±9.09 | 65.41±8.48 | 75.10±6.13 | 87.08±1.66 | 75.99±1.49 | 89.59±0.49 | 5.82 |
| GAT | cos-node | $\{\mathcal{G},\mathcal{G}'\}$ | $\theta_1=\theta_2$ | 53.16±7.93 | 63.67±1.08 | 44.83±2.04 | 73.46±1.07 | 68.92±1.53 | 37.14±2.13 | 39.85±2.87 | 32.06±1.12 | 57.03±8.70 | 67.30±4.67 | 75.10±5.85 | 64.84±1.45 | 67.82±0.62 | 86.47±0.66 | 9.46 |
| GAT | cos-node | $\{\mathcal{G},\mathcal{G}'\}$ | $\theta_1\neq\theta_2$ | 90.03±0.78 | 77.56±2.75 | 50.36±0.70 | 82.72±1.16 | 76.83±1.16 | 38.97±3.12 | 40.56±3.77 | 33.49±1.35 | 70.39±7.34 | 65.95±6.77 | 78.63±6.59 | 86.64±1.78 | 75.32±1.04 | 87.87±0.61 | 4.21 |
| GAT | kNN | $\{\mathcal{G}'\}$ | - | 84.27±5.25 | 68.73±1.47 | 46.05±0.90 | 77.57±1.75 | 71.58±1.62 | 38.82±2.33 | 40.12±3.69 | 33.84±1.07 | 61.68±8.71 | 67.57±7.43 | 74.90±5.86 | 86.77±1.90 | 75.64±1.45 | 88.29±0.48 | 6.50 |
| GAT | kNN | $\{\mathcal{G},\mathcal{G}'\}$ | $\theta_1=\theta_2$ | 53.16±7.93 | 63.67±1.08 | 44.83±2.04 | 73.46±1.07 | 68.92±1.53 | 37.14±2.13 | 39.85±2.87 | 32.06±1.12 | 57.03±8.70 | 67.30±4.67 | 75.10±5.85 | 64.84±1.45 | 67.82±0.62 | 86.47±0.66 | 9.46 |
| GAT | kNN | $\{\mathcal{G},\mathcal{G}'\}$ | $\theta_1\neq\theta_2$ | 89.96±0.79 | 77.23±1.63 | 49.79±0.72 | 82.78±0.95 | 76.67±1.13 | 39.65±2.76 | 41.11±3.92 | 33.54±1.36 | 70.38±7.22 | 65.95±6.52 | 77.84±7.23 | 86.97±1.75 | 75.20±1.55 | 87.97±0.51 | 4.18 |

Table 1 shows the performance of MLP, GNN baselines, and GNN+GSL across 8 datasets, using the best-performing GSL bases. For each GNN backbone, the best-performing method is highlighted in red, while the second-best method is highlighted in blue. Notably, under fair comparison conditions, all 4 baseline GNNs outperform their GNN+GSL counterparts. This suggests that incorporating GSL into these GNN baselines does not consistently yield performance improvements and leads to worse results in many instances. However, these results alone are insufficient to conclusively determine the effectiveness of GSL, as the method may require specific training procedures or more complex model designs. Therefore, we further examine the performance of state-of-the-art (SOTA) GSL approaches to more fairly evaluate GSL's potential within GNNs.

**SOTA-GSL.** To fairly reassess the impact of GSL in state-of-the-art (SOTA) methods, we compare the performance of SOTA models with their SOTA-GSL counterparts within the same hyperparameter search space. Corresponding to the analysis of GCN and MLP in Section 4.1, the SOTA-GSL methods include two variants: (1) SOTA, $\mathcal{G}' = \mathcal{G}$, which replaces the GSL graph $\mathcal{G}'$ with the original graph $\mathcal{G}$; and (2) SOTA, $\mathcal{G}' = $ MLP, which substitutes the graph convolution layers of GSL $\mathcal{G}'$ with MLP layers. The results are presented in Table 2, where "OOM" denotes out-of-memory. It is evident that removing GSL does not diminish model performance; in fact, it is often comparable to or even exceeds the original results. Furthermore, GSL-based SOTA methods require significantly more GPU memory and longer running times compared to their non-GSL counterparts. Based on

these findings, we conclude that GSL not only fails to enhance performance across most datasets but also increases model complexity. In conjunction with the results in Table 1, we assert that GSL is unnecessary for effective GNN design in most cases.

Table 2: Model Performance and training time per epoch of SOTA methods and SOTA-GSL. The results for methods marked with "*" are reported in Zhiyao et al. (2024).

| Model | Questions | | Minesweeper | | Roman-empire | | Amazon-ratings | | Tolokers | | Cora | | Pubmed | | Citeseer | |
|---|---|---|---|---|---|---|---|---|---|---|---|---|---|---|---|---|
| | AUC | Time | AUC | Time | Acc | Time | Acc | Time | AUC | Time | Acc | Time | Acc | Time | Acc | Time |
| GAug* | OOM | - | 77.93±0.64 | - | OOM | - | 48.42±0.39 | - | OOM | - | 82.48±0.66 | 7s | 78.73±0.77 | 20s | 72.79±0.86 | 10s |
| GAug, $\mathcal{G}' = \mathcal{G}$ | OOM | - | 80.56±0.36 | 11s | OOM | - | 48.45±0.37 | 12s | OOM | - | 81.73±0.38 | 1s | 79.38±0.46 | 6s | 72.34±0.18 | 2s |
| GAug, $\mathcal{G}' = $ MLP | OOM | - | 64.31±1.40 | 4.8s | OOM | - | 48.05±0.66 | 37s | OOM | - | 78.90±0.00 | 3.2s | 77.40±0.00 | 8.1s | 72.91±0.32 | 9s |
| GEN* | OOM | - | 79.56±1.09 | 260s | OOM | - | 49.17±0.68 | - | OOM | - | 81.66±0.91 | 214s | 78.49±3.98 | 1384s | 73.21±0.62 | 470s |
| GEN, $\mathcal{G}' = \mathcal{G}$ | OOM | - | 80.81±0.23 | 75s | OOM | - | 50.08±0.30 | 130s | OOM | - | 82.16±0.37 | 39s | 80.49±0.13 | 114s | 71.52±0.34 | 25s |
| GEN, $\mathcal{G}' = $ MLP | OOM | - | 71.81±0.98 | 12s | OOM | - | 49.29±0.65 | 49s | OOM | - | 80.20±0.00 | 140s | 66.80±0.00 | 1592s | 73.50±0.00 | 310s |
| GRCN* | 74.50±0.84 | - | 72.57±0.49 | 60s | 44.41±0.41 | 180s | 50.06±0.38 | 220s | 71.27±0.42 | 37s | 84.61±0.34 | 13s | 79.30±0.34 | 17s | 72.34±0.34 | 20s |
| GRCN, $\mathcal{G}' = \mathcal{G}$ | 75.69±0.52 | 8s | 71.15±0.05 | 10s | 45.84±0.52 | 8s | 46.07±1.02 | 10s | 71.73±0.42 | 10s | 81.66±1.10 | 2s | 79.35±0.26 | 3s | 69.55±1.28 | 2s |
| GRCN, $\mathcal{G}' = $ MLP | 63.59±2.35 | 3.9s | 72.18±1.09 | 2s | 45.89±0.83 | 7.5s | 48.77±0.60 | 8.1s | 70.45±1.39 | 8s | 79.40±0.00 | 1.3s | 78.10±0.00 | 5s | 71.40±0.00 | 4.2s |
| IDGL* | OOM | - | 50.00±0.00 | 157s | 47.10±0.65 | 186s | 45.87±0.58 | - | 50.00±0.00 | 279s | 84.19±0.61 | 123s | 82.78±0.44 | 146s | 73.26±0.53 | 332s |
| IDGL, $\mathcal{G}' = \mathcal{G}$ | OOM | - | 50.00±0.00 | 51s | 41.24±0.86 | 42s | OOM | - | 50.00±0.00 | 52s | 82.43±0.45 | 13s | 73.50±1.85 | 23s | 73.13±0.49 | 36s |
| IDGL, $\mathcal{G}' = $ MLP | OOM | - | 79.56±1.26 | 13.7s | 50.35±0.36 | 35s | 39.93±0.88 | 15s | 71.55±1.08 | 11s | 83.20±0.00 | 6.6s | 79.20±0.00 | 13s | 72.60±0.00 | 13.9s |
| NodeFormer* | OOM | - | 77.29±1.71 | - | 56.54±3.73 | - | 41.33±1.25 | - | OOM | - | 78.81±1.21 | 213s | 78.38±1.94 | - | 70.39±2.04 | 219s |
| NodeFormer, $\mathcal{G}' = \mathcal{G}$ | OOM | - | 80.66±0.82 | 215s | 68.37±1.95 | 236s | OOM | - | OOM | - | 77.01±1.99 | 152s | OOM | - | 70.82±0.13 | 139s |
| NodeFormer, $\mathcal{G}' = $ MLP | OOM | - | 80.04±1.42 | 21s | 53.08±2.37 | 7.2s | 71.55±1.08 | 26s | OOM | - | 78.82±0.00 | 8s | 76.30±0.00 | 127s | 72.80±0.00 | 15s |
| GloGNN | 68.67±1.07 | 66.6s | 52.45±0.30 | 13.0s | 66.21±0.17 | 26.1s | 50.72±0.88 | 31.1s | 79.81±0.20 | 47.4s | 78.07±1.66 | 6.6s | 87.88±0.26 | 18.2s | 71.95±1.90 | 21.8s |
| GloGNN, $\mathcal{G}' = \mathcal{G}$ | 68.32±1.23 | 49.4s | 52.30±0.21 | 3.6s | 66.03±0.14 | 15.3s | 50.23±0.83 | 21.7s | 80.02±0.16 | 25.1s | 73.49±2.01 | 5.1s | 87.62±0.20 | 14.4s | 72.27±2.08 | 21.2s |
| GloGNN, $\mathcal{G}' = $ MLP | 69.69±0.22 | 25.7s | 52.30±0.20 | 2.1s | 66.49±0.16 | 12.4s | 49.56±0.73 | 12.3s | 74.85±0.12 | 2.8s | 73.93±1.81 | 3.2s | 87.64±0.27 | 10.2s | 72.09±1.81 | 13.8s |
| WRGAT* | OOM | - | 90.22±0.64 | 168.0s | OOM | - | OOM | - | 78.69±1.21 | 153.0s | 84.28±1.52 | 19.5s | 88.82±0.50 | 421.6s | 73.50±1.41 | 22.1s |
| WRGAT, $\mathcal{G}' = \mathcal{G}$ | 74.67±0.95 | 64.1s | 89.79±0.37 | 18.6s | OOM | - | 50.41±0.53 | 49.9s | 78.81±0.89 | 47.0s | 83.48±1.48 | 3.4s | 88.92±0.43 | 26.5s | 73.22±1.90 | 4.7s |
| WRGAT, $\mathcal{G}' = $ MLP | 68.07±2.62 | 75.8s | 87.08±2.11 | 16.2s | OOM | - | 41.38±1.46 | 24.4s | 76.41±1.25 | 37.7s | 76.99±1.10 | 2.9s | 80.27±6.23 | 23.9s | 65.28±2.11 | 4.5s |
| WRGCN | 74.70±1.71 | 358.3s | 90.63±0.64 | 40.9s | OOM | - | 52.76±0.95 | 508.4s | 82.68±0.82 | 52.3s | 88.30±1.48 | 23.7s | OOM | - | 73.74±1.60 | 54.2s |
| WRGCN, $\mathcal{G}' = \mathcal{G}$ | 75.91±1.30 | 43.3s | 90.65±0.49 | 5.5s | OOM | - | 52.54±0.56 | 50.1s | 82.65±0.86 | 15.6s | 88.32±0.79 | 3.9s | 89.26±0.45 | 19.4s | 74.45±1.51 | 10.5s |
| WRGCN, $\mathcal{G}' = $ MLP | 64.59±1.48 | 23.1s | 70.66±1.37 | 7.7s | OOM | - | 37.05±0.46 | 8.0s | 69.10±0.91 | 12.2s | 70.00±3.59 | 2.2s | 67.29±2.49 | 9.9s | 70.84±1.36 | 4.1s |

## 5.2 QUALITY OF GSL GRAPHS

Previous studies (Li et al., 2022b; Zheng et al., 2024b) suggest that GSL constructs graphs with properties that improve intra-class node connectivity, which can be measured by homophily. This improvement can be visualized by inspecting graph structures with nodes sorted by their class labels. A graph that appears closer to a block diagonal matrix indicates stronger intra-class connectivity. However, this enhancement may not always be essential and can be achieved through non-GSL methods as well. In Figure 4, we visualize the original and reconstructed structures of a heterophilous graph from the Wisconsin dataset. The GSL graphs are constructed using various bases: $\mathbf{X}$, $\hat{\mathbf{A}}\mathbf{X}$, MLP($\mathbf{X}$), GCN($\mathbf{X}, \mathbf{A}$), and GCL($\mathbf{X}, \mathbf{A}$). We also include reconstructed graphs using a simple method that samples edges between nodes of the same class based on label predictions, *i.e.*, $\hat{\mathbf{Y}} = $ GCN($\mathbf{X}, \mathbf{A}$) or $\hat{\mathbf{Y}} = $ MLP($\mathbf{X}, \mathbf{A}$). Figure 4 demonstrates that, although GSL improves intra-class connectivity, the improvement is not as substantial as that achieved by non-GSL methods, as seen in the last two subfigures. Thus, the improvement in homophily within GSL graphs is unnecessary, as it can be easily achieved through simple methods.

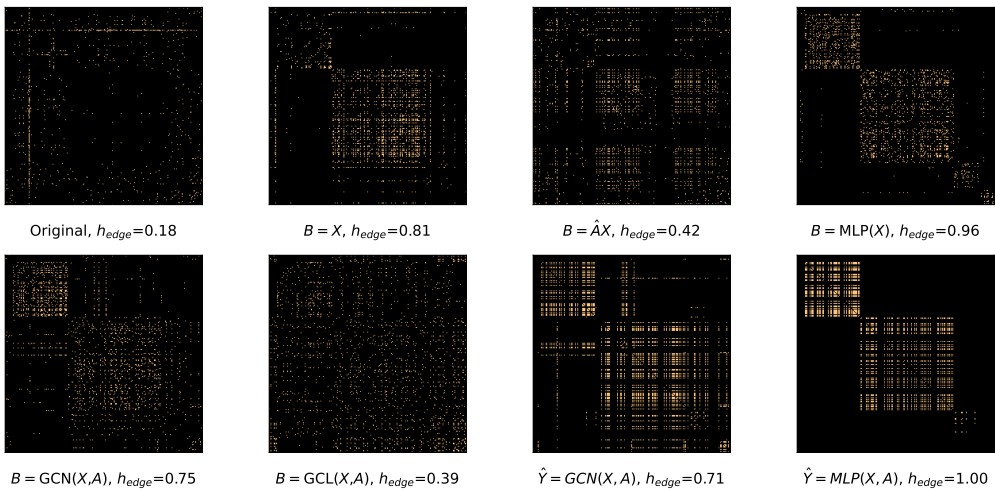

Original, $h_{edge}$=0.18    $B = X$, $h_{edge}$=0.81    $B = \hat{A}X$, $h_{edge}$=0.42    $B = $ MLP($X$), $h_{edge}$=0.96

$B = $ GCN($X,A$), $h_{edge}$=0.75    $B = $ GCL($X,A$), $h_{edge}$=0.39    $\hat{Y} = $ GCN($X,A$), $h_{edge}$=0.71    $\hat{Y} = $ MLP($X,A$), $h_{edge}$=1.00

Figure 4: Visualization of original graph and reconstructed graphs on Wisconsin

## 5.3 GSL COMPONENTS

Since the performance of GNN and GNN+GSL models is comparable under the same GSL bases, as shown in Table 1, we further investigate how different components of GSL influence GNNs. As illustrated in Figure 5, our results indicate that: (1) Pretrained node representations, such as MLP($\mathbf{X}$) and GCN($\mathbf{X}, \mathbf{A}$), significantly enhance GNN performance, (2) GSL graph generation has minimal impact on model performance, (3) two view fusion with parameter separation improves GNN performance, and (4) early fusion generally outperforms late fusion. These results explain why prior comparisons of GNNs are unfair since those pretrained GSL bases greatly improve GNN performance. This improvement stems from self-training, a key component in many GSL approaches. As a result, incorporating self-training methods may be more advantageous for future GNN designs than relying solely on GSL. For additional results and an analysis of GSL modules, please refer to Appendix E.

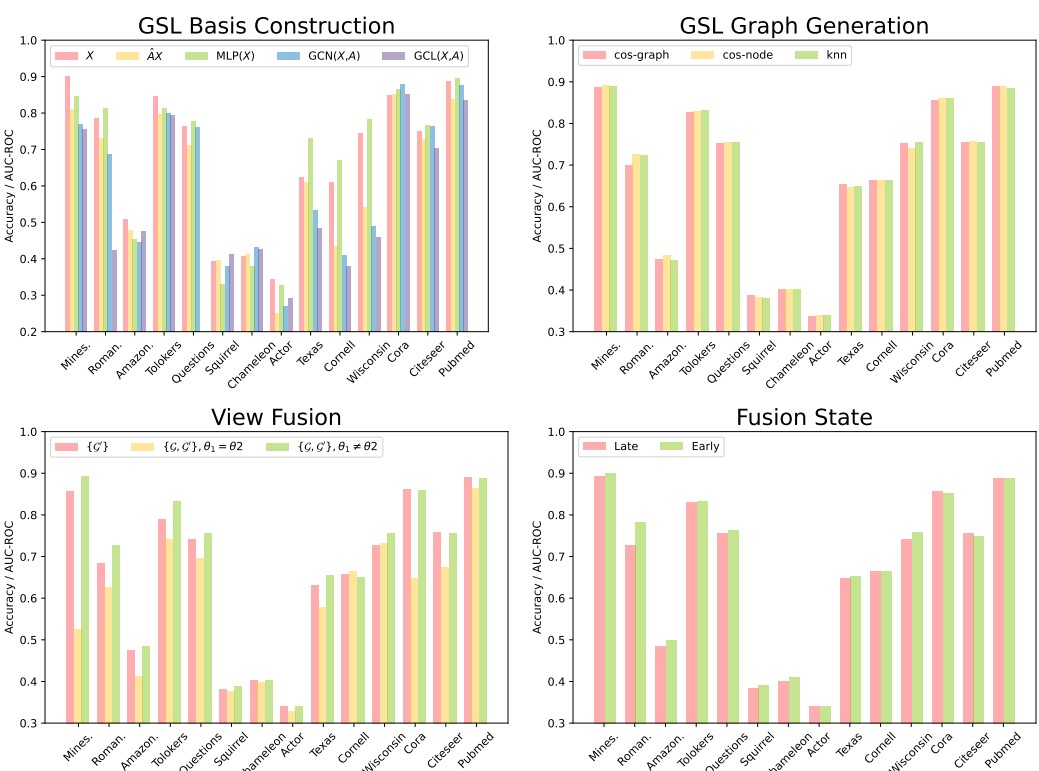

Figure 5: Influences of different GSL components to model performance.

## 6 CONCLUSION

In this paper, we revisit the role of Graph Structure Learning (GSL) in Graph Neural Networks (GNNs) with our proposed GSL framework. Motivated by the controversy of GSL, we demonstrate that graph convolution over GSL-constructed graphs does not improve mutual information, as confirmed by both empirical observations and theoretical analysis. By either adding GSL to baseline GNNs or removing it from state-of-the-art (SOTA) methods, we find that GSL does not enhance GNN performance when evaluated under the same GSL bases and hyperparameter tuning. These results suggest that the improvements attributed to GSL may stem from components other than GSL. Our findings contribute to a better understanding of GSL and offer insights into re-evaluating the essential components in future GNN design. While this paper primarily examines the influence of GSL on model performance in node classification tasks, future research could extend this analysis to other graph-related tasks. Additionally, investigating how GSL affects the robustness of GNNs would be a valuable direction for future study.

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

# A  TAXONOMY OF GRAPH STRUCTURE LEARNING METHODS

We present several representative GSL-based GNNs within our proposed GSL framework in Table 3. Below, we provide a detailed description of each method.

Table 3: Representative GSL methods under our proposed GSL framework

| Method | Bases | Construct | Refinement | View Fusion | Training Mode |
|---|---|---|---|---|---|
| LDS (Franceschi et al., 2020) | $X$ | $\{\mathcal{E}' = kNN(B)\ +\text{Opt.}\}$ | Bernoulli($\mathcal{E}'$) | Late Fusion, $\{\mathcal{G}'_1, \mathcal{G}'_2, \ldots, \mathcal{G}'_m\}$, $\theta_1 = \theta_2$ | 2-stage |
| Geom-GCN (Pei et al., 2020) | Isomap/Poincare/Struc2vec($X,A$) | $\{\mathcal{E}'|e'_{ij} = |B_i - B_j|\}$ | threshold($\mathcal{E}'$) | Late Fusion, $\{\mathcal{G}, \mathcal{G}'\}$, $\theta_1 \neq \theta_2$ | Static |
| ProGNN (Jin et al., 2020) | $\epsilon$ | $\{\mathcal{E}' = \text{Opt}(\epsilon)\}$ | Low Rank+Sparsity +Original | No Fusion, $\{\mathcal{G}'\}$ | Joint |
| IDGL (Chen et al., 2020) | MLP($X$) | $\{\mathcal{E}'|e'_{ij} = \cos(B_i,B_j)\}$ | topk($\mathcal{E}'$) | Early Fusion, $\{\mathcal{G} + \mathcal{G}'\}$ | Joint |
| GRCN (Yu et al., 2020) | GCN($X,A$) | $\{\mathcal{E}'|e'_{ij} = \sigma(B_i B_j^T)\}$ | topk($\mathcal{E}'$), sym($\mathcal{E}'$) | Early Fusion, $\{\mathcal{G} + \mathcal{G}'\}$ | Joint |
| GAug-M (Zhao et al., 2020) | GCN$^{(2)}$($X,A$) | $\{\mathcal{E}'|e'_{ij} = \sigma(B_i B_j^T)\}$ | $\mathcal{G}'_+ = $ topk($\mathcal{E}'$), $\mathcal{G}'_- = $ bottom($\mathcal{E}'$) | Early Fusion, $\{\mathcal{G} + \mathcal{G}'_+ - \mathcal{G}'_-\}$ | Joint |
| GAug-O (Zhao et al., 2020) | $X$ | $\{\mathcal{E}'|e'_{ij} = p(e_{ij}|\text{GAE}(B,A))\}$ | Gumbel($\mathcal{E}'$) | Early Fusion, $\{\mathcal{G} + \mathcal{G}'\}$ | Joint |
| SLAPS (Fatemi et al., 2021) | MLP($X$) | $\{\mathcal{E}' = kNN(B)\}$ | norm($\mathcal{E}'$),sym($\mathcal{E}'$) | No Fusion, $\{\mathcal{G}'\}$ | Joint |
| CoGSL (Liu et al., 2022a) | GCN($X$, $\{A$, kNN($X$), PPR($X$),Subgraph($X$)$\}$) | $\{\mathcal{E}'|e'_{ij} = p(e_{ij}|\text{MLP}(B,A))\}$ | - | Early Fusion, $\{\mathcal{G}^*| \min \mathcal{L}_{\text{CL}}(\mathcal{G},\mathcal{G}')\}$, $\theta_1 \neq \theta_2$ | 2-stage |
| GEN (Wang et al., 2021) | GCN($X,A$) | $\{\mathcal{E}' = kNN(B)\}$ | - | Late Fusion, $\{\mathcal{G}'_1, \mathcal{G}'_2, \ldots, \mathcal{G}'_m\}$, $\theta_1 \neq \theta_2$ | 2-stage |
| STABLE (Li et al., 2022a) | GCL($X,A$) | $\{\mathcal{E}'|e'_{ij} = \cos(B_i,B_j)$ or $\cos(B_i,B_j)\}$ | $\mathcal{G}'_+ = $ topk($\mathcal{E}'$), $\mathcal{G}'_- = $ threshold($\mathcal{E}'$) | Early Fusion, $\{\mathcal{G} + \mathcal{G}'_+ - \mathcal{G}'_-\}$ | Joint |
| SEGSL (Zou et al., 2023) | $X$ | $\{\mathcal{E}'| \min \mathcal{H}_S, e'_{ij} \in \text{EncTree}(kNN(B))\}$ | - | No Fusion, $\{\mathcal{G}'\}$ | Joint |
| SUBLIME (Liu et al., 2022b) | GCN($X,A$) | $\{\mathcal{E}' = \text{Opt}(\epsilon)\}$ or $\{\mathcal{E}'|e'_{ij} = \cos/\text{Minkowski}(B_i,B_j)\}$ | topk($\mathcal{E}'$),sym($\mathcal{E}'$),norm($\mathcal{E}'$) | Separation, $\{\mathcal{G},\mathcal{G}'\}$, $\theta_1 = \theta_2$ | Joint |
| BM-GCN (He et al., 2021) | $\hat{Y} = $ MLP($X$), $\min \mathcal{L}_{\text{CE}}(\hat{Y}, Y)$ | $\{\mathcal{E}' = BQB^T\}$ | norm($\mathcal{E}'$) | Early Fusion, $\{\mathcal{G} \odot \mathcal{G}'\}$ | Joint |
| WSGNN (Lao et al., 2022) | MLP($X$) | $\{\mathcal{E}'|e'_{ij} = cos(B_i,B_j)\}$ | - | Early Fusion, $\{\mathcal{G} + \mathcal{G}'\}$ | Joint |
| GLCN (Jiang et al., 2019) | $X$ | $\{\mathcal{E}'|e'_{ij} = \phi(|B_i - B_j|)\}$ | norm($\mathcal{E}'$), Original +Sparsity+Smoothness | No Fusion, $\{\mathcal{G}'\}$ | Joint |
| ASC (Li et al., 2023a) | SpectralCluster($X$) | $\{\mathcal{E}'|e'_{ij} = \|B_i - B_j\|\}$ | topk($\mathcal{E}'$) | No Fusion, $\{\mathcal{G}'\}$ | Static |
| WRGAT (Suresh et al., 2021) | GCN($X,A$) | $\{\mathcal{E}'|e'_{ij} \cdot \text{Opt}(B)\}$ | Sparsity + MultiHop | Early Fusion $\{\mathcal{G} + \mathcal{G}'\}$ | Static |
| HOG-GCN (Wang et al., 2022) | GCN($X,A$) | $\{\mathcal{E}'|e'_{ij} = \sigma(B_i B_j^T)\}$ | Sparsity + Smoothness | No Fusion $\{\mathcal{G}'\}$ | Joint |
| GGCN (Yan et al., 2022) | MLP($X$) | $\{\mathcal{E}'|e'_{ij} = \cos(B_i,B_j)\}$ | Low Rank + Sparsity | Early Fusion, $\{\mathcal{G} + \mathcal{G}'\}$ | Joint |
| GloGNN (Li et al., 2022b) | MLP($X$) | $\{\mathcal{E}' = \text{Opt}(B)\}$ | Sparsity+MultiHop | No Fusion, $\{\mathcal{G}'\}$ | Joint |
| HiGNN (Zheng et al., 2024b) | $\hat{Y} = $ GCN($X,A$), $\min \mathcal{L}_{CE}(\hat{Y}, Y)$ | $\{\mathcal{E}' = e'_{ij} = \cos(B_i,B_j))\}$ | topk($\mathcal{E}'$), sym($\mathcal{E}'$) | Late Fusion, $\{\mathcal{G}, \mathcal{G}'\}$, $\theta_1 \neq \theta_2$ | Static |

**LDS** (Franceschi et al., 2020). The GSL bases in LDS is constructed as node features $X$ and the GSL graph $\mathcal{G}'$ is initialized using a k-Nearest-Neighbors algorithm based on $B$. Then, $\mathcal{G}'$ is updated with a loss function of node classification. Then multiple graphs are sampled based on $\mathcal{G}'$ with a Bernoulli function and used to update the model parameters. The $\mathcal{G}'$ construction and model parameters are updated as a 2-stage mode.

**Geom-GCN** (Pei et al., 2020). Geom-GCN constructs the GSL bases from several graph-aware node embedding strategies using both of the $X$ and $A$: Isomap (), Poincare (), and struc2vec (). Then, new graphs are constructed by filtering node pairs with a higher similarity measured by Euclidean distance $\{\mathcal{E}'|e'_{ij} = |B_i - B_j| < \delta\}$ where $\delta$ is a threshold. Finally, both of the aggregated message from $\mathcal{G}$ and $\mathcal{G}'$ are fused after applying graph convolution layers with no parameter sharing. The $\mathcal{G}'$ is not updated through the training process.

**ProGNN** (Jin et al., 2020). The $\mathcal{G}'$ in ProGNN is purely learned by optimization without GSL bases. It optimizes the $\mathcal{G}'$ using low rank, sparsity, and similarity with the original graphs $\mathcal{G}$. It outputs a single graph $\mathcal{G}'$ without fusion and updates the $\mathcal{G}'$ together with model parameters.

**IDGL** (Chen et al., 2020). The GSL bases in LDS is constructed by linear transformation of node features MLP($X$). Then, a GSL graph $\mathcal{G}'$ is constructed using cosine similarity with topk threshold refinement. The early fusion is applied by fusing GSL graph $\mathcal{G}'$ with original graph $\mathcal{G}$ before training. The GSL graph $\mathcal{G}'$ is trained with model parameters jointly.

**GRCN** (Yu et al., 2020). GRCN constructs GSL bases by node embeddings of graph convolution GCN($X, A$). Then, the GSL graph $\mathcal{G}'$ is constructed by a kernel function with topk and symmetrization refinement $\{\mathcal{E}'|e'_{ij} = \sigma(B_i B_j) > \delta\}$. The final graph is obtained by early fusion and the GSL graph $\mathcal{G}'$ is updated together with model parameters.

**GAug-M** and **GAug-O** (Zhao et al., 2020). GAug-M constructs GSL bases using a 2-layer graph convolution GCN$^{(2)}$($X, A$). Then, the GSL graph $\mathcal{G}'$ is constructed by a kernel function. The final graph is obtained by adding some edges with highest probabilities and removing some edges with lowest probabilities on $\mathcal{G}$. GAug-O selects node features as GSL bases $X$, then trains a Graph Auto-Encoder to predict edges as $\mathcal{G}'$. Then, after gumbel sampling, the GSL graph $\mathcal{G}'$ is fused with original graph $\mathcal{G}$ before training. The $\mathcal{G}'$ in both of the GAug-M and GAug-O is updated together with model parameters.

**SLAPS** (Fatemi et al., 2021). SLAPS constructs the GSL bases by applying MLP(X) followed by a k-nearest neighbors (kNN) algorithm based on node feature similarities. The GSL graph $\mathcal{G}'$ is then processed by an adjacency processor that symmetrizes and normalizes the adjacency matrix to ensure non-negativity and symmetry. The final graph is obtained of the generated graph $\mathcal{G}'$ with the node features without fusion. Additionally, a self-supervised denoising autoencoder $L_{DAE} = L(X_i, GNN_{DAE}(\hat{X}_i; \theta_{GNN_{DAE}}))$ is introduced to address the supervision starvation problem, updating $\mathcal{G}'$ together with the model parameters.

**CoGSL** (Liu et al., 2022a). CoGSL constructs GSL bases using two views, one of them is the Origin graph. Another is selected from the Adjacency matrix $A$, Diffusion matrix $PPR(X)$, the KNN graph $KNN(X)$ and the Subgraph of the Origin. GCNs are applied to these views to obtain node embeddings. The GSL graph is constructed by applying a linear transformation to the node embeddings of each node pair to estimate the connection probability between them. This connection probability is then added to the original view to finalize the graph. The refinement $\mathcal{E}'|e'_{ij} = p(e_{ij}|\text{MLP}(\mathbf{B}, \mathbf{A}))$ step involves maximizing the mutual information between the two selected views and the newly constructed graph. InfoNCE loss is used to optimize the connection probability, where the same node serves as a positive sample, and different nodes serve as negative samples. The final graph $\mathcal{G}'$ is obtained via early fusion of the selected views, and the GSL graph is updated with model parameters.

**GEN** (Wang et al., 2021). GEN constructs the GSL bases by generating kNN graphs though several GCN layer, utilizing node representations from different layers. These kNN graphs are then combined using a Stochastic Block Model (SBM) to create a new graph $\mathcal{G}'$. The GSL graph $\mathcal{G}'$ is refined iteratively through Bayesian inference to maximize posterior probabilities $P(G, \alpha, \beta|O, Z, Y_l) = \frac{P(O|G,\alpha,\beta)P(G,\alpha,\beta)P(O,Z,Y_l)}{P(O,Z,Y_l)}$, considering both the original graph and node embeddings. The final graph is obtained by feeding the graph $Q$ back into the GCN for further optimization. The iterative process updates both the GSL graph and GCN parameters as a 2-stage mode, providing mutual reinforcement between the graph estimation and model learning.

**STABLE** (Li et al., 2022a). STABLE constructs the GSL bases by generating augmentations based on node similarity through kNN graph and perturbing edges to simulate adversarial attacks. The GSL graph $\mathcal{G}'$ is constructed by refining the structure using contrastive learning between positive samples (slightly perturbed graphs) and negative samples (undesirable views generated by feature shuffling). The refinement step applies a top-k filtering strategy on the node similarity matrix to retain helpful edges while removing adversarial ones. The final graph is obtained through early fusion, and the GSL graph $\mathcal{G}'$ is updated together with model parameters during joint training

**SE-GSL** (Zou et al., 2023). SE-GSL constructs the GSL bases using a kNN graph fused with the original graph. The GSL graph $\mathcal{G}'$ is constructed through a structural entropy minimization process that extracts hierarchical community structures in the form of an encoding tree. The final graph is optimized by sampling node pairs from the encoding tree and generating new edges based on the minimized entropy structure. The refined graph is then used for downstream tasks, and the GSL graph $\mathcal{G}'$ is updated jointly with model parameters during training.

**SUBLIME** (Liu et al., 2022b). SUBLIME constructs the GSL bases using both an anchor view (original graph) and a learner view (new graph). The new graph is initialized through kNN and further optimized either by parameter-based methods (using models like MLP, GCN, or GAT) or by non-parameter-based approaches (using cosine similarity or Minkowski distance). After obtaining the new graph, post-processing operations such as top-k filtering, symmetrization, and degree-based regularization are applied to ensure the graph's sparsity and structure. The GSL graph $\mathcal{G}'$ is refined by applying contrastive learning between the anchor and learner views, incorporating edge drop and feature masking to generate node embeddings. The final graph is used in downstream tasks, and both views are updated together with model parameters in a joint training process.

**BM-GCN** (He et al., 2021). BM-GCN constructs the GSL bases by introducing soft labels for nodes enbedding $\mathbf{B} = softmax(\sigma(MLP(X)))$ via a multilayer perceptron $\mathcal{L}_{MLP} = \sum_{v_i \in \mathcal{V}} f(B_i, Y_i)$. These soft labels are then used to compute a block matrix (H) , which models the connection probabilities between different node classes. The GSL graph $\mathcal{G}'$ is constructed by creating a block similarity matrix $Q = HH^T$ from the block matrix $Y_s = Y_i, B_i|\forall v_i \in \mathcal{T}_y, \forall v_j \notin \mathcal{T}_y, H = (Y_s^T AY_s) \circ (Y_s^T AE)$, reflecting similarities between classes. The new graph is optimized using $BQB^T$ and further fused with the original graph $A + \beta I$ for downstream tasks. The final graph is

obtained by optimizing $\mathcal{G}'$ through degree-based regularization and top-k filtering. The GSL graph $\mathcal{G}'$ is updated together with model parameters during joint training.

**WSGNN** (Lao et al., 2022). WSGNN introduces a two-branch graph structure learning method, where each branch operates on different aspects of the graph: Branch AZ learns node labels from the new graph structure, while Branch ZA learns the new graph structure from the labels. The GSL bases is constructed using the observed graph $A_{obs}$ and node features $X$. The new graph $A'$ is inferred via cosine similarity between node embeddings. After constructing two separate views from each branch, the final graph is obtained by averaging the graphs from both branches. The refinement process ensures sparsity through cosine-based edge calculation $\mathcal{E}'|e'_{ij} = cos(\mathbf{B}_i, \mathbf{B}_j)$. Finally, both views undergo early fusion, with graph structure and node labels optimized jointly using a composite loss function that includes ELBO for structure prediction and cross-entropy loss for label prediction. The final GSL graph $\mathcal{G}'$ is updated during joint training.

**GLCN** (Jiang et al., 2019). GLCN constructs the GSL bases by computing pairwise distances between node features and passing them through an MLP to obtain a block similarity score. This score is then processed with a softmax function to generate an $n \times n$ probability matrix that serves as the learned graph structure. The graph is refined using regularization techniques to ensure sparsity and feature smoothness $L_{GL} = \sum_{i,j=1}^{n} ||x_i - x_j||_2^2 S_{ij} + \gamma ||S||_F^2 + \beta ||S - A||_F^2$. The learned graph is then used for downstream graph tasks, where the task loss and the graph regularization loss are jointly optimized during joint training

**ASC** (Li et al., 2023a). ASC constructs the GSL bases is formed by using pseudo-eigenvectors from spectral clustering. They divide the Laplacian spectrum into slices, with each slice corresponding to an embedding matrix. The GSL graph $\mathcal{G}'$ is constructed by adaptive spectral clustering, where pseudo-eigenvectors are weighted based on alignment with node labels Where $f_i^{\mathcal{Z}}$. For refinement, they apply top-K edge selection by minimizing node embedding distance and maximizing homophily $\underset{\mathcal{Z}}{\arg\min} \sum_{i,j \in V_Y} (d(f_i^Z, f_j^Z), 1(y_i, y_j))$. This final restructured graph is training without fusion. Finally, the GSL graph is updated together with the model parameters.

**WRGAT** (Suresh et al., 2021). WRGAT constructs the GSL bases using the node features and a weighted relational GNN (WRGNN) framework that fuses structural and proximity information. A multi-relational graph is built by assigning different types of edges based on the structural equivalence of nodes at various neighborhood levels. This framework adapts to both assortative and disassortative mixing patterns, which helps improve node classification tasks. The GSL graph $\mathcal{G}'$ is refined through attention-based message passing across these relational edges, and early fusion of proximity and structural features is used. The GSL graph $\mathcal{G}'$ is trained jointly with the model parameters to optimize the node classification task.

**HOG-GCN** (Wang et al., 2022). HOG-GCN constructs the GSL bases by incorporating both topological information and node attributes to estimate a homophily degree matrix $S = BB^T, B = softmax(Z_m), Z_m^{(l)} = \sigma(Z_m^{(l-1)W_m^{(l)}})$. The GSL graph $\mathcal{G}'$ is constructed using a homophily-guided propagation mechanism, which adapts the feature propagation weights between neighborhoods based on the homophily degree matrix $Z^{(l)} = \sigma(\mu Z^{(l-1)} W_e^{(l)} + \xi \hat{D}^{(-1)} A_k \odot H Z^{(l-1)} W_n^{(l)})$. For refinement, the graph incorporates both k-order structures and class-aware information to model the homophily and heterophily relationships between nodes. The final graph is obtained through joint fusion of topological and attribute-based homophily degrees, and both graph structure and model parameters are updated during joint training.

**GGCN** (Yan et al., 2022). GGCN constructs the GSL bases using node features and structural properties such as node-level homophily $h_i$ and relative degree $\bar{r}_i$. It incorporates structure-based edge correction by learning new edge weights derived from structural properties like node degree, and feature-based edge correction by learning signed edge weights from node features, allowing for positive and negative influences between neighbors. The GSL graph $\mathcal{G}'$ is constructed by combining signed and unsigned edge information, aiming to capture both homophily and heterophily. The refinement process uses edge correction and decaying aggregation to mitigate oversmoothing and heterophily problems. The final graph is updated with early fusion, and the GSL graph $\mathcal{G}'$ is optimized during joint training

**GloGNN** (Li et al., 2022b). GloGNN constructs its GSL bases using node embeddings derived from MLP, combining both low-pass and high-pass convolutional filters. A coefficient matrix $Z^{(l)}$ is used to characterize the relationship between nodes and is optimized to capture both feature and structural similarities $H_X^{(0)} = (1 - \alpha)H_X^{(0)} + \alpha H_A^{(0)}$. Refinement is achieved via top-k selection based on the multi-hop adjacency matrix, and the matrix is symmetrized. The final graph is obtained through global aggregation of nodes, capturing both local and distant homophilous nodes. This graph is then used in downstream tasks, where the GSL graph $\mathcal{G}'$ is jointly optimized with the model parameters.

**HiGNN** (Zheng et al., 2024b). HiGNN constructs its GSL bases by utilizing heterophilous information as node neighbor distributions, which represent the likelihood of neighboring nodes belonging to different classes $\mathcal{H}_u = [p_1, p_2, ..., p_c], where \ p_i = \frac{|v|v\in\mathcal{N}_u, y_v=i|}{|\mathcal{N}_u|}$. A new graph structure $\mathcal{G}'$ is constructed by linking nodes with similar heterophilous distributions using cosine similarity. The refinement involves selecting top-k edges based on the similarity score and applying symmetrization. The final graph is fused with the original adjacency matrix $A$ and the newly constructed adjacency matrix $A'$ via late fusion during message passing, where the node embeddings from both $A$ and $A'$ are combined with a balance parameter $\lambda$. The graph $\mathcal{G}'$ and node embeddings are updated during static training.

## B   PROOF OF THEOREM

**Theorem 1.** Given a graph $\mathcal{G} = \{\mathcal{V}, \mathcal{E}\}$ with node labels $\mathbf{Y}$ and node features $\mathbf{X}$, the accuracy of graph convolution in node classification is upper bounded by the mutual information between the node label $Y$ and the aggregated node features $H$:

$$P_A \leq \frac{I(Y; H) + \log 2}{\log(C)} \tag{5}$$

*Proof.* For an arbitrary node $u$, the aggregated node features can be derived as $H_u = \frac{1}{|\mathcal{N}_u|} \sum_{v \in \mathcal{N}_u} X_v$ following the graph convolution operation. For a classifier predicting labels based on $H_u$, we have $\hat{Y}_u = \text{cls}(H_u)$. Consequently, the Markov chain $Y \to H \to \hat{Y}$ holds. By applying Fano's inequality (Gerchinovitz et al., 2020), we obtain

$$H(Y|H) \leq H_b(P_E) + P_E \log(C - 1) \tag{6}$$

where $P_E$ represents the error rate and $H_b(\cdot)$ is the binary entropy function. Rearranging this inequality gives us a lower bound on $P_E$:

$$P_E \geq \frac{H(Y|H) - H_b(P_E)}{\log(C - 1)} \tag{7}$$

Since $H(Y|H) = H(Y) - I(Y; H) = \log(C) - I(Y; H)$ and $H_b(P_E) \leq \log 2$, we can substitute these terms into the equation:

$$P_E \geq 1 - \frac{I(Y; H) + \log 2}{\log(C)} \tag{8}$$

Finally, by expressing the accuracy rate $P_A$, we find:

$$P_A = 1 - P_E \leq \frac{I(Y; H) + \log 2}{\log(C)} \tag{9}$$

This concludes the proof of Theorem 1.

**Proposition 1.** Consider a graph $\mathcal{G} = \{\mathcal{V}, \mathcal{E}\}$ characterized by node labels $Y$ and $n$-dimensional node bases $\mathbf{B} = \{B_1, B_2, \ldots, B_n\}$ with $C$ classes. Each base $B_i$ is independent and follows a class-dependent Gaussian distribution, *i.e.,* $B_i \sim \mathcal{N}(\mu_Y, \sigma_Y)$. A new graph $\mathcal{G}' = \{\mathcal{V}, \mathcal{E}'\}$ is generated using a non-parametric method based on the bases $\mathbf{B}$. For the aggregated bases $\boldsymbol{B}'$ on $\mathcal{G}'$, we have $\inf I(Y; \boldsymbol{B}') \leq \inf I(Y; \boldsymbol{B})$.

*Proof.* Let's first consider the mutual information for $i$-th node base $B_i$. For a non-parametric GSL method, we have the probability that class $k$ connects with class $j$ as:

$$p_{k,j} = \frac{g(B_i^k, B_i^j)}{\sum_{q=1}^{C} g(B_i^k, B_i^q)} \tag{10}$$

where $g(\cdot)$ is a non-parametric measurement of the probability of new connections, such as cosine similarity or Minkowski Distance. Then, we can get aggregated bases from the new graph by the operation of graph convolution (Ma et al., 2021; Luan et al., 2024b):

$$B_i'^k = \sum_{q=1}^{C} p_{k,q} B_i^q \tag{11}$$

Therefore, the Markow chain $Y \to B_i \to B_i'$ holds. From data processing inequality (Beaudry & Renner, 2012), we have

$$I(Y; B_i') \leq I(Y, B_i) \tag{12}$$

To extend this conclusion to multi-dimensional variables, we apply the chain rule of mutual information

$$I(Y; \mathbf{B}) = I(Y; \{B_1, \ldots, B_n\}) = \sum_{i=1}^{n} I(Y; B_i \mid \{B_1, \ldots, B_{i-1}\})$$
$$I(Y; \mathbf{B}') = I(Y; \{B_1', \ldots, B_n'\}) = \sum_{i=1}^{n} I(Y; B_i' \mid \{B_1', \ldots, B_{i-1}'\}) \tag{13}$$

Due to the property that conditioning reduces entropy, we have

$$I(Y; B_i \mid \{B_1, \ldots, B_{i-1}\}) \geq I(Y; B_i)$$
$$I(Y; B_i' \mid \{B_1', \ldots, B_{i-1}'\}) \geq I(Y; B_i') \tag{14}$$

Thus, we have

$$\inf I(Y; \mathbf{B}) = \sum_{i=1}^{n} I(Y; B_i) \text{ and } \inf I(Y; \mathbf{B}') = \sum_{i=1}^{n} I(Y; B_i') \tag{15}$$

where $\inf$ represents infimum. Since $I(Y; B_i') \leq I(Y, B_i)$ holds for each $i$, we have

$$\inf I(Y; \boldsymbol{B'}) \leq \inf I(Y; \boldsymbol{B}) \tag{16}$$

This concludes the proof of Proposition 1.

## C  DATASET DETAILS

The datasets used in our experiments include heterophilous graphs: Squirrel, Chameleon, Actor, Texas, Cornell, and Wisconsin (Pei et al., 2020; Rozemberczki et al., 2021), homophilous graphs: Cora, PubMed, and Citeseer (Yang et al., 2016), and Minesweeper, Roman-empire, Amazon-ratings, Tolokers, and Questions (Platonov et al., 2023). The dataset statistics are shown in 4. The descriptions of all the datasets are given below:

**Cora**, **Citeseer**, and **Pubmed** datasets are widely used citation networks in graph structure learning research. In each dataset, nodes represent academic papers, while edges capture citation relationships between them. The node features are bag-of-words vectors derived from the paper's content, and each node is assigned a label based on its research topic. These datasets offer a structured framework to evaluate GNN models on classification tasks within citation networks.

Table 4: Dataset Statistics

| Dataset | #Nodes | #Edges | #Classes | #Features | Edge Homophily |
|---|---|---|---|---|---|
| Cora | 2,708 | 5,278 | 7 | 1,433 | 0.81 |
| Pubmed | 19,717 | 44,324 | 3 | 500 | 0.80 |
| Citeseer | 3,327 | 4,552 | 6 | 3,703 | 0.74 |
| Roman-empire | 22,662 | 32,927 | 18 | 300 | 0.05 |
| Amazon-ratings | 24,492 | 93,050 | 5 | 300 | 0.38 |
| Minesweeper | 10,000 | 39,402 | 2 | 7 | 0.68 |
| Tolokers | 11,758 | 529,000 | 2 | 10 | 0.59 |
| Questions | 48,921 | 153,540 | 2 | 301 | 0.84 |
| Cornell | 183 | 295 | 5 | 1,703 | 0.30 |
| Chameleon | 2,277 | 36,101 | 5 | 2,325 | 0.23 |
| Wisconsin | 251 | 466 | 5 | 1,703 | 0.21 |
| Texas | 183 | 309 | 5 | 1,703 | 0.11 |
| Squirrel | 5,201 | 216,933 | 5 | 2,089 | 0.22 |
| Actor | 7,600 | 33,544 | 5 | 931 | 0.22 |

**Roman-Empire** is constructed from the Roman Empire Wikipedia article, with nodes representing words and edges formed by either word adjacency or dependency relations. It contains 22.7K nodes and 32.9K edges. The task is to classify words by their syntactic roles, and node features are fast-Text embeddings. The graph is chain-like, with an average degree of 2.9 and a large diameter of 6824. Adjusted homophily is low ($h_{adj}$ = -0.05), making it useful for GNN evaluation under low homophily and sparse connectivity.

**Amazon-Ratings** is based on Amazon's product co-purchasing network, this dataset includes nodes as products (books, CDs, DVDs, etc.) and edges linking frequently co-purchased items. It consists of the largest connected component of the graph's 5-core. The goal is to predict product ratings grouped into five classes.

**Minesweeper** is a synthetic dataset resembling the Minesweeper game, nodes in a 100x100 grid represent cells, with edges connecting adjacent cells. The task is to identify mines (20% of nodes). Node features indicate neighboring mine counts, with 50% of features missing. The average degree is 7.88, and the graph has near-zero homophily due to random mine placement.

**Tolokers** is derived from the Toloka crowdsourcing platform, where nodes represent workers connected by shared tasks. The graph has 11.8K nodes and an average degree of 88.28. The task is to predict which workers have been banned, using profile and task performance features. The graph is much denser than others in the benchmark.

**Questions** is based on user interactions from Yandex Q, this dataset focuses on users interested in medicine. Nodes are users, and edges represent questions answered between users. It contains 48.9K nodes with an average degree of 6.28. The task is to predict user activity at the end of a one-year period, with fastText embeddings from user descriptions as features. The graph is highly imbalanced (97% active users).

**Texas**, **Wisconsin**, **Cornell** are part of the WebKB project, representing web pages from university computer science departments. Nodes correspond to web pages, and edges represent hyperlinks between them. The node features are bag-of-words vectors from the web page content, and the labels classify each page into one of five categories: student, project, course, staff, and faculty.

**Chameleon**, **Squirrel** are page-page networks based on specific topics from Wikipedia. Nodes represent web pages, and edges correspond to mutual links between them. Node features are derived from the page content, and the classification task is based on average monthly traffic. These datasets are characterized by high heterophily, making them challenging for traditional GNN models.

**Actor** is an induced subgraph from a film-director-actor-writer network. Nodes represent actors, and edges are created when two actors co-occur on the same Wikipedia page. The task is to classify actors into five categories based on the keywords associated with their Wikipedia pages.

## D  IMPLEMENTATION DETAILS

All the experiments are conducted on a linux server(Operation system: Ubuntu 16.04.7 LTS) with one NVIDIA Tesla V100 card.

### D.1  GNN+GSL

We implement GSL on $4$ baseline GNNs with a variety of GSL approaches from the perspective of GSL bases, GSL graph construction, and view fusion. The baseline GNNs include:

- **GCN** (Kipf & Welling, 2016) performs layer-wise propagation of node features and aggregates information from neighboring nodes to capture local graph structures. Each layer applies a convolution operation to update node embeddings, combining the node's features with its neighbors.
- **GAT** (Velicković et al., 2017) employs self-attention to learn dynamic attention coefficients between nodes and their neighbors. These coefficients are normalized using softmax, and the final node representation is computed as a weighted sum of the neighbor features. Multi-head attention is used to enhance stability and expressiveness, with the number of attention heads set to $8$ by default in our experiments.
- **SAGE** (Hamilton et al., 2017) uses an inductive framework to aggregate features from a node's local neighborhood, allowing it to generalize to unseen nodes. The aggregation function, set to mean in our experiments, efficiently combines neighbor information at each layer.
- **SGC** (Wu et al., 2019a) simplifies the GCN model by removing non-linear activations and collapsing multiple layers into a single linear transformation. This reduction in complexity accelerates training. Node features are propagated using precomputed matrices, making the model faster and more efficient. In our experiments, the number of k-hops in SGC is set to 2 by default.

The GSL bases $\boldsymbol{B}$ includes the following options:

- $\boldsymbol{B} = \boldsymbol{X}$: The original node features are used as the GSL bases.
- $\boldsymbol{B} = \hat{\boldsymbol{A}}\boldsymbol{X}$: Aggregated node features from 1-hop neighbors, normalized by node degree, are used as the GSL bases.
- $\boldsymbol{B} = \mathrm{MLP}(\boldsymbol{X})$: Pretrained MLP embeddings are used as the GSL bases. A 2-layer MLP is trained using node features and labels on the training set for 1000 epochs per run. The hidden layer size is set to 128, the learning rate to $1e^{-2}$, the dropout rate to 0.5, and the weight decay to $5e^{-4}$. All parameters are optimized with Adam. After training, node embeddings are extracted from the last hidden layer, with a dimension of 128, prior to classifier input.
- $\boldsymbol{B} = \mathrm{GCN}(\boldsymbol{X}, \boldsymbol{A})$: Pretrained node embeddings are obtained from a 2-layer GCN model, following the same training procedure as for the MLP embeddings.
- $\boldsymbol{B} = \mathrm{GCL}(\boldsymbol{X}, \boldsymbol{A})$: Pretrained node embeddings are derived from a Graph Contrastive Learning (GCL) model without supervision, following the same training process as the MLP embeddings. GRACE (Zhu et al., 2020b) is used as the GCL model, with 2 views and 2 layers. The edge and feature dropout rates in each view are set to 0.2.

The approaches for the construction of GSL graph $\mathcal{G}'$ includes:

- Cos-graph: $\mathcal{G}' = \{e_{ij}|\cos(\boldsymbol{B_i}, \boldsymbol{B_j}) > \delta, i \in \mathcal{V}, j \in \mathcal{V}\}$. This method calculates the cosine similarity between all node pairs in the original graph $\mathcal{G}$. Node pairs with a similarity higher than the threshold $\delta$ are selected as the edge set for the GSL graph $\mathcal{G}'$.
- Cos-node: $\mathcal{G}' = \bigcup_{i \in \mathcal{V}}\{\{e_i j\}|\cos(\boldsymbol{B_i}, \boldsymbol{B_j}) > \delta_i, j \in \mathcal{N}_i\}$. Unlike Cos-graph, which operates at the graph level, Cos-node constructs $\mathcal{G}'$ at the node level. To prevent nodes from being left without neighbors (which may occur in Cos-graph), Cos-node selects neighbors based on node-level cosine similarity, ensuring each node has sufficient connections.

- kNN: $\mathcal{G}' = \text{kNN}(\boldsymbol{B})$. This method constructs a kNN graph using the k-Nearest Neighbors algorithm based on the GSL bases $\boldsymbol{B}$.

The view fusion in GSL includes:

- $\{\mathcal{G}'\}$: This approach uses only the GSL graph $\mathcal{G}'$ for subsequent GNN training, completely ignoring the original graph $\mathcal{G}$.
- $\{\mathcal{G}, \mathcal{G}'\}, \theta_1 = \theta_2$. Both the GSL graph $\mathcal{G}'$ and the original graph $\mathcal{G}$ are used for GNN training, with parameter sharing across each layer of the GNN.
- $\{\mathcal{G}, \mathcal{G}'\}, \theta_1 \neq \theta_2$. Both the GSL graph $\mathcal{G}'$ and the original graph $\mathcal{G}$ are used for GNN training, but with separate model parameters for each graph.

Especially, for graphs with two views, the fusion stage in GSL includes:

- Early Fusion: $\mathcal{G} + \mathcal{G}'$. Combine the two graphs, $\mathcal{G}$ and $\mathcal{G}'$, into a single new graph prior to GNN training.
- Late Fusion: $\boldsymbol{H} + \boldsymbol{H}'$. After training the GNN on the original graph $\mathcal{G}$ and the GSL graph $\mathcal{G}'$, merge the node embeddings, $\mathbf{H}$ and $\mathbf{H}'$, before passing them to the classifiers.

In addition to the original models based on 4 baseline GNNs, we implement GNN+GSL (GSL-augmented GNNs) by combining the aforementioned GSL modules, resulting in multiple variants for each type of GNN. For all models, we explore hyperparameters including hidden dimensions from the set $\{64, 128, 256\}$, learning rates from $\{1e\text{-}2, 1e\text{-}3, 1e\text{-}4\}$, weight decay values from $\{0, 1e\text{-}5, 1e\text{-}3\}$, the number of layers from $\{2, 3\}$, and dropout rates from $\{0.2, 0.4, 0.6, 0.8\}$.

For GSL graph generation, we also search for additional hyperparameters to ensure the performance quality of the GSL-augmented GNN. Specifically, for Cos-graph and Cos-node, we control the parameter $\delta$ to vary the ratio of the number of edges in $\mathcal{G}'$ to the number of edges in $\mathcal{G}$ across the set $\{0.1, 0.5, 1, 5\}$. For kNN, we investigate the number of neighbors from the set $\{2, 3, 5, 10\}$..

## D.2 SOTA-GSL

To fairly re-evaluate the effectiveness of GSL in state-of-the-art (SOTA) models, two methods are employed to compare performance within the same search space. The first method (SOTA, $\mathcal{G}' = \mathcal{G}$) replaces the GSL graph with the original graph. The second method (SOTA, $\mathcal{G}' = \text{MLP}$) substitutes the GSL graph with a linear transformation, connecting it to the subsequent model structures and ensuring the continuity of channels within the original network structure. We train each model for 1000 epochs and search the hidden dimensions from the set $\{16, 32, 64, 128, 256, 512\}$, learning rate from $\{1e\text{-}1, 1e\text{-}2, 1e\text{-}3, 1e\text{-}4, 1e\text{-}5\}$, weight decay values from $\{5e\text{-}4, 5e\text{-}5, 5e\text{-}6, 5e\text{-}7, 0\}$, the number of layers from $\{1, 2, 3\}$, and dropout rates from $\{0.2, 0.4, 0.6, 0.8\}$. The model-specific hyperparameters are shown as follows:

In **GRCN**, the hyperparameter K determines the number of nearest neighbors used to create a sparse graph from a dense similarity graph which helps balance efficiency and accuracy. We set the k as 5.

In **GAug**, the alpha is a hyperparameter that regulates the influence of the edge predictor on the original graph. We set the alpha as 0.1.

In **IDGL**, The parameter graph_learn_num_pers defines the number of perspectives for evaluating node similarities in the graph learning process. The parameter num_anchors specifies the number of anchor points used to reduce computational complexity and improve scalability in graph structure learning. The graph_skip_conn parameter controls the proportion of skip connections, preserving information from the original graph during new graph structure learning. The update_adj_ratio parameter determines the proportion of the adjacency matrix updated at each iteration, influencing the dynamic adjustment of the graph structure. We set the graph_learn_num_pers as 6, num_anchors as 500, graph_skip_conn as 0.7, and update_adj_ratio as 0.3.

In **NodeFormer**, The parameter k determines the number of neighbors considered for each node in constructing the local graph structure, influencing the strength of node connections and the propagation of features. The parameter tolerance controls the degree of error tolerance during optimization. A larger tolerance allows more flexibility in the search space near local optima, while a smaller one

results in stricter convergence. The number of attention heads in a graph attention network (GAT). Multi-head attention enables the model to focus on different subspace representations simultaneously, enhancing the diversity and stability of the representations. We set the k as 10, lambda as 0.01, and n_heads as 4.

In **GEN**,the parameter K in KNN refers to the number of nearest neighbors used to construct the graph structure, determining how many adjacent nodes are selected. The parameter tolerance defines the acceptable range of error during optimization, controlling the convergence criteria of the model. The parameter threshold determines the edge weight threshold in the graph, deciding which edges to retain in the graph structure.We set the k as 10, tolerance as 0.01, and threshold as 0.5.

In **GloGNN**, we set the Delta as 0.9, Gamma as 0.8, alpha as 0.5, beta as 2000, and orders as 5. Delta adjusts the balance between local and global node embeddings. Gamma controls the significance of global aggregation versus local information. Alpha balances the contributions of node features and graph structure. Beta regularizes the model, preventing overfitting. Order defines how many layers of neighbors are considered.

In **WRGAT**, we set the number of attention heads as 2 and the negative slope as 0.2. The number of attention heads determines how many attention mechanisms are used. The negative slope modifies the LeakyReLU activation.

The tables below show the best combination of hyperparameters based on the accuracy of test set.

Table 5: Hyperparameters for SOTA-GSL on Cora.

| Dataset | Model | Learning Rate | Weight Decay | Dropout | Hidden Dim | Num of Layers |
|---|---|---|---|---|---|---|
| | GAug | 1e-4 | 5e-7 | 0.8 | 512 | 2 |
| | GAug, $\mathcal{G}' = \mathcal{G}$ | 1e-4 | 5e-7 | 0.8 | 512 | 2 |
| | GAug, $\mathcal{G}' = $ MLP | 1e-4 | 5e-7 | 0.8 | 512 | 2 |
| | GEN | 1e-2 | 5e-4 | 0.5 | 16 | 2 |
| | GEN, $\mathcal{G}' = \mathcal{G}$ | 1e-2 | 5e-4 | 0.5 | 16 | 2 |
| | GEN, $\mathcal{G}' = $ MLP | 1e-2 | 5e-4 | 0.5 | 16 | 2 |
| | GRCN | 1e-3 | 5e-3 | 0.5 | 256 | 2 |
| | GRCN, $\mathcal{G}' = \mathcal{G}$ | 1e-3 | 5e-3 | 0.5 | 256 | 2 |
| | GRCN, $\mathcal{G}' = $ MLP | 1e-3 | 5e-3 | 0.5 | 256 | 2 |
| | IDGL | 1e-2 | 5e-4 | 0.5 | 512 | 2 |
| | IDGL, $\mathcal{G}' = \mathcal{G}$ | 1e-2 | 5e-4 | 0.5 | 512 | 2 |
| | IDGL, $\mathcal{G}' = $ MLP | 1e-2 | 5e-4 | 0.5 | 512 | 2 |
| Cora | NodeFormer | 1e-2 | 5e-4 | 0.2 | 64 | 2 |
| | NodeFormer, $\mathcal{G}' = \mathcal{G}$ | 1e-2 | 5e-4 | 0.2 | 64 | 2 |
| | NodeFormer, $\mathcal{G}' = $ MLP | 1e-2 | 5e-4 | 0.2 | 64 | 2 |
| | GloGNN | 1e-2 | 5e-5 | 0.5 | 64 | 1 |
| | GloGNN, $\mathcal{G}' = \mathcal{G}$ | 1e-2 | 5e-5 | 0.5 | 64 | 1 |
| | GloGNN, $\mathcal{G}' = $ MLP | 1e-2 | 5e-5 | 0.5 | 64 | 1 |
| | WRGAT | 1e-2 | 1e-5 | 0.5 | 128 | 2 |
| | WRGAT, $\mathcal{G}' = \mathcal{G}$ | 1e-2 | 5e-5 | 0.5 | 128 | 2 |
| | WRGAT, $\mathcal{G}' = $ MLP | 1e-2 | 1e-5 | 0.5 | 128 | 2 |
| | WRGCN | 1e-2 | 1e-5 | 0.5 | 128 | 2 |
| | WRGCN, $\mathcal{G}' = \mathcal{G}$ | 1e-2 | 5e-5 | 0.5 | 128 | 2 |
| | WRGCN, $\mathcal{G}' = $ MLP | 1e-2 | 1e-5 | 0.5 | 128 | 2 |

# E  ADDITIONAL EXPERIMENT RESULTS

In this section, we examine the impact of different GSL modules on GNN models. The GSL modules include graph bases, GSL graph generation, view fusion methods, and fusion stages, with details provided in Appendix D.1.

## E.1  GSL BASES

In addition to the analysis of the impact of GSL bases shown in Figure 5, Figure 6 presents further results on the performance of various GSL bases ($\mathbf{X}$, $\hat{\mathbf{X}}$, $MLP(\mathbf{X})$, $GCN(\mathbf{X}, \mathbf{A})$, $GCL(\mathbf{X}, \mathbf{A})$) across GAT, SGC, and GraphSAGE. The results are consistent with those observed in GCN and MLP, where the original node features do not always yield the best input. Some pretrained features,

Table 6: Hyperparameters for SOTA-GSL on PubMed.

| Dataset | Model | Learning Rate | Weight Decay | Dropout | Hidden Dim | Num of Layers |
|---------|-------|---------------|--------------|---------|------------|---------------|
| | GAug | 1e-2 | 5e-4 | 0.5 | 128 | 2 |
| | GAug, $\mathcal{G}' = \mathcal{G}$ | 1e-2 | 5e-4 | 0.5 | 128 | 2 |
| | GAug, $\mathcal{G}' = $ MLP | 1e-2 | 5e-4 | 0.5 | 128 | 2 |
| | GEN | 1e-3 | 5e-4 | 0.2 | 32 | 2 |
| | GEN, $\mathcal{G}' = \mathcal{G}$ | 1e-3 | 5e-4 | 0.2 | 32 | 2 |
| | GEN, $\mathcal{G}' = $ MLP | 1e-3 | 5e-4 | 0.2 | 32 | 2 |
| | GRCN | 1e-3 | 5e-3 | 0.5 | 32 | 2 |
| | GRCN, $\mathcal{G}' = \mathcal{G}$ | 1e-3 | 5e-3 | 0.5 | 32 | 2 |
| | GRCN, $\mathcal{G}' = $ MLP | 1e-3 | 5e-3 | 0.5 | 32 | 2 |
| | IDGL | 1e-2 | 5e-4 | 0.5 | 16 | 2 |
| | IDGL, $\mathcal{G}' = \mathcal{G}$ | 1e-2 | 5e-4 | 0.5 | 16 | 2 |
| | IDGL, $\mathcal{G}' = $ MLP | 1e-2 | 5e-4 | 0.5 | 16 | 2 |
| PubMed | NodeFormer | 1e-3 | 5e-4 | 0.2 | 64 | 2 |
| | NodeFormer, $\mathcal{G}' = \mathcal{G}$ | 1e-3 | 5e-4 | 0.2 | 64 | 2 |
| | NodeFormer, $\mathcal{G}' = $ MLP | 1e-3 | 5e-4 | 0.2 | 32 | 2 |
| | GloGNN | 1e-3 | 5e-5 | 0.7 | 64 | 3 |
| | GloGNN, $\mathcal{G}' = \mathcal{G}$ | 1e-3 | 5e-5 | 0.7 | 64 | 3 |
| | GloGNN, $\mathcal{G}' = $ MLP | 1e-3 | 5e-5 | 0.7 | 64 | 3 |
| | WRGAT | 1e-2 | 5e-5 | 0.5 | 64 | 2 |
| | WRGAT, $\mathcal{G}' = \mathcal{G}$ | 1e-2 | 1e-5 | 0.5 | 64 | 2 |
| | WRGAT, $\mathcal{G}' = $ MLP | 1e-2 | 5e-5 | 0.5 | 64 | 2 |
| | WRGCN | 1e-2 | 5e-5 | 0.5 | 64 | 2 |
| | WRGCN, $\mathcal{G}' = \mathcal{G}$ | 1e-2 | 5e-5 | 0.5 | 64 | 2 |
| | WRGCN, $\mathcal{G}' = $ MLP | 1e-2 | 5e-5 | 0.5 | 64 | 2 |

Table 7: Hyperparameters for SOTA-GSL on Citeseer.

| Dataset | Model | Learning Rate | Weight Decay | Dropout | Hidden Dim | Num of Layers |
|---------|-------|---------------|--------------|---------|------------|---------------|
| | GAug | 1e-4 | 5e-7 | 0.8 | 512 | 2 |
| | GAug, $\mathcal{G}' = \mathcal{G}$ | 1e-4 | 5e-7 | 0.8 | 512 | 2 |
| | GAug, $\mathcal{G}' = $ MLP | 1e-4 | 5e-7 | 0.8 | 512 | 2 |
| | GEN | 1e-2 | 5e-4 | 0.5 | 16 | 2 |
| | GEN, $\mathcal{G}' = \mathcal{G}$ | 1e-2 | 5e-4 | 0.5 | 16 | 2 |
| | GEN, $\mathcal{G}' = $ MLP | 1e-2 | 5e-4 | 0.5 | 16 | 2 |
| | GRCN | 1e-3 | 5e-3 | 0.8 | 512 | 3 |
| | GRCN, $\mathcal{G}' = \mathcal{G}$ | 1e-3 | 5e-3 | 0.8 | 512 | 3 |
| | GRCN, $\mathcal{G}' = $ MLP | 1e-2 | 5e-3 | 0.5 | 256 | 3 |
| | IDGL | 1e-2 | 5e-4 | 0.5 | 32 | 2 |
| | IDGL, $\mathcal{G}' = \mathcal{G}$ | 1e-3 | 5e-4 | 0.5 | 16 | 2 |
| | IDGL, $\mathcal{G}' = $ MLP | 1e-3 | 5e-4 | 0.5 | 16 | 2 |
| Citeseer | NodeFormer | 1e-2 | 5e-4 | 0.2 | 64 | 2 |
| | NodeFormer, $\mathcal{G}' = \mathcal{G}$ | 1e-2 | 5e-4 | 0.2 | 64 | 2 |
| | NodeFormer, $\mathcal{G}' = $ MLP | 1e-2 | 5e-4 | 0.2 | 64 | 2 |
| | GloGNN | 1e-2 | 1e-5 | 0.7 | 64 | 2 |
| | GloGNN, $\mathcal{G}' = \mathcal{G}$ | 1e-2 | 1e-5 | 0.7 | 64 | 2 |
| | GloGNN, $\mathcal{G}' = $ MLP | 1e-2 | 1e-5 | 0.7 | 64 | 2 |
| | WRGAT | 1e-2 | 5e-5 | 0.5 | 128 | 2 |
| | WRGAT, $\mathcal{G}' = \mathcal{G}$ | 1e-2 | 5e-5 | 0.5 | 128 | 2 |
| | WRGAT, $\mathcal{G}' = $ MLP | 1e-2 | 5e-5 | 0.5 | 128 | 2 |
| | WRGCN | 1e-2 | 5e-5 | 0.3 | 128 | 2 |
| | WRGCN, $\mathcal{G}' = \mathcal{G}$ | 1e-2 | 5e-5 | 0.5 | 128 | 2 |
| | WRGCN, $\mathcal{G}' = $ MLP | 1e-2 | 1e-5 | 0.5 | 128 | 1 |

Table 8: Hyperparameters for SOTA-GSL on Minesweeper.

| Dataset | Model | Learning Rate | Weight Decay | Dropout | Hidden Dim | Num of Layers |
|---|---|---|---|---|---|---|
| Minesweeper | GAug | 1e-3 | 5e-6 | 0.8 | 256 | 3 |
| | GAug, $\mathcal{G}' = \mathcal{G}$ | 1e-3 | 5e-6 | 0.8 | 256 | 3 |
| | GAug, $\mathcal{G}' = $ MLP | 1e-3 | 5e-6 | 0.8 | 256 | 3 |
| | GEN | 1e-4 | 5e-6 | 0.8 | 256 | 3 |
| | GEN, $\mathcal{G}' = \mathcal{G}$ | 1e-4 | 5e-6 | 0.8 | 256 | 3 |
| | GEN, $\mathcal{G}' = $ MLP | 1e-4 | 5e-6 | 0.8 | 256 | 3 |
| | GRCN | 1e-3 | 5e-7 | 0.2 | 128 | 2 |
| | GRCN, $\mathcal{G}' = \mathcal{G}$ | 1e-3 | 5e-6 | 0.2 | 128 | 2 |
| | GRCN, $\mathcal{G}' = $ MLP | 1e-3 | 5e-6 | 0.2 | 128 | 2 |
| | IDGL | 1e-1 | 5e-6 | 0.2 | 128 | 3 |
| | IDGL, $\mathcal{G}' = \mathcal{G}$ | 1e-1 | 5e-6 | 0.2 | 128 | 3 |
| | IDGL, $\mathcal{G}' = $ MLP | 1e-1 | 5e-6 | 0.2 | 128 | 3 |
| | NodeFormer | 1e-2 | 5e-4 | 0.8 | 32 | 2 |
| | NodeFormer, $\mathcal{G}' = \mathcal{G}$ | 1e-2 | 5e-4 | 0.8 | 32 | 2 |
| | NodeFormer, $\mathcal{G}' = $ MLP | 1e-2 | 5e-4 | 0.8 | 32 | 2 |
| | GloGNN | 1e-2 | 5e-4 | 0.5 | 512 | 5 |
| | GloGNN, $\mathcal{G}' = \mathcal{G}$ | 1e-2 | 5e-4 | 0.5 | 512 | 5 |
| | GloGNN, $\mathcal{G}' = $ MLP | 1e-2 | 5e-4 | 0.5 | 512 | 5 |
| | WRGAT | 1e-2 | 5e-5 | 0.5 | 128 | 2 |
| | WRGAT, $\mathcal{G}' = \mathcal{G}$ | 1e-2 | 5e-5 | 0.5 | 128 | 2 |
| | WRGAT, $\mathcal{G}' = $ MLP | 1e-2 | 5e-5 | 0.5 | 128 | 2 |
| | WRGCN | 1e-2 | 5e-5 | 0.5 | 128 | 2 |
| | WRGCN, $\mathcal{G}' = \mathcal{G}$ | 1e-2 | 5e-5 | 0.5 | 128 | 2 |
| | WRGCN, $\mathcal{G}' = $ MLP | 1e-2 | 5e-5 | 0.5 | 128 | 2 |

Table 9: Hyperparameters for SOTA-GSL on Roman-Empire.

| Dataset | Model | Learning Rate | Weight Decay | Dropout | Hidden Dim | Num of Layers |
|---|---|---|---|---|---|---|
| Roman-empire | GAug | 1e-1 | 5e-5 | 0.5 | 32 | 2 |
| | GAug, $\mathcal{G}' = \mathcal{G}$ | 1e-1 | 5e-5 | 0.5 | 32 | 2 |
| | GAug, $\mathcal{G}' = $ MLP | 1e-1 | 5e-5 | 0.5 | 32 | 2 |
| | GEN | 1e-2 | 5e-7 | 0.2 | 128 | 2 |
| | GEN, $\mathcal{G}' = \mathcal{G}$ | 1e-2 | 5e-7 | 0.2 | 128 | 2 |
| | GEN, $\mathcal{G}' = $ MLP | 1e-2 | 5e-7 | 0.2 | 128 | 2 |
| | GRCN | 1e-3 | 5e-5 | 0.5 | 128 | 2 |
| | GRCN, $\mathcal{G}' = \mathcal{G}$ | 1e-2 | 5e-5 | 0.5 | 128 | 2 |
| | GRCN, $\mathcal{G}' = $ MLP | 1e-2 | 5e-5 | 0.5 | 128 | 2 |
| | IDGL | 1e-1 | 5e-5 | 0.5 | 128 | 2 |
| | IDGL, $\mathcal{G}' = \mathcal{G}$ | 1e-1 | 5e-5 | 0.5 | 128 | 2 |
| | IDGL, $\mathcal{G}' = $ MLP | 1e-1 | 5e-5 | 0.5 | 128 | 2 |
| | NodeFormer | 1e-3 | 5e-6 | 0.2 | 128 | 3 |
| | NodeFormer, $\mathcal{G}' = \mathcal{G}$ | 1e-3 | 5e-6 | 0.2 | 128 | 3 |
| | NodeFormer, $\mathcal{G}' = $ MLP | 1e-3 | 5e-5 | 0.8 | 128 | 3 |
| | GloGNN | 1e-2 | 5e-5 | 0.7 | 128 | 3 |
| | GloGNN, $\mathcal{G}' = \mathcal{G}$ | 1e-2 | 5e-5 | 0.7 | 128 | 3 |
| | GloGNN, $\mathcal{G}' = $ MLP | 1e-2 | 5e-5 | 0.7 | 128 | 3 |
| | WRGAT | 1e-2 | 5e-5 | 0.5 | 128 | 2 |
| | WRGAT, $\mathcal{G}' = \mathcal{G}$ | 1e-2 | 1e-5 | 0.5 | 128 | 2 |
| | WRGAT, $\mathcal{G}' = $ MLP | 1e-2 | 5e-5 | 0.5 | 128 | 2 |
| | WRGCN | 1e-2 | 5e-5 | 0.5 | 128 | 2 |
| | WRGCN, $\mathcal{G}' = \mathcal{G}$ | 1e-2 | 5e-5 | 0.5 | 128 | 2 |
| | WRGCN, $\mathcal{G}' = $ MLP | 1e-2 | 5e-5 | 0.5 | 128 | 2 |

Table 10: Hyperparameters for SOTA-GSL on Amazon-ratings.

| Dataset | Model | Learning Rate | Weight Decay | Dropout | Hidden Dim | Num of Layers |
|---|---|---|---|---|---|---|
| | GAug | 1e-2 | 5e-7 | 0.2 | 128 | 2 |
| | GAug, $\mathcal{G}' = \mathcal{G}$ | 1e-2 | 5e-7 | 0.2 | 128 | 2 |
| | GAug, $\mathcal{G}' = $ MLP | 1e-2 | 5e-7 | 0.2 | 128 | 2 |
| | GEN | 1e-2 | 5e-7 | 0.2 | 128 | 2 |
| | GEN, $\mathcal{G}' = \mathcal{G}$ | 1e-2 | 5e-7 | 0.2 | 128 | 2 |
| | GEN, $\mathcal{G}' = $ MLP | 1e-2 | 5e-7 | 0.2 | 128 | 2 |
| | GRCN | 1e-3 | 5e-7 | 0.2 | 128 | 2 |
| | GRCN, $\mathcal{G}' = \mathcal{G}$ | 1e-2 | 5e-7 | 0.2 | 64 | 2 |
| | GRCN, $\mathcal{G}' = $ MLP | 1e-2 | 5e-7 | 0.2 | 128 | 2 |
| | IDGL | 1e-2 | 5e-7 | 0.2 | 128 | 2 |
| | IDGL, $\mathcal{G}' = \mathcal{G}$ | 1e-2 | 5e-7 | 0.2 | 128 | 2 |
| | IDGL, $\mathcal{G}' = $ MLP | 1e-2 | 5e-7 | 0.2 | 128 | 2 |
| Amazon-ratings | NodeFormer | 1e-4 | 5e-5 | 0.5 | 128 | 3 |
| | NodeFormer, $\mathcal{G}' = \mathcal{G}$ | 1e-4 | 5e-5 | 0.5 | 64 | 3 |
| | NodeFormer, $\mathcal{G}' = $ MLP | 1e-4 | 5e-5 | 0.5 | 64 | 3 |
| | GloGNN | 1e-2 | 5e-5 | 0.3 | 128 | 3 |
| | GloGNN, $\mathcal{G}' = \mathcal{G}$ | 1e-2 | 5e-5 | 0.3 | 128 | 3 |
| | GloGNN, $\mathcal{G}' = $ MLP | 1e-2 | 5e-5 | 0.3 | 128 | 3 |
| | WRGAT | 1e-2 | 5e-5 | 0.3 | 128 | 2 |
| | WRGAT, $\mathcal{G}' = \mathcal{G}$ | 1e-2 | 1e-5 | 0.3 | 128 | 2 |
| | WRGAT, $\mathcal{G}' = $ MLP | 1e-2 | 1e-5 | 0.3 | 128 | 2 |
| | WRGCN | 1e-2 | 5e-5 | 0.7 | 128 | 3 |
| | WRGCN, $\mathcal{G}' = \mathcal{G}$ | 1e-2 | 5e-5 | 0.7 | 128 | 3 |
| | WRGCN, $\mathcal{G}' = $ MLP | 1e-2 | 1e-5 | 0.7 | 128 | 3 |

Table 11: Hyperparameters for SOTA-GSL on Questions.

| Dataset | Model | Learning Rate | Weight Decay | Dropout | Hidden Dim | Num of Layers |
|---|---|---|---|---|---|---|
| | GAug | 1e-2 | 5e-4 | 0.5 | 64 | 3 |
| | GAug, $\mathcal{G}' = \mathcal{G}$ | 1e-2 | 5e-4 | 0.5 | 64 | 3 |
| | GAug, $\mathcal{G}' = $ MLP | 1e-2 | 5e-4 | 0.5 | 64 | 3 |
| | GEN | 1e-2 | 5e-7 | 0.2 | 256 | 2 |
| | GEN, $\mathcal{G}' = \mathcal{G}$ | 1e-2 | 5e-7 | 0.2 | 256 | 2 |
| | GEN, $\mathcal{G}' = $ MLP | 1e-2 | 5e-7 | 0.2 | 256 | 2 |
| | GRCN | 1e-2 | 5e-6 | 0.5 | 64 | 2 |
| | GRCN, $\mathcal{G}' = \mathcal{G}$ | 1e-2 | 5e-6 | 0.5 | 64 | 2 |
| | GRCN, $\mathcal{G}' = $ MLP | 1e-2 | 5e-6 | 0.5 | 64 | 2 |
| | IDGL | 1e-2 | 5e-7 | 0.2 | 128 | 2 |
| | IDGL, $\mathcal{G}' = \mathcal{G}$ | 1e-2 | 5e-7 | 0.2 | 128 | 2 |
| Questions | IDGL, $\mathcal{G}' = $ MLP | 1e-2 | 5e-7 | 0.2 | 128 | 2 |
| | NodeFormer | 1e-4 | 5e-3 | 0.5 | 128 | 3 |
| | NodeFormer, $\mathcal{G}' = \mathcal{G}$ | 1e-4 | 5e-3 | 0.5 | 64 | 3 |
| | NodeFormer, $\mathcal{G}' = $ MLP | 1e-4 | 5e-3 | 0.5 | 64 | 3 |
| | GloGNN | 1e-2 | 5e-5 | 0.7 | 128 | 3 |
| | GloGNN, $\mathcal{G}' = \mathcal{G}$ | 1e-2 | 5e-5 | 0.7 | 128 | 3 |
| | GloGNN, $\mathcal{G}' = $ MLP | 1e-2 | 5e-5 | 0.7 | 128 | 3 |
| | WRGAT | 5e-3 | 5e-5 | 0.3 | 64 | 2 |
| | WRGAT, $\mathcal{G}' = \mathcal{G}$ | 5e-3 | 1e-5 | 0.3 | 64 | 2 |
| | WRGAT, $\mathcal{G}' = $ MLP | 5e-3 | 5e-5 | 0.3 | 64 | 2 |
| | WRGCN | 5e-3 | 5e-5 | 0.7 | 64 | 2 |
| | WRGCN, $\mathcal{G}' = \mathcal{G}$ | 5e-3 | 5e-5 | 0.7 | 64 | 2 |
| | WRGCN, $\mathcal{G}' = $ MLP | 5e-3 | 1e-5 | 0.7 | 64 | 2 |

Table 12: Hyperparameters for SOTA-GSL on Tolokers.

| Dataset | Model | Learning Rate | Weight Decay | Dropout | Hidden Dim | Num of Layers |
|---------|-------|---------------|--------------|---------|------------|---------------|
| | GAug | 1e-1 | 5e-5 | 0.5 | 32 | 2 |
| | GAug, $\mathcal{G}' = \mathcal{G}$ | 1e-1 | 5e-5 | 0.5 | 32 | 2 |
| | GAug, $\mathcal{G}' = $ MLP | 1e-1 | 5e-5 | 0.5 | 32 | 2 |
| | GEN | 1e-2 | 5e-5 | 0.2 | 128 | 2 |
| | GEN, $\mathcal{G}' = \mathcal{G}$ | 1e-2 | 5e-6 | 0.2 | 128 | 2 |
| | GEN, $\mathcal{G}' = $ MLP | 1e-2 | 5e-6 | 0.2 | 128 | 2 |
| | GRCN | 1e-2 | 5e-5 | 0.5 | 32 | 2 |
| | GRCN, $\mathcal{G}' = \mathcal{G}$ | 1e-2 | 5e-6 | 0.5 | 32 | 2 |
| | GRCN, $\mathcal{G}' = $ MLP | 1e-1 | 5e-6 | 0.5 | 64 | 2 |
| | IDGL | 1e-2 | 5e-4 | 0.5 | 64 | 2 |
| | IDGL, $\mathcal{G}' = \mathcal{G}$ | 1e-2 | 5e-4 | 0.5 | 64 | 2 |
| | IDGL, $\mathcal{G}' = $ MLP | 1e-2 | 5e-4 | 0.5 | 64 | 2 |
| Tolokers | NodeFormer | 1e-2 | 5e-4 | 0.2 | 64 | 2 |
| | NodeFormer, $\mathcal{G}' = \mathcal{G}$ | 1e-2 | 5e-4 | 0.2 | 64 | 2 |
| | NodeFormer, $\mathcal{G}' = $ MLP | 1e-2 | 5e-4 | 0.2 | 64 | 2 |
| | GloGNN | 1e-2 | 5e-5 | 0.3 | 128 | 3 |
| | GloGNN, $\mathcal{G}' = \mathcal{G}$ | 1e-2 | 5e-5 | 0.3 | 128 | 3 |
| | GloGNN, $\mathcal{G}' = $ MLP | 1e-2 | 5e-5 | 0.3 | 128 | 3 |
| | WRGAT | 1e-2 | 5e-5 | 0.5 | 128 | 2 |
| | WRGAT, $\mathcal{G}' = \mathcal{G}$ | 1e-2 | 1e-5 | 0.5 | 128 | 2 |
| | WRGAT, $\mathcal{G}' = $ MLP | 1e-2 | 5e-5 | 0.5 | 128 | 2 |
| | WRGCN | 1e-2 | 5e-5 | 0.5 | 128 | 1 |
| | WRGCN, $\mathcal{G}' = \mathcal{G}$ | 1e-2 | 5e-5 | 0.5 | 128 | 2 |
| | WRGCN, $\mathcal{G}' = $ MLP | 1e-2 | 5e-5 | 0.5 | 128 | 2 |

such as $MLP(\mathbf{X})$ on the Texas, Cornell, and Wisconsin datasets, demonstrate significant improvement compared to the original features $\mathbf{X}$, highlighting the necessity of self-training. Since many GSL methods (Zheng et al., 2024b; Suresh et al., 2021) utilize self-training during the training process, a fair comparison of these GSL methods and baseline GNNs should be conducted in the context of self-training, such as by using pretrained node features as input, as shown in Table 1.

### E.2 GSL GRAPH GENERATION

Figure 7 compares the Cos-graph, Cos-node, and kNN methods for GSL graph generation. Across most datasets, the performance differences among these methods are minimal. In certain datasets, such as Roman-empire and Pubmed, the models exhibit comparable performance regardless of the graph generation technique employed. This suggests that variations in graph generation have a limited effect on overall performance.

### E.3 VIEW FUSION

Figure 8 illustrates the impact of different view fusion approaches, comparing the use of only the GSL graph $\mathcal{G}'$, the combination of the original graph $\mathcal{G}$ with $\mathcal{G}'$ using shared parameters $\theta_1 = \theta_2$, and the use of separate parameters $\theta_1 \neq \theta_2$. Notably, using only the GSL graph $\mathcal{G}'$ underperforms compared to employing both graph views with separate model parameters. This indicates that incorporating information from the original graph $\mathcal{G}$ is beneficial for maximizing GNN+GSL performance. Furthermore, for the two graph views, parameter sharing significantly underperforms parameter separation. We speculate that the messages aggregated under $\mathcal{G}$ and $\mathcal{G}'$ differ considerably, suggesting that different graphs should be treated with distinct model parameters.

### E.4 FUSION STAGE

Figure 9 compares early fusion and late fusion for GNN+GSL with multiple graph views. The performance difference between the two fusion states is often minimal. While early fusion tends to perform slightly better on complex datasets like Actor and Pubmed, the overall impact of switching between early and late fusion is limited across most datasets. For simpler datasets like Minesweeper and Amazon, both fusion methods yield nearly identical performance, indicating that the choice of fusion state does not drastically alter the model's outcome in most cases.

### E.5 HETEROPHILY-ORIENTED GNN WITH GSL

Table 13: Performance of heterophily-oriented GNNs with GNN+GSL

| Model | Construct | Fusion | Param Sharing | Mines. | Roman. | Amazon. | Tolokers | Questions | Squirrel | Chameleon | Actor | Texas | Cornell | Wisconsin | Cora | CiteSeer | PubMed | Rank |
|---|---|---|---|---|---|---|---|---|---|---|---|---|---|---|---|---|---|---|
| MLP | - | - | - | 79.55±1.23 | 65.45±0.99 | 46.65±0.83 | 75.94±1.38 | 74.92±1.39 | 39.29±2.22 | 43.57±4.18 | 35.40±1.38 | 80.46±6.44 | 73.78±7.34 | 85.88±7.78 | 87.97±1.80 | 76.68±2.10 | 87.39±2.18 | 2.93 |
| ACMGNN | - | - | - | 90.56±1.03 | 84.86±0.73 | 52.07±1.72 | 84.41±1.12 | 77.72±1.59 | 41.53±2.43 | 44.65±4.43 | 34.86±1.22 | 82.62±5.97 | 75.68±8.99 | 87.65±7.15 | 88.23±1.81 | 76.63±2.34 | 89.37±0.56 | 1.21 |
| ACMGNN | cos-graph | $\{\mathcal{G}'\}$ | - | 47.36±3.47 | 60.97±0.76 | 41.50±0.75 | 70.21±1.51 | 67.32±1.37 | 38.12±1.92 | 39.90±3.64 | 33.43±0.95 | 79.80±6.99 | 59.46±8.35 | 71.57±6.68 | 77.47±2.41 | 73.68±0.97 | 87.19±0.38 | 7.64 |
| ACMGNN | cos-graph | $\{\mathcal{G},\mathcal{G}'\}$ | $\theta_1 = \theta_2$ | 52.74±5.22 | 51.18±2.12 | 33.11±1.38 | 69.06±4.65 | 62.30±3.23 | 31.58±4.39 | 38.79±4.73 | 29.06±2.60 | 54.10±7.59 | 59.19±8.87 | 70.39±9.58 | 59.74±1.87 | 65.17±1.94 | 79.53±1.69 | 9.86 |
| ACMGNN | cos-graph | $\{\mathcal{G},\mathcal{G}'\}$ | $\theta_1 \neq \theta_2$ | 87.46±1.02 | 74.63±0.76 | 49.35±0.58 | 81.63±0.87 | 73.84±1.41 | 38.54±1.89 | 41.16±4.18 | 34.23±0.98 | 67.67±5.97 | 70.00±5.90 | 80.78±5.21 | 80.83±1.84 | 73.43±1.47 | 88.98±0.47 | 3.64 |
| ACMGNN | cos-node | $\{\mathcal{G}'\}$ | - | 52.83±3.52 | 61.26±0.62 | 42.47±0.53 | 74.14±1.14 | 72.23±1.36 | 38.23±1.97 | 40.77±3.68 | 34.74±0.90 | 61.45±6.13 | 63.51±5.87 | 74.31±6.43 | 75.84±2.93 | 73.05±1.18 | 87.22±0.41 | 6.21 |
| ACMGNN | cos-node | $\{\mathcal{G},\mathcal{G}'\}$ | $\theta_1 = \theta_2$ | 52.74±5.22 | 51.18±2.12 | 33.11±1.38 | 69.06±4.65 | 62.30±3.23 | 31.58±4.39 | 38.79±4.73 | 29.06±2.60 | 54.10±7.59 | 59.19±8.87 | 70.39±9.58 | 59.74±1.87 | 65.17±1.94 | 79.53±1.69 | 9.86 |
| ACMGNN | cos-node | $\{\mathcal{G},\mathcal{G}'\}$ | $\theta_1 \neq \theta_2$ | 87.80±0.97 | 73.55±0.51 | 49.04±0.57 | 80.74±0.92 | 74.11±1.40 | 39.19±2.12 | 40.28±4.30 | 34.19±1.16 | 69.86±5.56 | 69.46±7.21 | 80.39±5.23 | 80.33±1.90 | 73.31±1.26 | 88.94±0.36 | 4.07 |
| ACMGNN | kNN | $\{\mathcal{G}'\}$ | - | 51.68±3.38 | 60.86±0.87 | 41.68±0.95 | 71.31±0.64 | 69.56±1.41 | 38.58±1.96 | 40.56±2.34 | 34.88±0.77 | 62.51±6.16 | 62.70±5.95 | 76.47±4.43 | 75.99±2.85 | 70.20±1.51 | 87.20±0.45 | 6.64 |
| ACMGNN | kNN | $\{\mathcal{G},\mathcal{G}'\}$ | $\theta_1 = \theta_2$ | 52.74±5.22 | 51.18±2.12 | 33.11±1.38 | 69.06±4.65 | 62.30±3.23 | 31.58±4.39 | 38.79±4.73 | 29.06±2.60 | 54.10±7.59 | 59.19±8.87 | 70.39±9.58 | 59.74±1.87 | 65.17±1.94 | 79.53±1.69 | 9.86 |
| ACMGNN | kNN | $\{\mathcal{G},\mathcal{G}'\}$ | $\theta_1 \neq \theta_2$ | 87.59±0.88 | 73.21±0.63 | 49.06±0.53 | 81.34±0.85 | 73.95±1.35 | 39.18±2.18 | 41.70±3.71 | 34.67±1.11 | 68.48±5.78 | 68.92±5.87 | 80.20±3.13 | 80.46±2.26 | 73.14±1.31 | 88.87±0.51 | 4.07 |
| MLP | - | - | - | 79.55±1.23 | 65.45±0.99 | 46.65±0.83 | 75.94±1.38 | 74.92±1.39 | 39.29±2.22 | 43.57±4.18 | 35.40±1.38 | 80.46±6.44 | 73.78±7.34 | 85.88±7.78 | 87.97±1.80 | 76.68±2.10 | 87.39±2.18 | 2.29 |
| MixHop | - | - | - | 90.10±5.59 | 81.70±0.89 | 50.95±0.71 | 84.56±1.19 | 77.66±1.24 | 41.22±2.66 | 43.11±4.73 | 33.59±1.23 | 72.54±8.98 | 62.43±9.54 | 75.88±8.27 | 87.76±1.94 | 76.51±1.93 | 89.42±0.81 | 1.86 |
| MixHop | cos-graph | $\{\mathcal{G}'\}$ | - | 64.75±4.59 | 51.83±0.53 | 41.47±2.00 | 68.78±1.94 | 71.45±1.38 | 37.75±2.41 | 37.79±2.10 | 31.77±1.75 | 55.72±6.39 | 60.27±5.85 | 70.20±4.60 | 84.42±1.35 | 74.20±0.83 | 88.74±0.29 | 8.21 |
| MixHop | cos-graph | $\{\mathcal{G},\mathcal{G}'\}$ | $\theta_1 = \theta_2$ | 54.22±10.75 | 63.50±0.86 | 44.21±1.36 | 74.22±2.21 | 70.64±1.32 | 37.16±1.34 | 39.06±3.08 | 32.24±1.33 | 58.16±9.18 | 66.22±5.59 | 73.73±7.80 | 65.14±2.62 | 68.66±1.24 | 86.63±0.51 | 7.54 |
| MixHop | cos-graph | $\{\mathcal{G},\mathcal{G}'\}$ | $\theta_1 \neq \theta_2$ | 84.71±1.19 | 55.41±1.63 | 43.37±0.75 | 74.41±1.33 | 69.63±2.03 | 37.64±2.19 | 38.71±4.36 | 31.73±1.77 | 61.13±7.96 | 61.35±7.10 | 75.29±6.00 | 85.42±1.21 | 74.57±1.34 | 88.16±0.46 | 6.50 |
| MixHop | cos-node | $\{\mathcal{G}'\}$ | - | 60.56±7.08 | 51.74±0.68 | 42.71±0.97 | 74.27±1.84 | 72.83±1.12 | 38.35±1.99 | 38.88±3.00 | 33.05±1.04 | 58.42±6.52 | 62.27±5.98 | 71.57±4.91 | 83.22±1.16 | 74.11±1.12 | 88.23±0.45 | 6.71 |
| MixHop | cos-node | $\{\mathcal{G},\mathcal{G}'\}$ | $\theta_1 = \theta_2$ | 54.22±10.75 | 63.50±0.86 | 44.21±1.36 | 74.22±2.21 | 70.64±1.32 | 37.16±1.34 | 39.06±3.08 | 32.24±1.33 | 58.16±9.18 | 66.22±5.59 | 73.73±7.80 | 65.14±2.62 | 68.66±1.24 | 86.63±0.51 | 7.54 |
| MixHop | cos-node | $\{\mathcal{G},\mathcal{G}'\}$ | $\theta_1 \neq \theta_2$ | 85.43±0.57 | 55.95±2.35 | 44.15±0.59 | 76.54±0.91 | 72.03±2.45 | 37.47±2.07 | 39.52±3.33 | 32.50±1.10 | 60.61±8.73 | 62.97±6.75 | 75.10±6.20 | 85.36±0.89 | 74.68±1.13 | 88.18±0.52 | 4.79 |
| MixHop | kNN | $\{\mathcal{G}'\}$ | - | 59.50±6.26 | 50.39±0.72 | 42.07±0.93 | 70.49±1.70 | 69.57±1.32 | 38.07±1.72 | 38.76±2.91 | 33.23±1.30 | 59.25±4.49 | 57.30±6.96 | 69.22±7.22 | 83.99±1.28 | 74.96±1.18 | 87.99±0.40 | 8.00 |
| MixHop | kNN | $\{\mathcal{G},\mathcal{G}'\}$ | $\theta_1 = \theta_2$ | 54.22±10.75 | 63.50±0.86 | 44.21±1.36 | 74.22±2.21 | 70.64±1.32 | 37.16±1.34 | 39.06±3.08 | 32.24±1.33 | 58.16±9.18 | 66.22±5.59 | 73.73±7.80 | 65.14±2.62 | 68.66±1.24 | 86.63±0.51 | 7.54 |
| MixHop | kNN | $\{\mathcal{G},\mathcal{G}'\}$ | $\theta_1 \neq \theta_2$ | 85.53±0.50 | 57.48±1.98 | 43.28±0.68 | 77.24±1.61 | 70.34±1.76 | 38.15±2.01 | 40.12±3.76 | 32.30±1.53 | 60.05±9.45 | 63.51±7.56 | 74.90±8.21 | 85.18±1.26 | 74.59±1.19 | 88.20±0.57 | 4.93 |

We also include heterophily-oriented GNNs, specifically ACMGNN (Luan et al., 2022a) and Mix-Hop (Abu-El-Haija et al., 2019), in our experiments that incorporate GSL into GNN baselines. These experiments follow the same setup as described in Table 1. The results, presented in Table 13, demonstrate that, under fair comparison conditions, both ACMGNN and MixHop outperform their GNN+GFS counterparts. This suggests that adding GSL to these heterophily-oriented GNNs may be unnecessary.

### E.6 TRAINABLE GSL

Table 14: Performance of GNNs with their counterparts of trainable GSL.

| Model | GSL Type | Mines. | Roman. | Amazon. | Tolokers | Questions | Cora | CiteSeer | PubMed | Rank |
|---|---|---|---|---|---|---|---|---|---|---|
| GCN | No GSL | **90.07±5.79** | **81.46±1.25** | 50.89±1.16 | **84.61±0.99** | **77.68±1.10** | **87.97±1.51** | **76.75±2.30** | **89.47±0.64** | 1.19 |
| | Trainable GSL | **90.07±0.58** | 78.76±0.46 | 50.89±0.65 | **84.61±0.65** | OOM | 84.92±1.51 | 74.89±1.13 | 88.66±0.45 | 2.31 |
| | Non-trainable GSL | 89.17±0.68 | 72.63±1.45 | 48.31±0.96 | 82.91±0.97 | 75.56±1.05 | 85.69±1.73 | 75.49±1.42 | 88.72±0.71 | 2.50 |
| SGC | No GSL | **83.45±4.47** | **78.04±0.69** | **51.38±0.68** | **84.88±1.13** | **77.39±1.23** | **88.10±1.89** | **77.52±2.20** | **89.39±0.62** | 1.19 |
| | Trainable GSL | **83.45±1.03** | 74.74±0.57 | **51.38±0.57** | **84.88±0.65** | OOM | 86.99±1.64 | 75.13±1.26 | 88.94±0.31 | 2.31 |
| | Non-trainable GSL | 79.03±3.76 | 67.84±1.87 | 47.93±0.94 | 78.09±1.84 | 75.46±1.43 | 87.47±1.86 | 76.36±1.27 | 89.37±0.41 | 2.50 |
| GraphSAGE | No GSL | 90.66±0.88 | **85.02±0.97** | **52.93±0.83** | **83.31±1.12** | **75.95±1.41** | **88.13±1.77** | **76.65±2.00** | 89.18±0.65 | 1.31 |
| | Trainable GSL | 90.66±0.58 | 82.54±0.60 | **52.93±0.59** | **83.31±0.50** | OOM | 83.48±1.69 | 74.18±1.02 | 88.67±0.39 | 2.44 |
| | Non-trainable GSL | **90.67±0.66** | 79.02±1.21 | 52.10±0.84 | 82.17±0.89 | 75.38±0.96 | 83.60±1.78 | 74.39±1.35 | **88.88±0.50** | 2.25 |
| GAT | No GSL | **90.41±1.34** | **84.51±0.84** | 52.00±2.84 | **84.37±0.96** | **77.78±1.27** | **88.02±1.92** | **76.77±2.02** | **89.21±0.67** | 1.19 |
| | Trainable GSL | **90.41±0.61** | 83.10±0.58 | **52.10±0.62** | 84.35±0.56 | OOM | 86.23±1.58 | 74.39±1.14 | 88.13±0.56 | 2.19 |
| | Non-trainable GSL | 89.96±0.79 | 77.23±1.63 | 49.79±0.72 | 82.78±0.95 | 76.67±1.13 | 86.97±1.75 | 75.20±1.55 | 87.97±0.51 | 2.62 |
| ACMGNN | No GSL | **90.56±1.03** | **84.86±0.73** | **52.07±1.72** | **84.41±1.12** | **77.72±1.59** | **88.23±1.81** | **76.63±2.34** | 89.37±0.56 | 1.06 |
| | Trainable GSL | **90.56±0.68** | 81.90±0.71 | 51.87±0.44 | 84.40±0.79 | OOM | 83.11±1.81 | 73.91±1.16 | 88.55±0.39 | 2.19 |
| | Non-trainable GSL | 87.46±1.02 | 74.63±0.76 | 49.35±0.58 | 81.63±0.87 | 73.84±1.41 | 80.83±1.84 | 73.43±1.47 | **88.98±0.47** | 2.75 |
| MixHop | No GSL | **90.10±5.59** | **81.70±0.89** | **50.95±0.71** | **84.56±1.19** | **77.66±1.24** | **87.76±1.94** | **76.51±1.93** | **89.42±0.81** | 1.12 |
| | Trainable GSL | **90.10±0.52** | 79.07±0.75 | **50.95±0.71** | 84.55±0.67 | OOM | 84.84±1.28 | 74.45±1.11 | 88.48±0.62 | 2.25 |
| | Non-trainable GSL | 85.43±0.57 | 55.95±2.35 | 44.15±0.59 | 76.54±0.91 | 72.03±2.45 | 85.36±0.89 | 74.68±1.13 | 88.18±0.52 | 2.62 |

In Table 14, we present the results of applying trainable GSL to baseline GNNs. Specifically, we select the best-performing GSL variants, as shown in Tables 1 and 13, for each backbone GNN. The best-performing method is highlighted in bold, while the runner-up is indicated with an underline. "OOM" refers to "out of memory." The results demonstrate the following: (1) The average rank indicates that trainable GSL improves GNN performance on 5 out of 6 GNN backbones; (2) Although trainable GSL outperforms non-trainable GSL, it remains inferior to GNN backbones without GSL, indicating that GSL could be unnecessary in improving GNN performance on node classification.

### E.7 PERFORMANCE ON GRAPH CLASSIFICATION

In addition to the node classification experiments, we further investigate whether GSL consistently improves GNN performance in graph classification. Specifically, we conduct ablation experiments by replacing the GSL graph with the original graph, following the methodology outlined in (Li et al., 2023c). As shown in 15, removing GSL from 4 state-of-the-art GNNs, including ProGNN (Jin et al., 2020), GEN (Wang et al., 2021), GRCN (Yu et al., 2020), and IDGL (Chen et al., 2020), results in significantly reduced training time. At the same time, the GNN performance remains comparable to that of the GSL-enhanced counterparts. This suggests that GSL does not consistently enhance GNN performance in graph classification. Due to page limitations, we only tested a few methods in this paper. We believe it would be valuable to explore additional state-of-the-art methods, datasets, and theoretical justifications for the effectiveness of GSL in graph classification in future work.

Table 15: Ablation study of GSL-enhanced methods for graph classification.

| Model | Cora | | PubMed | | CiteSeer | |
|---|---|---|---|---|---|---|
| | AUC | Time | AUC | Time | Acc | Time |
| ProGNN | 76.28±0.52 | 959s | OOM | - | 67.14±0.23 | 1776s |
| ProGNN,w/o. GSL | 78.96±0.64 | 30s | 75.80±0.95 | 326s | 67.24±1.48 | 44s |
| GEN | 79.88 ± 0.93 | 219s | OOM | - | 66.98 ± 1.28 | 320s |
| GEN,w/o. GSL | 78.32 ± 1.21 | 3s | 76.94 ± 0.40 | 47s | 64.66 ± 1.46 | 3s |
| GRCN | 83.04 ± 0.33 | 56s | 74.55 ± 0.96 | 249s | 70.85 ± 0.87 | 113s |
| GRCN,w/o. GSL | 71.82 ± 0.61 | 9s | 74.18 ± 0.63 | 28s | 58.33 ± 0.17 | 24s |
| IDGL | 83.32 ± 0.59 | 144s | OOM | - | 70.57 ± 0.26 | 330s |
| IDGL,w/o. GSL | 83.32 ± 0.59 | 129s | OOM | - | 71.12 ± 0.31 | 401s |

## E.8 ROBUSTNESS OF GSL

Table 16: Ablation study on model robustness by adding edges in GSL-enhanced methods.

| Use GSL | Method | Dataset | 0% | 10% | 20% | 30% | 40% | 50% | 60% | 80% |
|---|---|---|---|---|---|---|---|---|---|---|
| w/o. GSL | Gaug | CiteSeer | 72.34 | 68.85 | 67.12 | 62.87 | 65.03 | 62.22 | 60.32 | 56.58 |
| w. GSL | Gaug | CiteSeer | 72.79 | 71.62 | 68.08 | 63.96 | 63.68 | 62.08 | 60.14 | 56.78 |
| w/o. GSL | Gaug | Cora | 81.73 | 76.83 | 74.07 | 70.53 | 68.67 | 67.70 | 67.42 | 54.93 |
| w. GSL | Gaug | Cora | 82.43 | 79.14 | 77.86 | 73.54 | 72.54 | 70.88 | 69.94 | 59.83 |
| w/o. GSL | IDGL | CiteSeer | 73.13 | 68.55 | 66.45 | 64.33 | 64.55 | 59.66 | 60.66 | 57.59 |
| w. GSL | IDGL | CiteSeer | 73.26 | 71.11 | 68.62 | 67.83 | 65.79 | 63.89 | 63.17 | 60.23 |
| w/o. GSL | IDGL | Cora | 82.43 | 78.65 | 78.18 | 77.98 | 74.33 | 74.53 | 72.36 | 65.23 |
| w. GSL | IDGL | Cora | 84.12 | 82.86 | 81.28 | 79.79 | 77.13 | 76.87 | 75.42 | 69.17 |
| w/o. GSL | GRCN | CiteSeer | 69.55 | 61.11 | 61.36 | 56.39 | 56.67 | 53.87 | 51.62 | 52.08 |
| w. GSL | GRCN | CiteSeer | 73.21 | 69.89 | 67.87 | 64.94 | 63.56 | 61.53 | 60.36 | 56.53 |
| w/o. GSL | GRCN | Cora | 81.66 | 74.89 | 72.01 | 69.98 | 66.76 | 66.19 | 61.85 | 59.63 |
| w. GSL | GRCN | Cora | 84.64 | 81.49 | 77.37 | 76.34 | 74.27 | 72.13 | 71.53 | 68.38 |

Table 17: Ablation study on model robustness by deleting edges in graphs for GSL-enhanced methods.

| Use GSL | Method | Dataset | 0% | 10% | 20% | 30% | 40% | 50% | 60% | 80% |
|---|---|---|---|---|---|---|---|---|---|---|
| w/o. GSL | Gaug | CiteSeer | 72.34 | 71.72 | 70.83 | 69.65 | 68.52 | 67.75 | 65.65 | 65.22 |
| w. GSL | Gaug | CiteSeer | 72.79 | 70.97 | 71.88 | 69.86 | 68.66 | 70.48 | 68.42 | 65.72 |
| w/o. GSL | Gaug | Cora | 81.73 | 78.65 | 78.18 | 77.98 | 74.33 | 74.53 | 72.30 | 65.24 |
| w. GSL | Gaug | Cora | 82.40 | 79.87 | 79.05 | 78.83 | 77.54 | 77.16 | 75.31 | 68.91 |
| w/o. GSL | IDGL | CiteSeer | 73.13 | 71.13 | 72.42 | 70.40 | 69.19 | 67.07 | 67.59 | 65.67 |
| w. GSL | IDGL | CiteSeer | 73.26 | 72.45 | 71.83 | 72.85 | 70.87 | 69.05 | 68.80 | 67.73 |
| w/o. GSL | IDGL | Cora | 82.43 | 81.39 | 80.15 | 79.58 | 78.72 | 76.12 | 74.44 | 67.99 |
| w. GSL | IDGL | Cora | 84.14 | 82.12 | 81.87 | 80.96 | 80.53 | 79.12 | 77.25 | 71.63 |
| w/o. GSL | GRCN | CiteSeer | 69.55 | 67.72 | 66.64 | 64.53 | 61.79 | 62.64 | 61.43 | 58.61 |
| w. GSL | GRCN | CiteSeer | 73.21 | 73.23 | 72.23 | 73.15 | 70.84 | 70.49 | 69.82 | 67.73 |
| w/o. GSL | GRCN | Cora | 81.66 | 76.51 | 74.39 | 74.71 | 71.64 | 71.77 | 68.02 | 60.63 |
| w. GSL | GRCN | Cora | 84.64 | 83.16 | 82.26 | 81.36 | 80.57 | 79.38 | 77.62 | 74.52 |

To investigate the robustness of GSL under noisy graph structures, we randomly add or delete edges in graphs following (Li et al., 2023c). As shown in Table 16 and Table 17, we randomly add or remove $[0\%, 10\%, \ldots, 80\%]$ edges in original graphs. The results show that, generally GSL-enhanced GNNs are more robust to a higher ratio of noises in graph structures. There are still some cases where GSL cannot improve the model robustness, such as GAug on CiteSeer. It will be interesting to empirically and theoretically investigate the robustness of GSL-enhanced models in the future.

## E.9 WEAKLY SUPERVISED GSL

Table 18: Ablation experiments of GSL-enhanced methods in weakly-supervised settings.

| Method | Dataset | Use GSL | 20 labels | 10 labels | 5 labels | 3 labels |
|--------|---------|---------|-----------|-----------|----------|----------|
| Gaug | PubMed | w/o. GSL | 79.68 | 72.87 | 67.66 | 63.30 |
| | | w. GSL | 78.73 | 75.48 | 69.84 | 65.90 |
| | CiteSeer | w/o. GSL | 72.30 | 64.62 | 57.68 | 51.53 |
| | | w. GSL | 72.79 | 67.02 | 58.38 | 54.37 |
| | Cora | w/o. GSL | 81.71 | 74.02 | 65.67 | 60.13 |
| | | w. GSL | 82.48 | 76.12 | 69.46 | 65.97 |
| Grcn | Mines. | w/o. GSL | 71.61 | 67.11 | 61.66 | 61.31 |
| | | w. GSL | 71.15 | 64.72 | 59.47 | 58.85 |
| | Cora | w/o. GSL | 81.66 | 72.52 | 67.88 | 64.09 |
| | | w. GSL | 84.60 | 81.74 | 76.85 | 75.42 |
| | CiteSeer | w/o. GSL | 69.55 | 59.35 | 55.66 | 51.72 |
| | | w. GSL | 72.30 | 70.28 | 69.54 | 68.63 |
| Idgl | CiteSeer | w/o. GSL | 73.13 | 64.62 | 56.62 | 50.79 |
| | | w. GSL | 73.26 | 66.08 | 62.69 | 55.61 |
| | Cora | w/o. GSL | 82.43 | 75.85 | 69.21 | 64.47 |
| | | w. GSL | 84.19 | 78.33 | 73.46 | 69.94 |

We further conduct an ablation study on the performance of GSL-enhanced methods. As shown in Table 18, the performance of these GSL-enhanced methods is comparable to their counterparts without GSL when there are 20 labels per class. However, as we decrease the supervision ratio (such as in scenarios with 5 or 3 labels per class) the GSL-enhanced methods demonstrate improved performance compared to those without GSL. These results indicate that GSL can effectively enhance GNN performance in settings with low supervision.

## E.10 ADDITIONAL ABLATION STUDY ON GSL-ENHANCED GNNS

Table 19: Ablation study on additional GSL-enhanced methods including Grale and MetricNN.

| Model | Questions | | Minesweeper | | Roman-empire | | Amazon-ratings | | Tolokers | |
|-------|-----------|------|-------------|------|--------------|------|----------------|------|----------|------|
| | AUC | Time | AUC | Time | Acc | Time | Acc | Time | AUC | Time |
| Grale | 68.96±0.23 | 320s | 66.36±0.08 | 31s | 35.42±0.57 | 129s | 46.57±0.37 | 145s | 73.32±0.72 | 110s |
| Grale,w/o. GSL | 68.34±1.18 | 283s | 62.42±0.20 | 9s | 64.90±0.26 | 130s | 48.38±0.94 | 136s | 74.49±0.45 | 48s |
| MetricNN | 64.28 ± 1.34 | 23s | 68.51 ± 1.63 | 2.1s | 38.55 ± 1.68 | 8.3s | 42.10 ± 1.22 | 10s | 69.41 ± 5.50 | 5.4s |
| MetricNN,w/o. GSL | 65.27 ± 0.86 | 11s | 73.51 ± 0.01 | 2.1s | 37.60 ± 2.26 | 4.3s | 42.28 ± 1.11 | 5.8s | 75.20 ± 0.48 | 3.4s |

We also conducted experiments on two GSL-enhanced GNNs, Grale (Halcrow et al., 2020) and MetricNN (Garcia & Bruna, 2017), using the same settings as in our previous experiments. As shown in Table 19, removing GSL from these methods resulted in improved model performance on most datasets, along with reduced training time. These findings further support the conclusion drawn in this paper that GSL is unnecessary for enhancing model performance in node classification.

Table 20: Ablation study on model robustness by introducing feature noises in graphs for GSL-enhanced methods.

| Use GSL | Method | Dataset | 0% | 10% | 20% | 30% | 40% | 50% | 60% | 80% |
|---------|--------|---------|------|------|------|------|------|------|------|------|
| w/o. GSL | Gaug | CiteSeer | 72.34 | 69.83 | 69.37 | 69.72 | 67.24 | 65.87 | 62.67 | 57.07 |
| w. GSL | Gaug | CiteSeer | 72.79 | 71.14 | 71.27 | 69.90 | 67.51 | 67.98 | 63.56 | 58.09 |
| w/o. GSL | Gaug | Cora | 81.73 | 78.96 | 80.80 | 78.86 | 77.84 | 76.87 | 73.27 | 67.40 |
| w. GSL | Gaug | Cora | 82.48 | 53.87 | 51.81 | 51.20 | 48.66 | 54.41 | 48.05 | 35.63 |
| w/o. GSL | Gaug | PubMed | 79.38 | 78.75 | 77.08 | 77.97 | 77.50 | 76.68 | 74.43 | 68.90 |
| w. GSL | Gaug | PubMed | 78.73 | 79.09 | 78.57 | 77.93 | 77.42 | 77.95 | 76.28 | 71.20 |
| w/o. GSL | IDGL | CiteSeer | 73.13 | 71.41 | 70.84 | 69.31 | 66.57 | 65.98 | 62.50 | 53.44 |
| w. GSL | IDGL | CiteSeer | 73.26 | 72.24 | 71.86 | 71.03 | 69.89 | 68.46 | 66.02 | 58.28 |
| w/o. GSL | IDGL | Cora | 82.43 | 80.72 | 81.24 | 79.11 | 78.53 | 78.08 | 73.57 | 68.73 |
| w. GSL | IDGL | Cora | 84.19 | 83.11 | 82.08 | 80.64 | 80.31 | 80.02 | 77.51 | 74.92 |
| w/o. GSL | GRCN | CiteSeer | 69.55 | 65.42 | 64.17 | 65.99 | 63.96 | 59.64 | 57.22 | 47.74 |
| w. GSL | GRCN | CiteSeer | 72.34 | 70.58 | 67.70 | 67.11 | 64.28 | 61.13 | 58.10 | 53.25 |
| w/o. GSL | GRCN | Cora | 81.66 | 76.40 | 76.86 | 76.48 | 74.32 | 74.86 | 72.98 | 64.71 |
| w. GSL | GRCN | Cora | 84.61 | 80.30 | 79.83 | 76.36 | 77.96 | 75.23 | 73.51 | 68.49 |
| w/o. GSL | GRCN | PubMed | 79.35 | 74.57 | 74.96 | 74.74 | 73.46 | 76.09 | 74.35 | 71.26 |
| w. GSL | GRCN | PubMed | 79.30 | 78.89 | 78.36 | 77.40 | 77.11 | 76.75 | 75.95 | 73.85 |

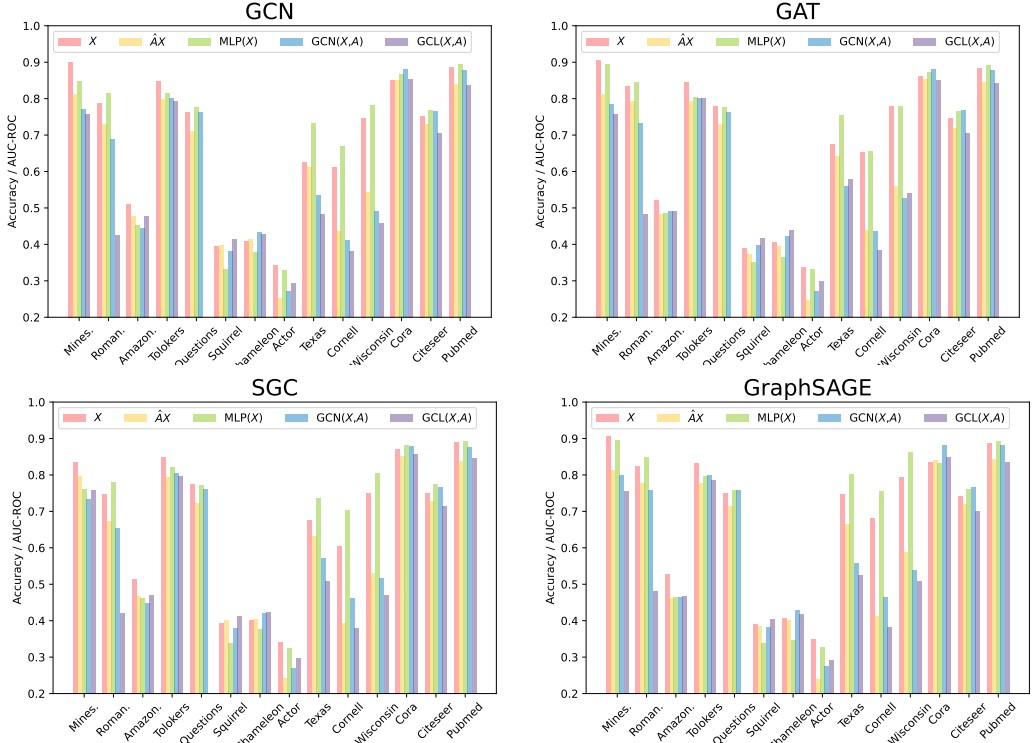

Figure 6: Influences of different GSL bases to more GNNs.

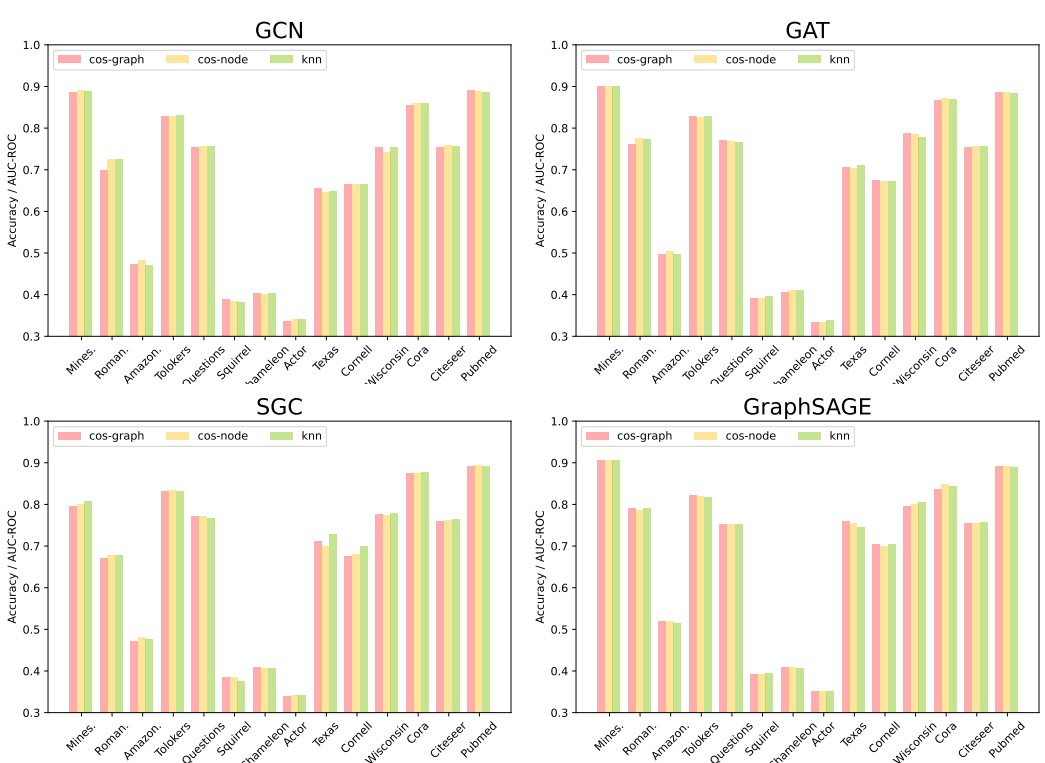

Figure 7: Influences of the approaches of GSL generation to GNN+GSL.

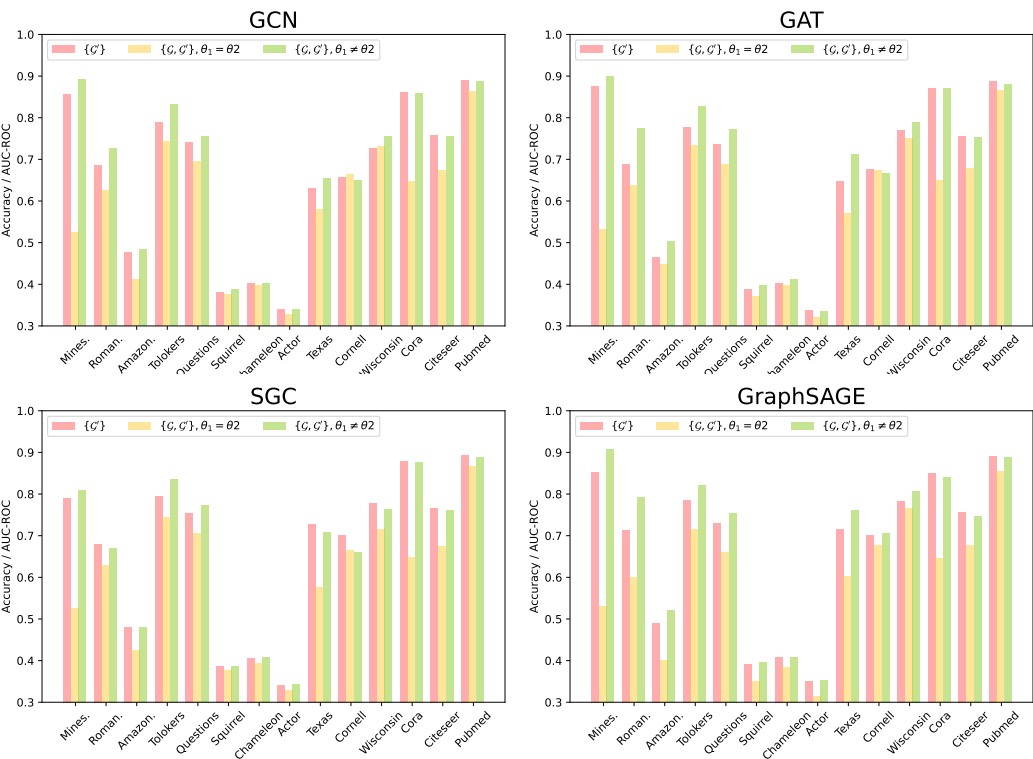

Figure 8: Influences of the approaches of view fusion in GSL to GNN+GSL.

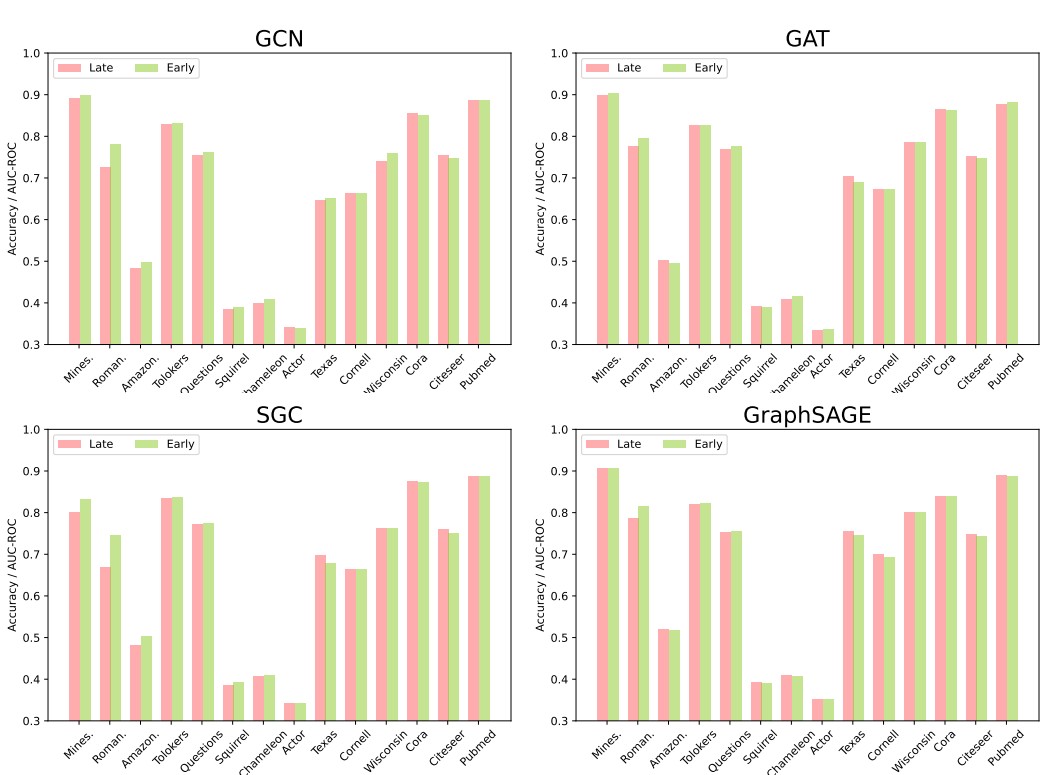

Figure 9: Influences of the states of view fusion in GSL to GNN+GSL.

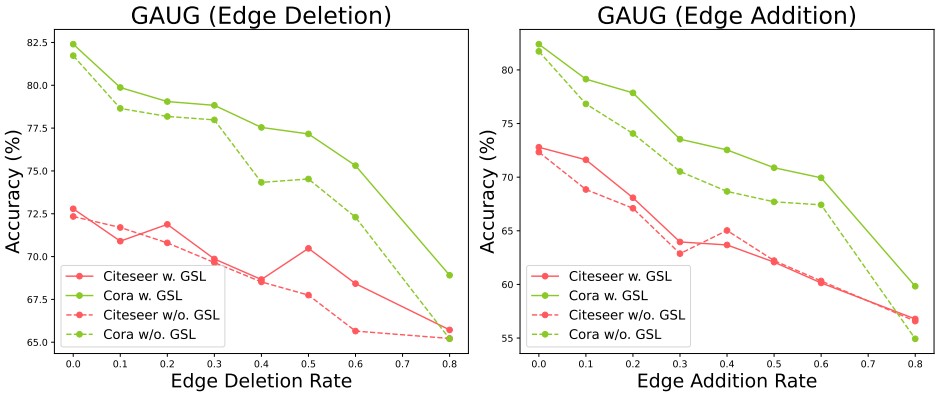

Figure 10: Model robustness when injecting random noise on GAug.

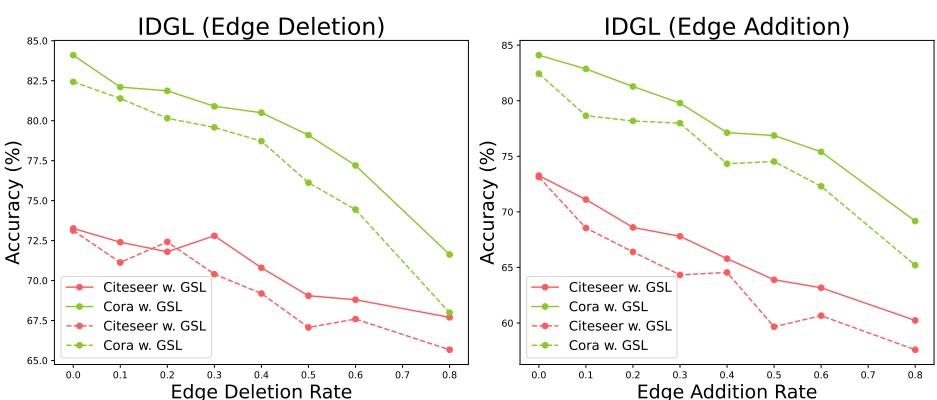

Figure 11: Model robustness when injecting random noise on Idgl.

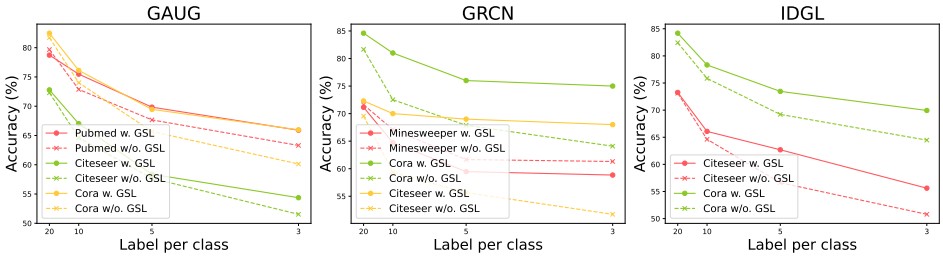

Figure 12: Ablation experiments of GSL-enhanced methods in weakly-supervised settings.

