{\boldsymbol{Y}} = MLP(\boldsymbol{X})$, $\min \mathcal{L}_{CE}(\hat{\boldsymbol{Y}}, \boldsymbol{Y})$ | $\{\mathcal{E}' = \boldsymbol{B}\boldsymbol{Q}\boldsymbol{B}^T\}$ | norm($\mathcal{E}'$) | Early Fusion, $\{\mathcal{G} \odot \mathcal{G}'\}$ | Joint |
| WSGNN (Lao et al., 2022) | MLP($\boldsymbol{X}$) | $\{\mathcal{E}'|e'_{ij} = cos(\boldsymbol{B}_i, \boldsymbol{B}_j)\}$ | - | Early Fusion, $\{\mathcal{G} + \mathcal{G}'\}$ | Joint |
| GLCN (Jiang et al., 2019) | $\boldsymbol{X}$ | $\{\mathcal{E}'|e'_{ij} = \phi(|\boldsymbol{B}_i - \boldsymbol{B}_j|)\}$ | norm($\mathcal{E}'$), Original +Sparsity+Smoothness | No Fusion, $\{\mathcal{G}'\}$ | Joint |
| ASC (Li et al., 2023a) | SpectralCluster($\boldsymbol{X}$) | $\{\mathcal{E}'|e'_{ij} = \|\boldsymbol{B}_i - \boldsymbol{B}_j\|\}$ | topk($\mathcal{E}'$) | No Fusion, $\{\mathcal{G}'\}$ | Static |
| WRGAT (Suresh et al., 2021) | GCN($\boldsymbol{X}, \boldsymbol{A}$) | $\{\mathcal{E}'|e'_{ij} \cdot Opt(\boldsymbol{B})\}$ | Sparsity + MultiHop | Early Fusion $\{\mathcal{G} + \mathcal{G}'\}$ | Static |
| HOG-GCN (Wang et al., 2022) | GCN($\boldsymbol{X}, \boldsymbol{A}$) | $\{\mathcal{E}'|e'_{ij} = \sigma(\boldsymbol{B}_i \boldsymbol{B}_j^T)\}$ | Sparsity + Smoothness | No Fusion $\{\mathcal{G}'\}$ | Joint |
| GGCN (Yan et al., 2022) | MLP($\boldsymbol{X}$) | $\{\mathcal{E}'|e'_{ij} = \cos(\boldsymbol{B}_i, \boldsymbol{B}_j)\}$ | Low Rank + Sparsity | Early Fusion, $\{\mathcal{G} + \mathcal{G}'\}$ | Joint |
| GloGNN (Li et al., 2022b) | MLP($\boldsymbol{X}$) | $\{\mathcal{E}' = Opt(\boldsymbol{B})\}$ | Sparsity+MultiHop | No Fusion, $\{\mathcal{G}'\}$ | Joint |
| HiGNN (Zheng et al., 2024b) | $\hat{\boldsymbol{Y}} = GCN(\boldsymbol{X}, \boldsymbol{A})$, $\min \mathcal{L}_{CE}(\hat{\boldsymbol{Y}}, \boldsymbol{Y})$ | $\{\mathcal{E}' =

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

 | {G'} | - | 47.36±3.47 | 60.97±0.76 | 41.50±0.75 | 70.21±1.51 | 67.32±1.37 | 38.12±1.92 | 39.90±3.64 | 33.43±0.95 | 79.80±6.99 | 59.46±8.35 | 71.57±6.68 | 77.47±2.41 | 73.68±0.97 | 87.19±0.38 | 7.64 |
| ACMGNN | cos-graph | {G,G'} | θ₁=θ₂ | 52.74±5.22 | 51.18±2.12 | 33.11±1.38 | 69.06±4.65 | 62.30±3.23 | 31.58±4.39 | 38.79±4.73 | 29.06±2.60 | 54.10±7.59 | 59.19±8.87 | 70.39±9.58 | 59.74±1.87 | 65.17±1.94 | 79.53±1.69 | 9.86 |
| ACMGNN | cos-graph | {G,G'} | θ₁≠θ₂ | 87.46±1.02 | 74.63±0.76 | 49.35±0.58 | 81.63±0.87 | 73.84±1.41 | 38.54±1.89 | 41.16±4.18 | 34.23±0.98 | 67.67±5.97 | 70.00±5.90 | 80.78±5.21 | 80.83±1.84 | 73.43±1.47 | 88.98±0.47 | 3.64 |
| ACMGNN | cos-node | {G'} | - | 52.83±3.52 | 61.26±0.62 | 42.47±0.53 | 74.14±1.14 | 72.23±1.36 | 38.23±1.97 | 40.77±3.68 | 34.74±0.90 | 61.45±6.13 | 63.51±5.87 | 74.31±6.43 | 75.84±2.93 | 73.05±1.18 | 87.22±0.41 | 6.21 |
| ACMGNN | cos-node | {G,G'} | θ₁=θ₂ | 52.74±5.22 | 51.18±2.12 | 33.11±1.38 | 69.06±4.65 | 62.30±3.23 | 31.58±4.39 | 38.79±4.73 | 29.06±2.60 | 54.10±7.59 | 59.19±8.87 | 70.39±9.58 | 59.74±1.87 | 65.17±1.94 | 79.53±1.69 | 9.86 |
| ACMGNN | cos-node | {G,G'} | θ₁≠θ₂ | 87.80±0.97 | 73.55±0.51 | 49.04±0.57 | 80.74±0.92 | 74.11±1.40 | 39.19±2.12 | 40.28±4.30 | 34.19±1.16 | 69.86±5.56 | 69.46±7.21 | 80.39±5.23 | 80.33±1.90 | 73.31±1.26 | 88.94±0.36 | 4.07 |
| ACMGNN | kNN | {G'} | - | 51.68±3.38 | 60.86±0.87 | 41.68±0.95 | 71.31±0.64 | 69.56±1.41 | 38.58±1.96 | 40.56±2.34 | 34.88±0.77 | 62.51±6.16 | 62.70±5.95 | 76.47±4.43 | 75.99±2.85 | 70.20±1.51 | 87.20±0.45 | 6.64 |
| ACMGNN | kNN | {G,G'} | θ₁=θ₂ | 52.74±5.22 | 51.18±2.12 | 33.11±1.38 | 69.06±4.65 | 62.30±3.23 | 31.58±4.39 | 38.79±4.73 | 29.06±2.60 | 54.10±7.59 | 59.19±8.87 | 70.39±9.58 | 59.74±1.87 | 65.17±1.94 | 79.53±1.69 | 9.86 |
| ACMGNN | kNN | {G,G'} | θ₁≠θ₂ | 87.59±0.88 | 73.21±0.63 | 49.06±0.53 | 81.34±0.85 | 73.95±1.35 | 39.18±2.18 | 41.70±3.71 | 34.67±1.11 | 68.48±5.78 | 68.92±5.87 | 80.20±3.13 | 80.46±2.26 | 73.14±1.31 | 88.87±0.51 | 4.07 |
| MLP | - | - | - | 79.55±1.23 | 65.45±0.99 | 46.65±0.83 | 75.94±1.38 | 74.92±1.39 | 39.29±2.22 | 43.57±4.18 | 35.40±1.38 | 80.46±6.44 | 73.78±7.34 | 85.88±7.78 | 87.97±1.80 | 76.68±2.10 | 87.39±2.18 | 2.29 |
| MixHop | - | - | - | 90.10±5.59 | 81.70±0.89 | 50.95±0.71 | 84.56±1.19 | 77.66±1.24 | 41.22±2.66 | 43.11±4.73 | 33.59±1.23 | 72.54±8.98 | 62.43±9.54 | 75.88±8.27 | 87.76±1.94 | 76.51±1.93 | 89.42±0.81 | 1.86 |
| MixHop | cos-graph | {G'} | - | 64.75±4.59 | 51.83±0.53 | 41.47±2.00 | 68.78±1.94 | 71.45±1.38 | 37.75±2.41 | 37.79±2.10 | 31.77±1.75 | 55.72±6.39 | 60.27±5.85 | 70.20±4.60 | 84.42±1.35 | 74.20±0.83 | 88.74±0.29 | 8.21 |
| MixHop | cos-graph | {G,G'} | θ₁=θ₂ | 54.22±10.75 | 63.50±0.86 | 44.21±1.36 | 74.22±2.21 | 70.64±1.32 | 37.16±1.34 | 39.06±3.08 | 32.24±1.33 | 58.16±9.18 | 66.22±5.59 | 73.73±7.80 | 65.14±2.62 | 68.66±1.24 | 86.63±0.51 | 7.54 |
| MixHop | cos-graph | {G,G'} | θ₁≠θ₂ | 84.71±1.19 | 55.41±1.63 | 43.37±0.75 | 74.41±1.33 | 69.63±2.03 | 37.64±2.19 | 38.71±4.36 | 31.73±1.77 | 61.13±7.96 | 61.35±7.10 | 75.29±6.00 | 85.42±1.21 | 74.57±1.34 | 88.16±0.46 | 6.50 |
| MixHop | cos-node | {G'} | - | 60.56±7.08 | 51.74±0.68 | 42.71±0.97 | 74.27±1.84 | 72.83±1.12 | 38.35±1.99 | 38.88±3.00 | 33.05±1.04 | 58.42±6.52 | 62.27±5.98 | 71.57±6.91 | 83.22±1.16 | 74.11±1.12 | 88.23±0.45 | 6.71 |
| MixHop | cos-node | {G,G'} | θ₁=θ₂ | 54.22±10.75 | 63.50±0.86 | 44.21±1.36 | 74.22±2.21 | 70.64±1.32 | 37.16±1.34 | 39.06±3.08 | 32.24±1.33 | 58.16±9.18 | 66.22±5.59 | 73.73±7.80 | 65.14±2.62 | 68.66±1.24 | 86.63±0.51 | 7.64 |
| MixHop | cos-node | {G,G'} | θ₁≠θ₂ | 85.43±0.57 | 55.95±2.35 | 44.15±0.59 | 76.54±0.91 | 72.03±2.45 | 37.47±2.07 | 39.52±3.33 | 32.50±1.10 | 60.61±8.73 | 62.97±6.75 | 75.10±6.20 | 85.36±0.89 | 74.68±1.13 | 88.18±0.52 | 4.79 |
| MixHop | kNN | {G'} | - | 59.50±6.26 | 50.39±0.72 | 42.07±0.93 | 70.49±1.70 | 69.57±1.32 | 38.07±1.72 | 38.76±2.91 | 33.23±1.30 | 59.25±4.49 | 57.30±6.96 | 69.22±7.22 | 83.99±1.28 | 74.96±1.18 | 87.99±0.40 | 8.00 |
| MixHop | kNN | {G,G'} | θ₁=θ₂ | 54.22±10.75 | 63.50±0.86 | 44.21±1.36 | 74.22±2.21 | 70.64±1.32 | 37.16±1.34 | 39.06±3.08 | 32.24±1.33 | 58.16±9.18 | 66.22±5.59 | 73.73±7.80 | 65.14±2.62 | 68.66±1.24 | 86.63±0.51 | 7.54 |
| MixHop | kNN | {G,G'} | θ₁≠θ₂ | 85.53±0.50 | 57.48±1.98 | 43.28±0.68 | 77.24±1.61 | 70.34±1.76 | 38.15±2.01 | 40.12±3.76 | 32.30±1.53 | 60.05±9.45 | 63.51±7.56 | 74.90±8.21 | 85.18±1.26 | 74.59±1.19 | 88.20±0.57 | 4.93 |