# OpenReview forum: "Rethinking Structure Learning For Graph Neural Networks"
_ICLR.cc/2025/Conference — Submitted to ICLR 2025_

### Official Review · Reviewer_VCb5 · 2024-10-27

**Soundness:** 3
**Presentation:** 3
**Contribution:** 2
**Rating:** 5
**Confidence:** 4

**Summary:**

This study examines critically on the role of Graph Structure Learning (GSL) in improving Graph Neural Networks (GNNs).
Although GSL has been used to improve GNN performance by capturing semantically similar nodes, the authors claim that there is insufficient theoretical and empirical evidence to support its necessity. The authors also show GSL's training times and the need for extensive hyperparameter tuning.

the paper proposes a new framework for categorizing GSL methods into three components: GSL base generation, new structure construction, and view fusion, which provides a better understanding of GSL's elements.
According to the authors' empirical analysis, GSL does not consistently outperform standard GNN approaches.
Their findings show that GSL fails to provide more mutual information gain than traditional methods.
Finally, the authors conclude that non-optimization-based GSL methods are frequently unnecessary because the quality of the original GSL bases guarantees informative node representations.
This calls into question the prevailing assumptions about GSL's role in effective GNN performance and suggests reassessing its importance in future GNN designs.

**Strengths:**

1. The authors present a GSL framework, which is a valuable contribution. It provides a common terminology and aligns the community on the same page, facilitating clearer communication. Additionally, a unified framework benefits the subsequent empirical comparisons and discussions.

2. The empirical findings are solid. The observations in Section 4.1, particularly Observations 1-3, offer clear takeaways for practitioners. Furthermore, the empirical comparisons in Section 5.1 are thorough and comprehensive.

**Weaknesses:**

1. The primary weakness of this work is its lack of novelty. The key conclusion (especially the second point in the contribution list) seems rather obvious, at least to those familiar with GSL methods. Even for those who haven't empirically tested GSL methods, the result isn't surprising. If one assumes that the GNN model is sufficiently strong, it logically follows that the GSL method would be redundant. GSL appears more like an intermediate tool, and a good GNN model should outperform or make GSL unnecessary in an end-to-end training/inference.

2. In the last paragraph of the Introduction, the mentioned "theoretical analysis" (Appendix B) feels more like a derivation than a true theoretical analysis. Moreover, the analysis is too coarse-grained and offers little insight. In line 83, the claim that "GSL bases serve as the upper bound" is poorly framed. The GSL bases in this context refer to features, while the upper bound is a scalar term derived from these bases. The way this upper bound argued is a non-professional theoretical claim.

**Questions:**

1. Novelty of the Main Contribution: Can the authors elaborate more on the novelty of their key conclusions, especially regarding the second point in the contribution list? The observation that a strong GNN model might diminish the need for GSL methods feels intuitive and not surprising. How does this work differentiate itself from previous studies in this area? Are there specific aspects of GSL not yet addressed by prior research that the authors are tackling?

2. Theoretical Analysis vs. Derivation: In the introduction, the authors reference a theoretical analysis presented in Appendix B. However, the analysis appears to be more of a mathematical derivation rather than a comprehensive theoretical insight. Could the authors clarify why they classify it as a theoretical analysis? How does this analysis provide deeper understanding beyond the mere derivation of terms?

---

> ### Author Response · Authors · 2024-11-22
> **Part (1/2)**
>
> ### W1: 1) The primary weakness of this work is its lack of novelty. The key conclusion (especially the second point in the contribution list) seems rather obvious, at least to those familiar with GSL methods. Even for those who haven't empirically tested GSL methods, the result isn't surprising. 2) If one assumes that the GNN model is sufficiently strong, it logically follows that the GSL method would be redundant. GSL appears more like an intermediate tool, and a good GNN model should outperform or make GSL unnecessary in an end-to-end training/inference.
>
> 1. Thanks for your valuable feedback. To the best of our knowledge, the conclusion, "GSL is unnecessary for improving GNN performance on node classification", is first proposed in our paper. This conclusion is not obvious, otherwise, there won't be so many papers [1, 2, 3, 4] that use GSL to improve GNN performance while adding tremendous computational complexity. While many approaches are trying to add more and more **complex model designs** in GNNs to get **tiny improvement**, **we want to do reduction in the model design** by rethinking the necessity of the GSL. Therefore, we think our work remains a high novelty in this area.
>
> 2. We clarify that, in our paper, the redundancy of GSL in improving GNN on node classification tasks is **irrelevant** to the performance of GNN, because there is no mutual information gain after the operation of graph convolution on GSL graphs, as shown by our empirical studies and theoretical analysis. Besides, for some heterophilous datasets, such as Squirrel, Chameleon, and Actor, where GNN performance is not strong, GSL still cannot improve model performance, as shown in Table 1.
>
> ### W2: 1)In the last paragraph of the Introduction, the mentioned "theoretical analysis" (Appendix B) feels more like a derivation than a true theoretical analysis. Moreover, the analysis is too coarse-grained and offers little insight. 2) In line 83, the claim that "GSL bases serve as the upper bound" is poorly framed. 3) The GSL bases in this context refer to features, while the upper bound is a scalar term derived from these bases. The way this upper bound argued is a non-professional theoretical claim.
>
> 1.  Thank you for your constructive suggestions. We have revised "Theorem 1" to "Proposition 1" to enhance its rigor. Additionally, we have clarified the conditions of our proof, which include the independence assumption, Gaussian distribution, and the non-parametric GSL method. We believe these conditions contribute to a more rigorous analysis.
> 2. We have revised our paper to improve its readability and rigor: "the mutual information (MI) between node representations and labels does not increase after applying graph convolution on GSL graphs constructed by similarity."
> 3. To make the theoretical claim more professional, we have revised the proof to extend the analysis to multi-dimensional features:
>
> Specifically, given the conclusion we get from a single feature $B_i'^k = \sum_{q=1}^C p_{k,q} B_i^q$ , we apply the chain rule of mutual information to extend this conclusion to multi-dimensional variables.
>
> $$
> I(Y; \mathbf{B}) = I(Y; \{B_1, \ldots, B_n\}) = \sum_{i=1}^n I(Y; B_i \mid \{B_1, \ldots, B_{i-1}\})
> $$
>
> $$
> I(Y; \mathbf{B'}) = I(Y; \{B_1', \ldots, B_n'\}) = \sum_{i=1}^n I(Y; B_i' \mid \{B_1', \ldots, B_{i-1}'\})
> $$
>
> Due to the property that conditioning reduces entropy, we have
>
> $$
> I(Y; B_i \mid \{B_1, \ldots, B_{i-1}\}) \geq I(Y; B_i)
> $$
>
> $$
> I(Y; B_i' \mid \{B_1', \ldots, B_{i-1}'\}) \geq I(Y; B_i')
> $$
>
> Thus, we have
>
> $$
> \inf I(Y; \mathbf{B})=\sum_{i=1}^n I(Y; B_i) \ \text{and} \ \inf I(Y; \mathbf{B'})=\sum_{i=1}^n I(Y; B_i')
> $$
> where $\inf$ represents infimum. Since $I(Y;B'_i)\le I(Y,B_i)$ holds for each $i$, we have
>
> $$
> \inf I(Y;\textbf{B'}) \le \inf I(Y;\textbf{B})
> $$
>
> This concludes the proof of Proposition 1.

---

> ### Author Response · Authors · 2024-11-22
> **Part (2/2)**
>
> ### Q1: 1) Novelty of the Main Contribution: Can the authors elaborate more on the novelty of their key conclusions, especially regarding the second point in the contribution list? 2) The observation that a strong GNN model might diminish the need for GSL methods feels intuitive and not surprising. 3) How does this work differentiate itself from previous studies in this area? Are there specific aspects of GSL not yet addressed by prior research that the authors are tackling?
>
> For 1) and 2), please refer to our responses to W1.
>
> For 3), we restate how this work differs from previous studies:
>
> 1. We are the first to propose a framework that de-compose GSL into 3 aspects, which helps us better understand GSL.
> 2. We first empirically and theoretically show that there is no mutual information gain for the GSL bases and labels $I(X;Y)$ after applying graph convolution on GSL graphs. This result indicates that GSL is unnecessary in improving graph representation learning in node classification in graphs, as verified by our experiments.
> 3. We analyze how different GSL components influence the performance of GSL in GNN, which help the model designs in GNNs in the future.
>
> ### Q2: Theoretical Analysis vs. Derivation: In the introduction, the authors reference a theoretical analysis presented in Appendix B. However, the analysis appears to be more of a mathematical derivation rather than a comprehensive theoretical insight. Could the authors clarify why they classify it as a theoretical analysis? How does this analysis provide deeper understanding beyond the mere derivation of terms?
>
> Please refer our responses to W2. Thank you.
>
> ### References
>
> [1] Wei Jin, Yao Ma, Xiaorui Liu, Xianfeng Tang, Suhang Wang, and Jiliang Tang. Graph Structure
> Learning for Robust Graph Neural Networks. In Proceedings of the 26th ACM SIGKDD In-
> ternational Conference on Knowledge Discovery & Data Mining, pp. 66–74, Virtual Event CA
> USA, August 2020. ACM. ISBN 978-1-4503-7998-4. doi: 10.1145/3394486.3403049.
>
> [2] Ruijia Wang, Shuai Mou, Xiao Wang, Wanpeng Xiao, Qi Ju, Chuan Shi, and Xing Xie. Graph
> Structure Estimation Neural Networks. In Proceedings of the Web Conference 2021, pp. 342–
> 353, Ljubljana Slovenia, April 2021. ACM. ISBN 978-1-4503-8312-7. doi: 10.1145/3442381.
> 3449952
>
> [3] Donghan Yu, Ruohong Zhang, Zhengbao Jiang, Yuexin Wu, and Yiming Yang. Graph-Revised
> Convolutional Network, December 2020. arXiv:1911.07123 [cs, stat].
>
> [4] Yu Chen, Lingfei Wu, and Mohammed Zaki. Iterative Deep Graph Learning for
> Graph Neural Networks: Better and Robust Node Embeddings. In Advances in
> Neural Information Processing Systems, volume 33, pp. 19314–19326. Curran Asso-
> ciates, Inc., 2020.

---

> ### Author Response · Authors · 2024-11-28
>
> Please let me know if our responses have addressed your concerns. Thank you!

---

### Official Review · Reviewer_XwUm · 2024-11-04

**Soundness:** 2
**Presentation:** 3
**Contribution:** 2
**Rating:** 5
**Confidence:** 5

**Summary:**

This paper summarizes the designs of graph structure learning (GSL) methods in graph neural networks (GNNs) using the proposed GSL framework. The author proposes to theoretically assess the effectiveness of the GSL methods by using mutual information. Both the empirical experiment results and the theoretical analysis conducted in this paper suggest that GSL methods do not enhance GNN performance when evaluated under the same GSL bases, that is, the representation used for GSL and hyperparameter tuning.

**Strengths:**

1. The paper is clearly written and is generally easy to follow.

2. The paper conducts extensive empirical evaluations.

3. The paper conducts theoretical analysis, which leads to a similar conclusion as the empirical evaluations.

4. The conclusion obtained is interesting and can be helpful in designing efficient graph neural networks.

**Weaknesses:**

1. Only the node-level learning task, i.e., node classification, is studied in this work. However, GSL is also widely used for graph-level learning tasks, such as graph classification.

2. Only small-scale datasets are studied in this work.

3. GSL methods are also widely used to improve the robustness of GNNs by purifying the perturbed graph structures. It would be interesting to see whether the observation and conclusion hold on adversarially perturbed graphs.

4. Existing works suggest GSL algorithms achieve the best results in scenarios with fewer labels available [1]. However, this work only studied the setting of splitting 50%/25%/25% of the nodes in train/validation/test sets. This data split setting differs from what is used in some of the GSL papers evaluated. I would suggest the author study the impact of label ratio on the effectiveness of the GSL methods.

[1] Li, Zhixun, et al. "GSLB: the graph structure learning benchmark." NeurIPS 2023.

**Questions:**

1. Does the conclusion hold for large-scale graphs, such as ogbn-arxiv?

2. Does the conclusion hold for GNNs with GSL for graph-level learning tasks, such as those in [1]?

3. Does the conclusion hold for node classification when the graph structure, node features, or class labels are adversarially perturbed as the settings in [1, 2]?

4. How does the label ratio impact the effectiveness of GSL methods? Can you make the same conclusion when fewer nodes are labeled?

[1] Li, Zhixun, et al. "GSLB: the graph structure learning benchmark." NeurIPS 2023.

[2] Zhiyao, Zhou, et al. "Opengsl: A comprehensive benchmark for graph structure learning." NeurIPS 2023.

---

> ### Author Response · Authors · 2024-11-22
> **Part (1/3)**
>
> ### W1: Only the node-level learning task, i.e., node classification, is studied in this work. However, GSL is also widely used for graph-level learning tasks, such as graph classification.
>
> Thanks for your constructive comments. In our submission, we only test the performance on node classification because node classification has been widely used to quantify the performance of GNN on graph representation learning [3,4]. In addition to the node classification experiments, we further investigate whether GSL consistently improves GNN performance in graph classification. Specifically, we conduct ablation experiments by replacing the GSL graph with the original graph, following the methodology outlined in [1]. As shown in 15, removing GSL from $4$ state-of-the-art GNNs, including ProGNN [5], GEN [6], GRCN [7], and IDGL [8], results in significantly reduced training time. At the same time, the GNN performance remains comparable to that of the GSL-enhanced counterparts. This suggests that GSL does not consistently enhance GNN performance in graph classification. Due to page limitations, we only tested a few methods in this paper. We believe it would be valuable to explore additional state-of-the-art methods, datasets, and theoretical justifications for the effectiveness of GSL in graph classification in future work.
>
> | Model            | Cora AUC     | Cora Time | PubMed AUC   | PubMed Time | CiteSeer Acc | CiteSeer Time |
> | ---------------- | ------------ | --------- | ------------ | ----------- | ------------ | ------------- |
> | ProGNN           | 76.28 ± 0.52 | 959s      | OOM          | -           | 67.14 ± 0.23 | 1776s         |
> | ProGNN, w/o. GSL | 78.96 ± 0.64 | 30s       | 75.80 ± 0.95 | 326s        | 67.24 ± 1.48 | 44s           |
> | GEN              | 79.88 ± 0.93 | 219s      | OOM          | -           | 66.98 ± 1.28 | 320s          |
> | GEN, w/o. GSL    | 78.32 ± 1.21 | 3s        | 76.94 ± 0.40 | 47s         | 64.66 ± 1.46 | 3s            |
> | GRCN             | 83.04 ± 0.33 | 56s       | 74.55 ± 0.96 | 249s        | 70.85 ± 0.87 | 113s          |
> | GRCN, w/o. GSL   | 71.82 ± 0.61 | 9s        | 74.18 ± 0.63 | 28s         | 58.33 ± 0.17 | 24s           |
> | IDGL             | 83.32 ± 0.59 | 144s      | OOM          | -           | 70.57 ± 0.26 | 330s          |
> | IDGL, w/o. GSL   | 83.32 ± 0.59 | 129s      | OOM          | -           | 71.12 ± 0.31 | 401s          |
>
> ### W2: Only small-scale datasets are studied in this work.
>
> The reason why we only conduct our experiments only on small-scale graph is because most GSL methods face the issue of high computation complexity. As shown in Section 4.3 complexity analysis in our submission, the total additional complexity of GSL is $O(|\mathcal{V}|^2F+|\mathcal{V}|F^2)$. Compared with the complexity in normal GCN $O(|\mathcal{E}|F+|\mathcal{V}|^2)$ [9], this additional complexity $O((|\mathcal{V}|^2-|\mathcal{E}|F)$ adds tremendous training time and grows exponentially with the number of nodes in graphs. To address this issue, further adaptations in node sampling, mini-batch training, or graph partitioning are required, which can vary considerably within our framework. Consequently, recent GSL studies [1, 2, 5, 6, 7, 8] have primarily focused on small-scale graphs. Nonetheless, it remains an intriguing question to explore the effectiveness of GSL when considering these additional adaptations. Thank you for your insightful review.

---

> ### Author Response · Authors · 2024-11-22
> **Part (2/3)**
>
> ### W3: GSL methods are also widely used to improve the robustness of GNNs by purifying the perturbed graph structures. It would be interesting to see whether the observation and conclusion hold on adversarially perturbed graphs.
>
> We further conduct experiments to investigate the robustness of GSL under noisy graph structures, by randomly adding or deleting edges in graphs following [1]. As shown in Table 16 and Table 17, we randomly add or remove $[0\%, 10\%,\dots, 80\%]$ edges in original graphs. The results show that, generally, GSL-enhanced GNNs are more robust to a higher ratio of noises in graph structures, which is consistent as the results in [1]. There are still some cases where GSL cannot improve the model robustness, such as GAug on CiteSeer. It will be interesting to empirically and theoretically investigate the robustness of GSL-enhanced models in the future. Thanks for your valuable comments.
>
>
>
> **Ablation study on model robustness by adding edges in GSL-enhanced methods.**
>
> | Use GSL  | Method | Dataset  | 0%    | 10%   | 20%   | 30%   | 40%   | 50%   | 60%   | 80%   |
> | -------- | ------ | -------- | ----- | ----- | ----- | ----- | ----- | ----- | ----- | ----- |
> | w/o. GSL | Gaug   | CiteSeer | 72.34 | 68.85 | 67.12 | 62.87 | 65.03 | 62.22 | 60.32 | 56.58 |
> | w. GSL   | Gaug   | CiteSeer | 72.79 | 71.62 | 68.08 | 63.96 | 63.68 | 62.08 | 60.14 | 56.78 |
> | w/o. GSL | Gaug   | Cora     | 81.73 | 76.83 | 74.07 | 70.53 | 68.67 | 67.70 | 67.42 | 54.93 |
> | w. GSL   | Gaug   | Cora     | 82.43 | 79.14 | 77.86 | 73.54 | 72.54 | 70.88 | 69.94 | 59.83 |
> | w/o. GSL | IDGL   | CiteSeer | 73.13 | 68.55 | 66.45 | 64.33 | 64.55 | 59.66 | 60.66 | 57.59 |
> | w. GSL   | IDGL   | CiteSeer | 73.26 | 71.11 | 68.62 | 67.83 | 65.79 | 63.89 | 63.17 | 60.23 |
> | w/o. GSL | IDGL   | Cora     | 82.43 | 78.65 | 78.18 | 77.98 | 74.33 | 74.53 | 72.36 | 65.23 |
> | w. GSL   | IDGL   | Cora     | 84.12 | 82.86 | 81.28 | 79.79 | 77.13 | 76.87 | 75.42 | 69.17 |
> | w/o. GSL | GRCN   | CiteSeer | 69.55 | 61.11 | 61.36 | 56.39 | 56.67 | 53.87 | 51.62 | 52.08 |
> | w. GSL   | GRCN   | CiteSeer | 73.21 | 69.89 | 67.87 | 64.94 | 63.56 | 61.53 | 60.36 | 56.53 |
> | w/o. GSL | GRCN   | Cora     | 81.66 | 74.89 | 72.01 | 69.98 | 66.76 | 66.19 | 61.85 | 59.63 |
> | w. GSL   | GRCN   | Cora     | 84.64 | 81.49 | 77.37 | 76.34 | 74.27 | 72.13 | 71.53 | 68.38 |
>
> **Ablation study on model robustness by deleting edges in GSL-enhanced methods.**
>
> | Use GSL    | Method | Dataset   | 0%    | 10%   | 20%   | 30%   | 40%   | 50%   | 60%   | 80%   |
> |------------|--------|-----------|-------|-------|-------|-------|-------|-------|-------|-------|
> | w/o. GSL  | Gaug  | CiteSeer  | 72.34 | 71.72 | 70.83 | 69.65 | 68.52 | 67.75 | 65.65 | 65.22 |
> | w. GSL    | Gaug  | CiteSeer  | 72.79 | 70.97 | 71.88 | 69.86 | 68.66 | 70.48 | 68.42 | 65.72 |
> | w/o. GSL  | Gaug  | Cora      | 81.73 | 78.65 | 78.18 | 77.98 | 74.33 | 74.53 | 72.30 | 65.24 |
> | w. GSL    | Gaug  | Cora      | 82.40 | 79.87 | 79.05 | 78.83 | 77.54 | 77.16 | 75.31 | 68.91 |
> | w/o. GSL  | IDGL  | CiteSeer  | 73.13 | 71.13 | 72.42 | 70.40 | 69.19 | 67.07 | 67.59 | 65.67 |
> | w. GSL    | IDGL  | CiteSeer  | 73.26 | 72.45 | 71.83 | 72.85 | 70.87 | 69.05 | 68.80 | 67.73 |
> | w/o. GSL  | IDGL  | Cora      | 82.43 | 81.39 | 80.15 | 79.58 | 78.72 | 76.12 | 74.44 | 67.99 |
> | w. GSL    | IDGL  | Cora      | 84.14 | 82.12 | 81.87 | 80.96 | 80.53 | 79.12 | 77.25 | 71.63 |
> | w/o. GSL  | GRCN  | CiteSeer  | 69.55 | 67.72 | 66.64 | 64.53 | 61.79 | 62.64 | 61.43 | 58.61 |
> | w. GSL    | GRCN  | CiteSeer  | 73.21 | 73.23 | 72.23 | 73.15 | 70.84 | 70.49 | 69.82 | 67.73 |
> | w/o. GSL  | GRCN  | Cora      | 81.66 | 76.51 | 74.39 | 74.71 | 71.64 | 71.77 | 68.02 | 60.63 |
> | w. GSL    | GRCN  | Cora      | 84.64 | 83.16 | 82.26 | 81.36 | 80.57 | 79.38 | 77.62 | 74.52 |

---

> ### Author Response · Authors · 2024-11-22
> **Part (3/3)**
>
> ### W4: Existing works suggest GSL algorithms achieve the best results in scenarios with fewer labels available [1]. However, this work only studied the setting of splitting 50\%/25\%/25\% of the nodes in train/validation/test sets. This data split setting differs from what is used in some of the GSL papers evaluated. I would suggest the author study the impact of label ratio on the effectiveness of the GSL methods.
>
> We further conduct an ablation study on the performance of GSL-enhanced methods. As shown in Table 18, the performance of these GSL-enhanced methods is comparable to their counterparts without GSL when there are $20$ labels per class. However, as we decrease the supervision ratio (such as in scenarios with $5$ or $3$ labels per class) the GSL-enhanced methods demonstrate improved performance compared to those without GSL. These results indicate that GSL can effectively enhance GNN performance in settings with low supervision. Thanks for your suggestions. We have revised the paper to include this valuable finding.
>
> | Method | Dataset  | Use GSL  | 20 labels | 10 labels | 5 labels | 3 labels |
> | ------ | -------- | -------- | --------- | --------- | -------- | -------- |
> | Gaug   | PubMed   | w/o. GSL | 79.68     | 72.87     | 67.66    | 63.30    |
> |        |          | w. GSL   | 78.73     | 75.48     | 69.84    | 65.90    |
> |        | CiteSeer | w/o. GSL | 72.30     | 64.62     | 57.68    | 51.53    |
> |        |          | w. GSL   | 72.79     | 67.02     | 58.38    | 54.37    |
> |        | Cora     | w/o. GSL | 81.71     | 74.02     | 65.67    | 60.13    |
> |        |          | w. GSL   | 82.48     | 76.12     | 69.46    | 65.97    |
> | Grcn   | Mines.   | w/o. GSL | 71.61     | 67.11     | 61.66    | 61.31    |
> |        |          | w. GSL   | 71.15     | 64.72     | 59.47    | 58.85    |
> |        | Cora     | w/o. GSL | 81.66     | 72.52     | 67.88    | 64.09    |
> |        |          | w. GSL   | 84.60     | 81.74     | 76.85    | 75.42    |
> |        | CiteSeer | w/o. GSL | 69.55     | 59.35     | 55.66    | 51.72    |
> |        |          | w. GSL   | 72.30     | 70.28     | 69.54    | 68.63    |
> | Idgl   | CiteSeer | w/o. GSL | 73.13     | 64.62     | 56.62    | 50.79    |
> |        |          | w. GSL   | 73.26     | 66.08     | 62.69    | 55.61    |
> |        | Cora     | w/o. GSL | 82.43     | 75.85     | 69.21    | 64.47    |
> |        |          | w. GSL   | 84.19     | 78.33     | 73.46    | 69.94    |
>
> ### Q1:
>
> See the above responses to W2.
>
> ### Q2:
>
> See the above responses to W1.
>
> ### Q3:
>
> See the above responses to W3.
>
> ### Q4:
>
> See the above responses to W4.
>
> ### Reference
>
> [1] Li, Zhixun, et al. "GSLB: the graph structure learning benchmark." NeurIPS 2023.
>
> [2] Zhiyao, Zhou, et al. "Opengsl: A comprehensive benchmark for graph structure learning." NeurIPS 2023.
>
> [3] Khoshraftar S, An A. A survey on graph representation learning methods[J]. ACM Transactions on Intelligent Systems and Technology, 2024, 15(1): 1-55.
>
> [4] Ju W, Fang Z, Gu Y, et al. A comprehensive survey on deep graph representation learning[J]. Neural Networks, 2024: 106207.
>
> [5] Wei Jin, Yao Ma, Xiaorui Liu, Xianfeng Tang, Suhang Wang, and Jiliang Tang. Graph Structure
> Learning for Robust Graph Neural Networks. In Proceedings of the 26th ACM SIGKDD In-
> ternational Conference on Knowledge Discovery & Data Mining, pp. 66–74, Virtual Event CA
> USA, August 2020. ACM. ISBN 978-1-4503-7998-4. doi: 10.1145/3394486.3403049.
>
> [6] Ruijia Wang, Shuai Mou, Xiao Wang, Wanpeng Xiao, Qi Ju, Chuan Shi, and Xing Xie. Graph
> Structure Estimation Neural Networks. In Proceedings of the Web Conference 2021, pp. 342–
> 353, Ljubljana Slovenia, April 2021. ACM. ISBN 978-1-4503-8312-7. doi: 10.1145/3442381.
> 3449952
>
> [7] Donghan Yu, Ruohong Zhang, Zhengbao Jiang, Yuexin Wu, and Yiming Yang. Graph-Revised
> Convolutional Network, December 2020. arXiv:1911.07123 [cs, stat].
>
> [8] Yu Chen, Lingfei Wu, and Mohammed Zaki. Iterative Deep Graph Learning for
> Graph Neural Networks: Better and Robust Node Embeddings. In Advances in
> Neural Information Processing Systems, volume 33, pp. 19314–19326. Curran Asso-
> ciates, Inc., 2020.
>
> [9] Derrick Blakely, Jack Lanchantin, and Yanjun Qi. Time and space complexity of graph convolutional
> networks. Accessed on: Dec, 31:2021, 2021.

---

> ### Author Response · Authors · 2024-11-28
>
> Please let me know if our responses have addressed your concerns. Thank you!

---

> ### Author Response · Authors · 2024-12-01
>
> We have added additional results for W3, where we investigate model robustness under noisy features. The findings indicate that as the amount of feature noise increases, GSL tends to outperform baseline GNNs. Please let us know if your concerns have been addressed. Thank you!
>
> | Use GSL    | Method | Dataset   | 0%    | 10%   | 20%   | 30%   | 40%   | 50%   | 60%   | 80%   |
> |------------|--------|-----------|-------|-------|-------|-------|-------|-------|-------|-------|
> | w/o. GSL  | Gaug   | CiteSeer  | 72.34 | 69.83 | 69.37 | 69.72 | 67.24 | 65.87 | 62.67 | 57.07 |
> | w. GSL    | Gaug   | CiteSeer  | 72.79 | 71.14 | 71.27 | 69.90 | 67.51 | 67.98 | 63.56 | 58.09 |
> | w/o. GSL  | Gaug   | Cora      | 81.73 | 78.96 | 80.80 | 78.86 | 77.84 | 76.87 | 73.27 | 67.40 |
> | w. GSL    | Gaug   | Cora      | 82.48 | 53.87 | 51.81 | 51.20 | 48.66 | 54.41 | 48.05 | 35.63 |
> | w/o. GSL  | Gaug   | PubMed    | 79.38 | 78.75 | 77.08 | 77.97 | 77.50 | 76.68 | 74.43 | 68.90 |
> | w. GSL    | Gaug   | PubMed    | 78.73 | 79.09 | 78.57 | 77.93 | 77.42 | 77.95 | 76.28 | 71.20 |
> | w/o. GSL  | IDGL   | CiteSeer  | 73.13 | 71.41 | 70.84 | 69.31 | 66.57 | 65.98 | 62.50 | 53.44 |
> | w. GSL    | IDGL   | CiteSeer  | 73.26 | 72.24 | 71.86 | 71.03 | 69.89 | 68.46 | 66.02 | 58.28 |
> | w/o. GSL  | IDGL   | Cora      | 82.43 | 80.72 | 81.24 | 79.11 | 78.53 | 78.08 | 73.57 | 68.73 |
> | w. GSL    | IDGL   | Cora      | 84.19 | 83.11 | 82.08 | 80.64 | 80.31 | 80.02 | 77.51 | 74.92 |
> | w/o. GSL  | GRCN   | CiteSeer  | 69.55 | 65.42 | 64.17 | 65.99 | 63.96 | 59.64 | 57.22 | 47.74 |
> | w. GSL    | GRCN   | CiteSeer  | 72.34 | 70.58 | 67.70 | 67.11 | 64.28 | 61.13 | 58.10 | 53.25 |
> | w/o. GSL  | GRCN   | Cora      | 81.66 | 76.40 | 76.86 | 76.48 | 74.32 | 74.86 | 72.98 | 64.71 |
> | w. GSL    | GRCN   | Cora      | 84.61 | 80.30 | 79.83 | 76.36 | 77.96 | 75.23 | 73.51 | 68.49 |
> | w/o. GSL  | GRCN   | PubMed    | 79.35 | 74.57 | 74.96 | 74.74 | 73.46 | 76.09 | 74.35 | 71.26 |
> | w. GSL    | GRCN   | PubMed    | 79.30 | 78.89 | 78.36 | 77.40 | 77.11 | 76.75 | 75.95 | 73.85 |

---

### Official Review · Reviewer_NzMd · 2024-11-04

**Soundness:** 3
**Presentation:** 3
**Contribution:** 3
**Rating:** 6
**Confidence:** 4

**Summary:**

This paper analyzes the graph structure learning methods from the perspectives of mutual information. The analysis suggests that no matter which type of GSL methods are used, after feeding the same GSL bases to the newly constructed graph, there is no mutual information gain compared to the original GSL bases. The paper then re-evaluates the graph structure learning using the same GSL bases. The results verify the the analysis.

**Strengths:**

1. This paper approaches GSL from an interesting perspective of information gain.
2. This paper reevaluated GSL methods using the same GSL bases and show the GSL can not consistently improve the performance of GNNs.
3. The paper is overall well written.

**Weaknesses:**

1. While this paper presents an interesting observation of GSL. It is not clear how this observation could help move GSL forward. The key observation is that GSL may not be the major contributor of model performance improvment but rather some other components are. This is a huge claim that tries to deny the previous efforts in GSL. I wonder if there is any certain scenario where GSL is still helpful?
2. In Table 2, there is a difference between the result of the current paper and result from Zhiyao et al. (2024). Is there an explanation for this?

**Questions:**

See waeknesses

---

> ### Author Response · Authors · 2024-11-22
>
> ### W1: 1) While this paper presents an interesting observation of GSL. It is not clear how this observation could help move GSL forward. The key observation is that GSL may not be the major contributor of model performance improvment but rather some other components are. 2) This is a huge claim that tries to deny the previous efforts in GSL. 3) I wonder if there is any certain scenario where GSL is still helpful?
>
> 1. Thanks for your valuable review. To help move GSL forward, in our submission, we demonstrate how different components in GSL influence model performance: 1) Figure 5 and Figure 6 show that using pertained node representations, e.g. MLP($X$,$A$), GCN($X$,$A$), or GCL($X$,$A$), as GSL bases significantly improve GNN performance on heterophilous datasets. 2) Figure 7 shows that kNN-based GSL graphs construction slightly outperforms other types of construction methods. 3) Figure 8 shows that that late fusion with no parameter sharing achieves the best performance compared with other types of fusion strategies. 4) Figure 9 shows that early fusion is better than late fusion for the fusion stage. Finally, we conduct extension experiments by removing GSL in GNN to show that it is the good quality of GSL bases instead of GSL graph construction that improves GNN performance, which is verified by extensive experiments that adding GSL to GNN baselines or removing GSL in SOTA GSL methods.
> 2. The reason why we make such a claim is because we have carefully reviewed most GSL methods and analyzed how they are different from each other in our proposed GSL framework as shown in Figure 2 and Table 3. Since it is impossible to implement all types of GSL methods in the experiments, we implement the most representative GSL components, which derive 360 combinations in our experiments. We believe these methods could cover the most representative GSL methods, plus with the SOTA-GSL experiments, which verifies our claim. Admittedly, we also revise our claim to make it more specific and accurate. Thank you.
> 3. We only show that GSL is not necessary for improving the quality of graph representation learning by node classification in our experiments. To see if GSL is still helpful in other scenarios, we conduct additional experiments on graph classification (Appendix E.7), perturbed graphs (Appendix E.8), and few-shot learning (Appendix E.9). Our results show that GSL is also unnecessary in graph classification, but it is more effective than baseline GNNs in perturbed graphs and few-shot learning.
>
> We have revised our paper. Thank you for your valuable feedback.
>
> ### W2: In Table 2, there is a difference between the result of the current paper and result from [1]. Is there an explanation for this?
>
> We follow the same setting as in [1]. In some datasets, we can get higher results than the reported results, such as GEN in pubmed and GRCN in amazon-ratings. While in other datasets, the results are slightly lower than the reported results, such as GAUG in citeseer and GRCN in roman-empire. Thanks for your suggestions, we have revised the table to get a higher performance of GSL-based SOTA methods. As shown in Table 1, this result still supports our conclusion that by removing GSL in SOTA methods we can still get comparable performance.
>
> Thank you again for your valuable feedback.
>
> ### References
>
> [1] Zhiyao Z, Zhou S, Mao B, et al. Opengsl: A comprehensive benchmark for graph structure learning[J]. Advances in Neural Information Processing Systems, 2024, 36.

---

> > ### Comment · Reviewer_NzMd · 2024-11-28
> > **Thanks for the response**
> >
> > I have read the response. I would like to keep my original score.

---

### Official Review · Reviewer_4unK · 2024-11-10

**Soundness:** 3
**Presentation:** 2
**Contribution:** 3
**Rating:** 5
**Confidence:** 5

**Summary:**

Paper summarizes the utility of Graph Structured Learning (GSL): Does inferring edges among nodes (when some edges or no edges are given) help classification tasks? For instance, given many examples, one can develop an MLP that can infer on all examples in parallel. Alternatively, one can induce edges among example pairs, and run a GNN on the features+induced graphs.

Authors show that: there are no cases where GSL is useful. In summary, either run an MLP or a GNN on the original graph. If edges are homophonous, then just use GNN on the real edges (no induced edges). If the edges are heterophilous, just ignore the edges altogether and just run an MLP.

**Strengths:**

* It is nice to know this summary [do not induce any edges!]
* The paper's arguments are easy to follow [though writing can be improved]

**Weaknesses:**

The mean weakness of the paper (reason for my rejection) is:

* The arguments are too general: GSL does not add information
* The construction is too-specific.

From reading the paper, I can only remove some generality from the main argument:

* GSL is not useful for **classification** settings where the graph construction function is set to non-learnable KNN.

Crucially, their edge function is the kNN graph. Their GNN is similar to GCN [Kipf&Welling], i.e., one that averages node features with their neighbors, at every layer. Using kNN edges and GCN should work well if neighbor nodes (i.e. k-nearest) have the same class as the center node. This assumption is not met for heterophilous graphs.

None of the findings seem to be surprising. I will make a few notes

* Why use GCN if you are connecting on-purpose heterophilous edges? Why not use something like MixHop which (promises to) handle heterophilous edges?

* Why use untrainable kNN?

Many methods show that graph structure is indeed useful for their application, including https://arxiv.org/abs/2007.12002 or https://arxiv.org/abs/1711.04043 -- the first has more sophisticated edge-creation function and the second is trainable

# Final note

I do like the paper, but I think it can be greatly simplified to one small argument: dont induce kNN graph use in a GCN, instead use MLP -- noting that kNN+GCN is one instantiation of GSL.

**Questions:**

* How did you calculate the mutual information? It seems that one must integrate over the input space.

---

> ### Author Response · Authors · 2024-11-22
> **Part (1/5)**
>
> ### W1: The arguments are too general: GSL does not add information.
>
> Thanks for your valuable comments on improving our paper. We have revised the claim to make it more specific: Similarity-based GSL cannot improve GNN performance on node classification. The reason why we claim this in our submission is because:
>
> 1. Similarity-based GSL are the most dominant approaches to construct new graphs. As the representative methods listed in Table 3, 18 out of 22 GSL methods are similarity-based GSL, showing that the analysis of GSL in this paper has more generality.
> 2.  Node classification has been widely used to quantify the performance of GNN on graph representation learning [1,2].
>
> Admittedly, even though we tried our best to consider the most representative approaches in GSL in this paper, there are still some cases cannot be covered. Therefore, we have carefully revise the manuscript to make it more specific. Thank you!
>
> ### W2: The construction is too-specific.
>
> First, we admit that GSL has been developed for many years and many new methods are proposed. Since it is impossible to implement all the GSL methods in our paper, that's why we first propose a framework to de-composite each component of GSL and then implement the most representative GSL approaches for each component. As shown in Appendix D in our submission, our implementation includes: 4 GNN backbones: GCN, GAT, GraphSAGE, and SGC, 5 GSL bases: $X$, $\hat{A}X$, MLP($X$), GCN($X$,$A$), and GCL($X$,$A$), 3 GSL graph construction approaches: Cos-graph, Cos-node, and kNN, 3 view fusion methods:  $\{\mathcal{G}'\}$, $\{,\mathcal{G},\mathcal{G}'\},\theta_1=\theta_2$, $\{,\mathcal{G},\mathcal{G}'\},\theta_1\neq \theta_2$, and 2 types of fusion strategies: early fusion and late fusion. This implementation covers most GSL approaches as shown in Table 3. The combination of these components derives 360 different methods of GSL, which has never been explored in previous studies.
>
> Besides, to consider more sophisticated GSL methods, in our submission, we further conduct an ablation study by removing GSL components in 8 state-of-the-art (SOTA) GSL-based GNNs in Table 2, the results also support our claim that GSL cannot improve GNN performance on node classification.
>
> ### W3: Why use GCN if you are connecting on-purpose heterophilous edges? Why not use something like MixHop which (promises to) handle heterophilous edges?
>
> Thanks to your suggestions, we further implement ACMGNN [3], MixHop [4] in our experiments that incorporate GSL into GNN baselines. These experiments follow the same setup as described in Table 1. The results, presented in Table 13, demonstrate that, under fair comparison conditions, both ACMGNN and MixHop outperform their GNN+GFS counterparts. This suggests that adding GSL to these heterophily-oriented GNNs cannot improve model performance on node classification.

---

> ### Author Response · Authors · 2024-11-22
> **Part (2/5)**
>
> | Model  | Construct | Fusion                         | Param Sharing          | Mines.         | Roman.         | Amazon.        | Tolokers       | Questions      | Squirrel       | Chameleon      | Actor          | Texas          | Cornell        | Wisconsin      | Cora           | CiteSeer       | PubMed         | Rank     |
> | ------ | --------- | ------------------------------ | ---------------------- | -------------- | -------------- | -------------- | -------------- | -------------- | -------------- | -------------- | -------------- | -------------- | -------------- | -------------- | -------------- | -------------- | -------------- | -------- |
> | MLP    | -         | -                              | -                      | 79.55±1.23     | 65.45±0.99     | 46.65±0.83     | 75.94±1.38     | 74.92±1.39     | 39.29±2.22     | 43.57±4.18     | **35.40±1.38** | 80.46±6.44     | 73.78±7.34     | 85.88±7.78     | 87.97±1.80     | **76.68±2.10** | 87.39±2.18     | 2.93     |
> | ACMGNN | -         | -                              | -                      | **90.56±1.03** | **84.86±0.73** | **52.07±1.72** | **84.41±1.12** | **77.72±1.59** | **41.53±2.43** | **44.65±4.43** | 34.86±1.22     | **82.62±5.97** | **75.68±8.99** | **87.65±7.15** | **88.23±1.81** | 76.63±2.34     | **89.37±0.56** | **1.21** |
> | ACMGNN | cos-graph | $\{\mathcal{G}'\}$             | -                      | 47.36±3.47     | 60.97±0.76     | 41.50±0.75     | 70.21±1.51     | 67.32±1.37     | 38.12±1.92     | 39.90±3.64     | 33.43±0.95     | 59.80±6.99     | 59.46±8.35     | 71.57±6.68     | 77.47±2.41     | 73.68±0.97     | 87.19±0.38     | 7.64     |
> | ACMGNN | cos-graph | $\{\mathcal{G},\mathcal{G}'\}$ | $\theta_1=\theta_2$    | 52.74±5.22     | 51.18±2.12     | 33.11±1.38     | 69.06±4.65     | 62.30±3.23     | 31.58±4.39     | 38.79±4.73     | 29.06±2.60     | 54.10±7.59     | 59.19±8.87     | 70.39±9.58     | 59.74±1.87     | 65.17±1.94     | 79.53±1.69     | 9.86     |
> | ACMGNN | cos-graph | $\{\mathcal{G},\mathcal{G}'\}$ | $\theta_1\neq\theta_2$ | 87.46±1.02     | 74.63±0.76     | 49.35±0.58     | 81.63±0.87     | 73.84±1.41     | 38.54±1.89     | 41.16±4.18     | 34.23±0.98     | 67.67±5.97     | 70.00±5.90     | 80.78±5.21     | 80.83±1.84     | 73.43±1.47     | 88.98±0.47     | 3.64     |
> | ACMGNN | cos-node  | $\{\mathcal{G}'\}$             | -                      | 52.83±3.52     | 61.26±0.62     | 42.47±0.53     | 74.14±1.14     | 72.23±1.36     | 38.23±1.97     | 40.77±3.68     | 34.74±0.90     | 61.45±6.13     | 63.51±5.87     | 74.31±6.43     | 75.84±2.93     | 73.05±1.18     | 87.22±0.41     | 6.21     |
> | ACMGNN | cos-node  | $\{\mathcal{G},\mathcal{G}'\}$ | $\theta_1=\theta_2$    | 52.74±5.22     | 51.18±2.12     | 33.11±1.38     | 69.06±4.65     | 62.30±3.23     | 31.58±4.39     | 38.79±4.73     | 29.06±2.60     | 54.10±7.59     | 59.19±8.87     | 70.39±9.58     | 59.74±1.87     | 65.17±1.94     | 79.53±1.69     | 9.86     |
> | ACMGNN | cos-node  | $\{\mathcal{G},\mathcal{G}'\}$ | $\theta_1\neq\theta_2$ | 87.80±0.97     | 73.55±0.51     | 49.04±0.57     | 80.74±0.92     | 74.11±1.40     | 39.19±2.12     | 40.28±4.30     | 34.19±1.16     | 69.86±5.56     | 69.46±7.21     | 80.39±5.23     | 80.33±1.90     | 73.31±1.26     | 88.94±0.36     | 4.07     |
> | ACMGNN | kNN       | $\{\mathcal{G}'\}$             | -                      | 51.68±3.38     | 60.86±0.87     | 41.68±0.95     | 71.31±0.64     | 69.56±1.41     | 38.58±1.96     | 40.56±2.34     | 34.88±0.77     | 62.51±6.16     | 62.70±5.95     | 76.47±4.43     | 75.99±2.85     | 70.20±1.51     | 87.20±0.45     | 6.64     |
> | ACMGNN | kNN       | $\{\mathcal{G},\mathcal{G}'\}$ | $\theta_1=\theta_2$    | 52.74±5.22     | 51.18±2.12     | 33.11±1.38     | 69.06±4.65     | 62.30±3.23     | 31.58±4.39     | 38.79±4.73     | 29.06±2.60     | 54.10±7.59     | 59.19±8.87     | 70.39±9.58     | 59.74±1.87     | 65.17±1.94     | 79.53±1.69     | 9.86     |
> | ACMGNN | kNN       | $\{\mathcal{G},\mathcal{G}'\}$ | $\theta_1\neq\theta_2$ | 87.59±0.88     | 73.21±0.63     | 49.06±0.53     | 81.34±0.85     | 73.95±1.35     | 39.18±2.18     | 41.70±3.71     | 34.67±1.11     | 68.48±5.78     | 68.92±5.87     | 80.20±3.13     | 80.46±2.26     | 73.14±1.31     | 88.87±0.51     | 4.07     |

---

> ### Author Response · Authors · 2024-11-22
> **Part (3/5)**
>
> | Model  | Construct | Fusion                         | Param Sharing          | Mines.         | Roman.         | Amazon.        | Tolokers       | Questions      | Squirrel       | Chameleon      | Actor          | Texas          | Cornell        | Wisconsin      | Cora           | CiteSeer       | PubMed         | Rank     |
> | ------ | --------- | ------------------------------ | ---------------------- | -------------- | -------------- | -------------- | -------------- | -------------- | -------------- | -------------- | -------------- | -------------- | -------------- | -------------- | -------------- | -------------- | -------------- | -------- |
> | MLP    | -         | -                              | -                      | 79.55±1.23     | 65.45±0.99     | 46.65±0.83     | 75.94±1.38     | 74.92±1.39     | 39.29±2.22     | **43.57±4.18** | **35.40±1.38** | **80.46±6.44** | **73.78±7.34** | **85.88±7.78** | **87.97±1.80** | **76.68±2.10** | 87.39±2.18     | 2.29     |
> | MixHop | -         | -                              | -                      | **90.10±5.59** | **81.70±0.89** | **50.95±0.71** | **84.56±1.19** | **77.66±1.24** | **41.22±2.66** | 43.11±4.73     | 33.59±1.23     | 72.54±8.98     | 62.43±9.54     | 75.88±8.27     | 87.76±1.94     | 76.51±1.93     | **89.42±0.81** | **1.86** |
> | MixHop | cos-graph | $\{\mathcal{G}'\}$             | -                      | 64.75±4.59     | 51.83±0.53     | 41.47±2.00     | 68.78±1.94     | 71.45±1.38     | 37.75±2.41     | 37.79±2.10     | 31.77±1.75     | 55.72±6.39     | 60.27±5.85     | 70.20±4.60     | 84.42±1.35     | 74.20±0.83     | 88.74±0.29     | 8.21     |
> | MixHop | cos-graph | $\{\mathcal{G},\mathcal{G}'\}$ | $\theta_1=\theta_2$    | 54.22±10.75    | 63.50±0.86     | 44.21±1.36     | 74.22±2.21     | 70.64±1.32     | 37.16±1.34     | 39.06±3.08     | 32.24±1.33     | 58.16±9.18     | 66.22±5.59     | 73.73±7.80     | 65.14±2.62     | 68.66±1.24     | 86.63±0.51     | 7.54     |
> | MixHop | cos-graph | $\{\mathcal{G},\mathcal{G}'\}$ | $\theta_1\neq\theta_2$ | 84.71±1.19     | 55.41±1.63     | 43.37±0.75     | 74.41±1.33     | 69.63±2.03     | 37.64±2.19     | 38.71±4.36     | 31.73±1.77     | 61.13±7.96     | 61.35±7.10     | 75.29±6.00     | 85.42±1.21     | 74.57±1.34     | 88.16±0.46     | 6.50     |
> | MixHop | cos-node  | $\{\mathcal{G}'\}$             | -                      | 60.56±7.08     | 51.74±0.68     | 42.71±0.97     | 74.27±1.84     | 72.83±1.12     | 38.35±1.99     | 38.88±3.00     | 33.05±1.04     | 58.42±6.52     | 60.27±5.98     | 71.57±4.91     | 83.22±1.16     | 74.11±1.12     | 88.23±0.45     | 6.71     |
> | MixHop | cos-node  | $\{\mathcal{G},\mathcal{G}'\}$ | $\theta_1=\theta_2$    | 54.22±10.75    | 63.50±0.86     | 44.21±1.36     | 74.22±2.21     | 70.64±1.32     | 37.16±1.34     | 39.06±3.08     | 32.24±1.33     | 58.16±9.18     | 66.22±5.59     | 73.73±7.80     | 65.14±2.62     | 68.66±1.24     | 86.63±0.51     | 7.64     |
> | MixHop | cos-node  | $\{\mathcal{G},\mathcal{G}'\}$ | $\theta_1\neq\theta_2$ | 85.43±0.57     | 55.95±2.35     | 44.15±0.59     | 76.54±0.91     | 72.03±2.45     | 37.47±2.07     | 39.52±3.33     | 32.50±1.10     | 60.61±8.73     | 62.97±6.75     | 75.10±6.20     | 85.36±0.89     | 74.68±1.13     | 88.18±0.52     | 4.79     |
> | MixHop | kNN       | $\{\mathcal{G}'\}$             | -                      | 59.50±6.26     | 50.39±0.72     | 42.07±0.93     | 70.49±1.70     | 69.57±1.32     | 38.07±1.72     | 38.76±2.91     | 33.23±1.30     | 59.25±4.49     | 57.30±6.96     | 69.22±7.22     | 83.99±1.28     | 74.96±1.18     | 87.99±0.40     | 8.00     |
> | MixHop | kNN       | $\{\mathcal{G},\mathcal{G}'\}$ | $\theta_1=\theta_2$    | 54.22±10.75    | 63.50±0.86     | 44.21±1.36     | 74.22±2.21     | 70.64±1.32     | 37.16±1.34     | 39.06±3.08     | 32.24±1.33     | 58.16±9.18     | 66.22±5.59     | 73.73±7.80     | 65.14±2.62     | 68.66±1.24     | 86.63±0.51     | 7.54     |
> | MixHop | kNN       | $\{\mathcal{G},\mathcal{G}'\}$ | $\theta_1\neq\theta_2$ | 85.53±0.50     | 57.48±1.98     | 43.28±0.68     | 77.24±1.61     | 70.34±1.76     | 38.15±2.01     | 40.12±3.76     | 32.30±1.53     | 60.05±9.45     | 63.51±7.56     | 74.90±8.21     | 85.18±1.26     | 74.59±1.19     | 88.20±0.57     | 4.93     |

---

> ### Author Response · Authors · 2024-11-22
> **Part (4/5)**
>
> ### W4: Why use untrainable kNN?
>
> 1. We clarify that kNN is one implementation of similarity-based GSL: As shown in Table 1, we further include Cos-graph and Cos-node as the GSL graph construction approaches.
> 2. Only the results in Table 1 are conducted with untrainable GSL, while in Table 2 GSL graphs in SOTA methods are trainable.
>
> In addition to the non-learnable kNN, we further conduct experiments on trainable GSL graphs on $6$ GNN backbones as shown in Table 14. Specifically, we select the best-performing GSL variants, as shown in Tables 1 and 13, for each backbone GNN. The best-performing method is highlighted in bold, while the runner-up is highlighted in Italic. "OOM" refers to "out of memory." The results demonstrate the following
>
> 1. The average rank indicates that trainable GSL improves GNN performance on 5 out of 6 GNN backbones.
> 2. Although trainable GSL outperforms non-trainable GSL, it remains inferior to GNN backbones without GSL, indicating that GSL could be unnecessary in improving GNN performance on node classification.
>
> | **Model**     | **GSL Type**      | **Mines.**     | **Roman.**     | **Amazon.**    | **Tolokers**   | **Questions**  | **Cora**       | **CiteSeer**   | **PubMed**     | **Rank** |
> | ------------- | ----------------- | -------------- | -------------- | -------------- | -------------- | -------------- | -------------- | -------------- | -------------- | -------- |
> | **GCN**       | No GSL            | **90.07±5.79** | **81.46±1.25** | **50.89±1.16** | **84.61±0.99** | **77.68±1.10** | **87.97±1.51** | **76.75±2.30** | **89.47±0.64** | **1.19** |
> |               | Trainable GSL     | **90.07±0.58** | _78.76±0.46_   | **50.89±0.65** | **84.61±0.65** | OOM            | 84.92±1.51     | 74.89±1.13     | 88.66±0.45     | 2.31     |
> |               | Non-trainable GSL | 89.17±0.68     | 72.63±1.45     | 48.31±0.96     | 82.91±0.97     | _75.56±1.05_   | _85.69±1.73_   | _75.49±1.42_   | _88.72±0.71_   | 2.50     |
> | **SGC**       | No GSL            | **83.45±4.47** | **78.04±0.69** | **51.38±0.68** | **84.88±1.13** | **77.39±1.23** | **88.10±1.89** | **77.52±2.20** | **89.39±0.62** | **1.19** |
> |               | Trainable GSL     | **83.45±1.03** | _74.74±0.57_   | **51.38±0.57** | **84.88±0.65** | OOM            | 86.99±1.64     | 75.13±1.26     | 88.94±0.31     | 2.31     |
> |               | Non-trainable GSL | 79.03±3.76     | 67.84±1.87     | 47.93±0.94     | 78.09±1.84     | _75.46±1.43_   | _87.47±1.86_   | _76.36±1.27_   | _89.37±0.41_   | 2.50     |
> | **GraphSAGE** | No GSL            | _90.66±0.88_   | **85.02±0.97** | **52.93±0.83** | **83.31±1.12** | **75.95±1.41** | **88.13±1.77** | **76.65±2.00** | **89.18±0.65** | **1.31** |
> |               | Trainable GSL     | _90.66±0.58_   | _82.54±0.60_   | **52.93±0.59** | **83.31±0.50** | OOM            | 83.48±1.69     | 74.18±1.02     | 88.67±0.39     | 2.44     |
> |               | Non-trainable GSL | **90.67±0.66** | 79.02±1.21     | 52.10±0.84     | 82.17±0.89     | _75.38±0.96_   | _83.60±1.78_   | _74.39±1.35_   | _88.88±0.50_   | 2.25     |
> | **GAT**       | No GSL            | **90.41±1.34** | **84.51±0.84** | _52.00±2.84_   | **84.37±0.96** | **77.78±1.27** | **88.02±1.92** | **76.77±2.02** | **89.21±0.67** | **1.19** |
> |               | Trainable GSL     | **90.41±0.61** | _83.10±0.58_   | **52.10±0.62** | _84.35±0.56_   | OOM            | 86.23±1.58     | 74.39±1.14     | _88.13±0.56_   | 2.19     |
> |               | Non-trainable GSL | 89.96±0.79     | 77.23±1.63     | 49.79±0.72     | 82.78±0.95     | _76.67±1.13_   | _86.97±1.75_   | _75.20±1.55_   | 87.97±0.51     | 2.62     |
> | **ACMGNN**    | No GSL            | **90.56±1.03** | **84.86±0.73** | **52.07±1.72** | **84.41±1.12** | **77.72±1.59** | **88.23±1.81** | **76.63±2.34** | **89.37±0.56** | **1.06** |
> |               | Trainable GSL     | **90.56±0.63** | _81.90±0.71_   | _51.87±0.44_   | _84.40±0.79_   | OOM            | _81.16±1.81_   | _73.91±1.16_   | 88.55±0.39     | 2.19     |
> |               | Non-trainable GSL | 87.46±1.02     | 74.63±0.76     | 49.35±0.58     | 81.63±0.87     | _73.84±1.41_   | 80.83±1.84     | 73.43±1.47     | _88.98±0.47_   | 2.75     |
> | **MixHop**    | No GSL            | **90.10±5.59** | **81.70±0.89** | **50.95±0.71** | **84.56±1.19** | **77.66±1.24** | **87.76±1.94** | **76.51±1.93** | **89.42±0.81** | **1.12** |
> |               | Trainable GSL     | **90.10±0.52** | _79.07±0.75_   | **50.95±0.71** | _84.55±0.67_   | OOM            | 84.84±1.28     | 74.45±1.11     | _88.48±0.62_   | 2.25     |
> |               | Non-trainable GSL | 85.43±0.57     | 55.95±2.35     | 44.15±0.59     | 76.54±0.91     | _72.03±2.45_   | _85.36±0.89_   | _74.68±1.13_   | 88.18±0.52     | 2.62     |

---

> ### Author Response · Authors · 2024-11-22
> **Part (5/5)**
>
> ### W5: Many methods show that graph structure is indeed useful for their application, including GSL approaches with more sophisticated edge-creation functions [5] or trainable graphs [6].
>
> Thanks for your suggestions. We further conducted experiments on two GSL-enhanced GNNs, Grale [5] and MetricNN [6], using the same settings as in our previous experiments. As shown in Table 19, removing GSL from these methods resulted in improved model performance on most datasets, along with reduced training time. These findings further support the conclusion drawn in this paper that GSL is unnecessary for enhancing model performance in node classification.
>
> | Model                 | Questions AUC | Questions Time | Minesweeper AUC | Minesweeper Time | Roman-empire Acc | Roman-empire Time | Amazon-ratings Acc | Amazon-ratings Time | Tolokers AUC | Tolokers Time |
> | --------------------- | ------------- | -------------- | --------------- | ---------------- | ---------------- | ----------------- | ------------------ | ------------------- | ------------ | ------------- |
> | **Grale**             | 68.96 ± 0.23  | 320s           | 66.36 ± 0.08    | 31s              | 35.42 ± 0.57     | 129s              | 46.57 ± 0.37       | 145s                | 73.32 ± 0.72 | 110s          |
> | **Grale, w/o GSL**    | 68.34 ± 1.18  | 283s           | 62.42 ± 0.20    | 9s               | 64.90 ± 0.26     | 130s              | 48.38 ± 0.94       | 136s                | 74.49 ± 0.45 | 48s           |
> | **MetricNN**          | 64.28 ± 1.34  | 23s            | 68.51 ± 1.63    | 2.1s             | 38.55 ± 1.68     | 8.3s              | 42.10 ± 1.22       | 10s                 | 69.41 ± 5.50 | 5.4s          |
> | **MetricNN, w/o GSL** | 65.27 ± 0.86  | 11s            | 73.51 ± 0.01    | 2.1s             | 37.60 ± 2.26     | 4.3s              | 42.28 ± 1.11       | 5.8s                | 75.20 ± 0.48 | 3.4s          |
>
> For more scenarios, we conduct additional experiments to investigate the effectiveness of GSL under more settings: graph classification (Appendix E.7), perturbed graphs (Appendix E.8), and few-shot learning (Appendix E.9). We find that GSL is also unnecessary in graph classification, but it is more effective than baseline GNNs in perturbed graphs and few-shot learning. We have revised our paper. Thank you.
>
> ### Q1: How did you calculate the mutual information? It seems that one must integrate over the input space.
>
> In the Preliminary Section, we describe how mutual information is calculated: we measure the mutual information $I(X;Y)$ based on entropy estimation from k-nearest neighbors distances following [7,8,9]. Here, we further clarify the calculation of mutual information.
>
> Given a continuous variable $X$ and a discrete variable $Y$, we start with $N$ data points $(x_i, y_i)$. To compute the mutual information $I(X;Y)$, for each point $i$, we find the k-th nearest neighbor in $X$-space, considering only points where $y = y_i$. We denote the distance to this neighbor as $d$. Next, we count the total number of points in the dataset within this distance $d$, denoting this count as $m_i$, and the total number of points where $y = y_i$ as $N_{y_i}$. We then use the formula
> $$I_i = \psi(N) - \psi(N_{y_i}) + \psi(k) - \psi(m_i)$$
> where $\psi$ is the digamma function [10], to calculate the contribution of each point to the mutual information. The overall mutual information is then given by the average of all $I_i$:
> $$I(X, Y) = \frac{1}{N} \sum_{i=1}^{N} I_i$$
> This method avoids binning [11] and works efficiently for both scalar and vector data, while addressing a small k-balancing detail and noise.
>
> Thanks for your review, we have carefully revised our manuscript with respect to your concerns.

---

> ### Author Response · Authors · 2024-11-22
>
> ### References
>
> [1] Khoshraftar S, An A. A survey on graph representation learning methods[J]. ACM Transactions on Intelligent Systems and Technology, 2024, 15(1): 1-55.
>
> [2] Ju W, Fang Z, Gu Y, et al. A comprehensive survey on deep graph representation learning[J]. Neural Networks, 2024: 106207.
>
> [3] Luan S, Hua C, Lu Q, et al. Revisiting heterophily for graph neural networks[J]. Advances in neural information processing systems, 2022, 35: 1362-1375.
>
> [4] Abu-El-Haija S, Perozzi B, Kapoor A, et al. Mixhop: Higher-order graph convolutional architectures via sparsified neighborhood mixing[C]//international conference on machine learning. PMLR, 2019: 21-29.
>
> [5] Halcrow J, Mosoi A, Ruth S, et al. Grale: Designing networks for graph learning[C]//Proceedings of the 26th ACM SIGKDD international conference on knowledge discovery \& data mining. 2020: 2523-2532.
>
> [6] Garcia V, Bruna J. Few-shot learning with graph neural networks[J]. arXiv preprint arXiv:1711.04043, 2017.
>
> [7] Alexander Kraskov, Harald Stogbauer, and Peter Grassberger. Estimating mutual information. ¨ Physical Review E—Statistical, Nonlinear, and Soft Matter Physics, 69(6):066138, 2004.
>
> [8] Brian C Ross. Mutual information between discrete and continuous data sets. PloS one, 9(2): e87357, 2014.
>
> [9] Lyudmyla F Kozachenko and Nikolai N Leonenko. Sample estimate of the entropy of a random vector. Problemy Peredachi Informatsii, 23(2):9–16, 1987
>
> [10] Abramowitz, M.; Stegun, I. A., eds. (1972). "6.3 psi (Digamma) Function.". Handbook of Mathematical Functions with Formulas, Graphs, and Mathematical Tables (10th ed.). New York: Dover. pp. 258–259.
>
> [11] Kraskov A, Sto¨gbauer H, Grassberger P (2004) Estimating mutual information.
> Physical Review E 69: 066138.

---

> ### Author Response · Authors · 2024-11-28
>
> Please let me know if our responses have addressed your concerns. Thank you!

---

> > ### Comment · Reviewer_4unK · 2024-11-29
> >
> > You have significantly improved the paper given reviews. I will increase my score by 1. I will also watch the discussion among reviewers and if the consensus move towards accepting the paper, then I can upvote again. At this point, this is a negative-finding paper, and the novelty is harder to assess.

---

> > > ### Author Response · Authors · 2024-12-01
> > >
> > > We greatly appreciate the time you took for a detailed review and for raising your score. In contrast to many current GSL approaches that introduce increasingly complex model structures with numerous hyperparameters and extensive training times for minor improvements, we have re-evaluated the contribution of GSL in GNNs through both empirical observations and theoretical analysis. Our extensive experiments demonstrate that under the same GSL bases, GSL-based methods cannot enhance GNN performance in node classification. To the best of our knowledge, this perspective has not been previously discussed and holds significant implications for the community.
> > >
> > > Again, thanks for your participation in the discussion. One of the other reviewers has given a positive rating, and we are currently awaiting responses from the remaining two reviewers. Welcome to upvote your score. Thank you.

---

### Author Response · Authors · 2024-11-25

We would like to express our gratitude to all the reviewers for their valuable feedback. Our paper proposes a general framework for a better understanding of Graph Structure Learning (GSL) in Graph Neural Networks (GNNs). We show that GSL cannot improve GNN performance for node classification tasks, supported by empirical observations and theoretical justifications. Our experiments thoroughly reassess the performance of GSL, revealing that adding GSL to GNN baselines (across 360 combinations) or removing GSL from state-of-the-art models has a negligible impact on node classification accuracy. While many approaches are trying to add more and more complex model designs in GNNs to get tiny improvement, we want to **do reduction in the model design by rethinking the necessity of the GSL**. Therefore, we think our work remains a high novelty in this area. We would like to clarify the following points during the rebuttal:

1. **Scope of Our Paper:** The primary focus of this paper is to study GNN performance on node classification tasks. This does not imply that we dismiss the broader GSL area. In fact, our findings also indicate that GSL could improve model robustness, an area worth investigating in the future.
2. **Additional Methods:** To explore stronger GNN backbones and more complex GSL methods, we conducted further experiments on two GNN baselines (MixHop[1] and ACMGNN[2], as shown in Table 13) and two GSL-based state-of-the-art methods (Grale[3] and MetricNN[4], as shown in Table 19). We also implemented a trainable GSL (Table 14). The results consistently support our original claim that GSL does not enhance model performance on node classification tasks.
3. **Additional Settings:**
(1) We further investigate the performance of GSL-based state-of-the-art methods on graph classification, as shown in Table 15. The results indicate that GSL does not consistently improve model performance in this context.
(2) We assess the robustness of GSL in Tables 16 and 17, where the results demonstrate that GSL can enhance the robustness of GNNs.
(3) We also conducted experiments on GSL under weakly supervised settings, showing that GSL-based methods outperform their counterparts without GSL in this scenario. However, the main focus of this paper is to determine whether GSL improves model performance on node classification, a question that our experiments have addressed. We believe that the additional settings warrant further investigation in the future.
4. **Improved Presentation:** We have carefully revised our paper to emphasize its contributions and enhance its rigor.

Thank you once again to all the reviewers. **We look forward to your response.**


**References**

[1] Abu-El-Haija S, Perozzi B, Kapoor A, et al. Mixhop: Higher-order graph convolutional architectures via sparsified neighborhood mixing[C]//international conference on machine learning. PMLR, 2019: 21-29.

[2] Luan S, Hua C, Lu Q, et al. Revisiting heterophily for graph neural networks[J]. Advances in neural information processing systems, 2022, 35: 1362-1375.

[3] Halcrow J, Mosoi A, Ruth S, et al. Grale: Designing networks for graph learning[C]//Proceedings of the 26th ACM SIGKDD international conference on knowledge discovery & data mining. 2020: 2523-2532.

[4] Garcia V, Bruna J. Few-shot learning with graph neural networks[J]. arXiv preprint arXiv:1711.04043, 2017.

---

### Meta-Review · Area_Chair_2wck · 2024-12-14

**Metareview:**

This paper studies Graph Structured Learning from the perspective of mutual information. The authors provided theoretical analysis and empirical results. The authors’ experimental results reveal that GSL methods fail to provide additional mutual information gain than the baselines.

**Strengths:**

1. The paper is well-written with clear takeaway messages and is overall easy to follow
2. It provides terminology for discussion and presents a framework for comparing GSL/GNN methods.
3. The paper provides empirical evaluations and theoretical analysis.

**Weaknesses:**

1. The authors present a too huge (or general) claim “GSL does not provide any additional performance gain”  but the authors studied only a limited setting, classification and KNN graphs. More advanced graph structure learning methods need to be studied. Also, the claim needs to be verified across a wide range of tasks, including graph-level tasks.
2. The main claim is inconsistent with recent efforts that demonstrate the benefits of GSL.
3. Empirical results are provided only for small-scale datasets.
4. More thorough theoretical analysis is needed to support the main claim.

In sum, this paper studies the benefits of GSL from the perspective of mutual information and claims that GSL does not provide any additional performance gain. However, this claim is inconsistent with recent results, and the claim needs to be verified in more diverse settings, including advanced GSL methods, graph-level tasks, and larger datasets.

**Additional Comments On Reviewer Discussion:**

This paper is overall below the borderline. In addition, as Reviewer XwUm mentioned, the major concerns were not sufficiently addressed. The contribution is limited (and somewhat overclaimed) and compared to recently published GSL benchmarks [1, 2], its novelty is also limited.

[1] Li, Zhixun, et al. "GSLB: the graph structure learning benchmark." NeurIPS 2023.
[2] Zhiyao, Zhou, et al. "Opengsl: A comprehensive benchmark for graph structure learning." NeurIPS 2023.

---

### Decision · Program_Chairs · 2025-01-22

Reject